# On Mitigating Affinity Bias through Bandits with Evolving Biased Feedback

**Matthew Faw** [1]   **Constantine Caramanis** [2]   **Jessica Hoffmann** [3]

## Abstract

Unconscious bias has been shown to influence how we assess our peers, with consequences for hiring, promotions and admissions. In this work, we focus on affinity bias, the component of unconscious bias which leads us to prefer people who are similar to us, despite no deliberate intention of favoritism. In a world where the people hired today become part of the hiring committee of tomorrow, we are particularly interested in understanding (and mitigating) how affinity bias affects this feedback loop. This problem has two distinctive features: 1) we only observe the *biased value* of a candidate, but we want to optimize with respect to their *real value* 2) the bias towards a candidate with a specific set of traits depends on the *fraction* of people in the hiring committee with the same set of traits. We introduce a new bandits variant that exhibits those two features, which we call affinity bandits. Unsurprisingly, classical algorithms such as UCB often fail to identify the best arm in this setting. We prove a new instance-dependent regret lower bound, which is larger than that in the standard bandit setting by a multiplicative function of $K$. Since we treat rewards that are *time-varying* and *dependent on the policy's past actions*, deriving this lower bound requires developing proof techniques beyond the standard bandit techniques. Finally, we design an elimination-style algorithm which nearly matches this regret, despite never observing the real rewards.

## 1. Introduction

"Unconscious bias" is a term coined to designate stereotypes (positive or negative) that we hold outside of our awareness. In recent years, numerous studies have argued that unconscious bias is pervasive, and that it shapes even well-meaning individuals' assessment of their peers (FitzGerald & Hurst, 2017; Oberai & Anand, 2018; Pager & Shepherd, 2008; Tate & Page, 2020; Holroyd et al., 2017; Buetow, 2019; Sukhera, 2019). These skewed assessments in turn impact employment (Bertrand & Mullainathan, 2004; Bohnet et al., 2016; Uhlmann & Cohen, 2005; Stoica et al., 2020; Crawford et al., 2018; Somashekhar, 2014). We want to design systemic mitigation strategies which relieve individuals of the difficult task of giving an unbiased assessment, while still leading towards a fairer outcome.

In this work, we choose to focus solely on one of the key aspects of unconscious bias: affinity (or similarity) bias (Huang et al., 2019; Oberai & Anand, 2018; Russell et al., 2019; Clifton et al., 2019). This bias captures the human tendency to favor people who are similar to ourselves, whether it's because of our skill-set, our language, or even the school we attended. We are particularly interested in the feedback loop which naturally arises in hiring: today's hired candidate will be part of tomorrow's hiring committee. Therefore, the more people with a specific set of attributes are hired, the higher the *proportion* of them in future decision processes, which means the stronger the overall affinity bias will be towards this set of attributes.

These types of decision-making processes with feedback loops have been modeled by non-stationary multi-armed bandits (Gittins, 1979; Whittle, 1988; Heidari et al., 2016; Levine et al., 2017; Malik et al., 2022; 2023; Kleinberg & Immorlica, 2018). In this framework, the decision-maker can interact with the system by pulling an arm, and the system can react by adapting its reward based on the past actions. Prior work has studied both stochastic systems and adversarial systems. As we are interested in modeling an ever-present unconscious effect–as opposed to conscious, chosen discriminatory actions–we assume the system reacts in a stochastic way.

One key feature of our problem is that although the perceived rewards evolve, the real reward of each arm remains unchanged. This is in stark contrast to most of the non-stationary bandits literature, in which it is assumed that previous actions change the environment. In our case, the environment remains unchanged, but as the composition of the hiring committee changes, so does its overall un-

---

[1]Georgia Institute of Technology [2]The University of Texas at Austin [3]Google DeepMind. Correspondence to: Matthew Faw <mfaw3@gatech.edu>.

*Proceedings of the 42ⁿᵈ International Conference on Machine Learning*, Vancouver, Canada. PMLR 267, 2025. Copyright 2025 by the author(s).

conscious bias, which leads to variations in the observed rewards. While the decision-maker only observes biased rewards, they want to optimize their actions with respect to the real rewards, which are never observed. The other key feature of our problem is that the biased feedback depends not only on the past actions (which has been studied in (Tang & Ho, 2019; Gaucher et al., 2022; Schumann et al., 2022)), but also on time $t$, since what matters is the *fraction* of times this arm has been selected. Indeed, when someone from a given group (defined as a set of attributes) is hired, not only does this group's relative importance increases, but all the other groups become proportionally less represented.

To gain insights into the effects and mitigation strategies of affinity bias in hiring, we propose a non-stationary bandit setting, in which each arm represents a group of people exhibiting the same set of traits. On each turn, the hiring committee picks an arm, which represents hiring someone with that set of traits. They then observe only the potentially biased feedback of this arm, which relates to the real reward in the following informal way: *the observed rewards (biased feedback) follow the same distribution as the real rewards, except that the average of each arm $i$ is multiplicatively reweighed by $W_i(t)$, which is expressed as a function[1] $f$ of the fraction of time $i$ has been pulled at time $t$, and the initial bias of the system.*

Trivially, if the decision-makers do not gain information about the real reward from the perceived reward, it is impossible for them to minimize the regret with respect to the real reward. We therefore assume that while the hiring committee knows neither the exact function $f$, the initial bias, nor the real reward, it does know that the perceived reward depends on the real reward in the way expressed above. It is important to note that unconscious bias is—by definition—shaping our judgment beyond our awareness. As such, no additional observations of the candidate over the bandit decision-making time-scale after their hiring would help reveal their true value: the bias comes from the assessor's perception, which will potentially take a longer time-scale to overcome. We now move on to our main contributions:

### 1.1. Main contributions

**Affinity bandits model.** We introduce a new variant of non-stationary multi-arm bandits called *affinity bandits*, for which we only observe evolving biased feedback. This biased feedback varies based on the fraction of times each arm has been selected in the past, while the real unobserved reward of each arm remain unchanged. Unsurprisingly, adding this feedback loop makes traditional algorithms (such as UCB or EXP3) incur linear regret.

---

[1]The function $f$ models the unknown relation between the amount of bias and size of the affinity group. The choice of $f(\cdot) = 1$ models an unbiased system.

**New lower bound through new techniques.** We prove this setting is inherently harder than the standard setting by obtaining a lower bound on the regret. This bound holds even in the full information setting, when the exact bias is known, and therefore *results only from the feedback loop effect* (and not, for example, from the lack of information). Compared to the standard regret bound, the regret in our setting incurs at least a multiplicative factor which depends on the total number of groups. We emphasize that the proof of this lower bound requires several new ideas beyond the standard regret lower bound techniques.

**Near optimal algorithm.** We provide an algorithm that attains logarithmic regret, and nearly matches the lower bound. Interestingly, this is a variant of the elimination algorithm, which keeps a set of potentially optimal arms, and play them one after the other until it is certain that it can eliminate some of them. We therefore prove that to compensate for unconscious bias, the strategy which gives a chance to everyone one after the other until enough information is gathered is almost optimal.

### 1.2. Related works

Aside from the tight connections with "non-stationary bandits" and "history-dependent biased bandits" mentioned above, our work is linked to a few other lines of research (see Appendix A for an extended discussion). There is a rich history of fairness-related work with bandits (Joseph et al., 2016; Liu et al., 2017; Gillen et al., 2018; Khalili et al., 2021; Wang et al., 2021), in which the goal is to minimize regret while satisfying some fairness constraints. Our setting could also be seen as a special case of partial monitoring (Rustichini, 1999; Bartók et al., 2014; Lattimore & Szepesvári, 2019; Bartók et al., 2011; Bar-On & Mansour, 2024) with adversarial feedback. However, their regret guarantees do not transfer meaningfully to our setting (see Appendix B for details). Finally, our work build on techniques for bandit lower bound, in particular asymptotic instance-dependent techniques (Lai & Robbins, 1985; Burnetas & Katehakis, 1996) and the framework to obtain bounds based on divergence decomposition (Garivier et al., 2019; Kaufmann et al., 2016). Our work generalizes this framework to handle the challenging setting where observed feedback is time-varying and dependent on the decisions of a policy which potentially knows the bias model exactly.

## 2. Problem Setting

We consider a variant of the $K$-armed stochastic multi-armed bandit problem where each arm $i \in [K]$ represents a group of people exhibiting the same set of traits relevant for the hiring task, e.g. skill-set. This arm is associated with a distribution $\nu_i$ with finite, unknown mean $\mu_i$. A bandit policy $\pi$ interacts with (a transformation of) this environment

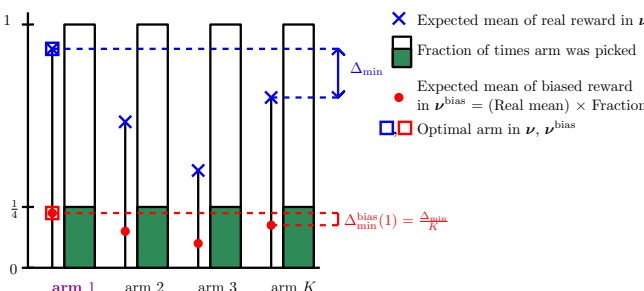

Figure 1: Representation of our setting when each arm has been picked exactly once. The expected biased feedback is the expected real reward divided by $K$. The ordering of the observed rewards is identical to that of the real rewards, but the suboptimality gaps are divided by $K$.

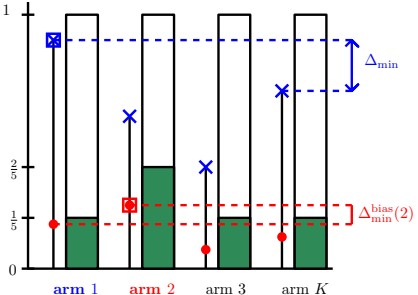

Figure 2: From the setting in Figure 1, we picked arm 2. The biased feedback for arm 2 now appears better than the one for arm 1, the real best arm. Moreover, while the fraction for arm 2 increases, the fraction for all the other arm decreases.

$\boldsymbol{\nu} = (\nu_i)_{i \in [K]}$ over $n$ time steps. At each time $t \leq n$, the policy first selects an arm $A_t \in [K]$, then observes stochastic feedback $Y_t$. The objective of a policy $\pi$ is to minimize the (pseudo-)regret:

$$R_{\boldsymbol{\nu}, \pi}(n) = \max_{i^* \in [K]} \mathbb{E} \left[ \sum_{t \in [n]} X_{i^*, t} - X_{A_t, t} \right]$$
$$= \sum_{i \in [K]} \Delta_i \mathbb{E} \left[ T_i(n) \right], \qquad (1)$$

where each $X_{i,t} \sim \nu_i$ is an (unobserved) sample from arm $i$'s associated distribution, $\Delta_i = \max_{i^* \in [K]} \mu_{i^*} - \mu_i$ is the suboptimality gap of arm $i$, and $T_i(n) = \sum_{t=1}^{n} \mathbb{1}\{A_t = i\}$ is the number of times arm $i$ is played from time 1 to $n$. As this objective depends on the *unobserved* $X_{i,t}$, not the feedback $Y_t$, achieving sublinear regret is not possible without an assumption on the feedback. We aim to model feedback systems with the following features:

1. The system has an initial, perhaps misleading, affinity for each arm.
2. Pulling an arm increases the system's affinity towards that arm.
3. Pulling an arm slightly decreases the system's affinity towards other arms[2].

We adopt a fairly general model on the feedback $Y_t$ capturing the essence of these features:

**Assumption 2.1** (Feedback model). At each time $t$, upon pulling an arm $A_t \in [K]$, the policy observes feedback $Y_t$ sampled from a distribution satisfying:

$$\mathbb{E} [Y_t \mid \mathcal{F}_{t-1}] = \mu_{A_t} W_{A_t}(t)$$
$$(Y_t - \mathbb{E} [Y_t \mid \mathcal{F}_{t-1}]) \text{ is 1-subGuassian}, \qquad (2)$$

where $\mathcal{F}_t$ is the filtration of observations $(A_s, Y_s)_{s \leq t}$ until

[2]This represents the relative affinity of a group slightly decreasing when the size of the hiring committee increases without the size of the group increasing.

$t$, and $W_i(t)$ is a multiplicative reweighting of arm $i$'s mean $\mu_i$. We assume this multiplicative reweighting satisfies:

$$W_i(t) \triangleq f \left( \frac{T_i^0 + T_i(t-1)}{t_0^{\text{bias}} + t - 1} \right) \triangleq f \left( \frac{T_i^{\text{bias}}(t-1)}{t^{\text{bias}} - 1} \right) \quad (3)$$

for some $T_i^0 \geq 1$, $t_0^{\text{bias}} = \sum_{i \in [K]} T_i^0$, and function $f(x)$ which is bounded on $(0, 1)$, non-decreasing, and $L$-Lipschitz for $x \in (0, 1)$. In other words, the reweighting $W_i(t)$ is a function of the total fraction of times arm $i$ has been pulled.

**Important features of feedback model:**

**(i) Generalizes subGaussian bandits.** Our setting subsumes the standard subGaussian bandit setting. Indeed, notice that $f(x) = 1$ and $Y_t = X_{A_t, t}$ where $X_{i,t} - \mu_i$ is 1-subGaussian satisfies Assumption 2.1.

**(ii) Admits polynomial bias functions.** Our setting allows the mean of $Y_t$ to scale with the fraction of times the selected arm $A_t$ has been played, or indeed any bounded polynomial of this fraction. More precisely, for any $\alpha \geq 1$, our model captures $f(x) = x^\alpha$ since this choice is $\alpha$-Lipschitz, increasing, and bounded in $[0, 1]$ for $x \in [0, 1]$.

**(iii) Extends beyond polynomial biases.** Our assumptions on $f(x)$ are more general than simply functions of the form $x^\alpha$. For example, the sigmoid function $f(x) = (1 + \exp(-x))^{-1}$ is $1/4$-Lipschitz, increasing, and bounded between $[1/2, 1)$ for $x \in [0, 1]$. Further, for any function $f(x)$ satisfying Assumption 2.1, the functions $\min \{c_1, f(x)\}$ and $\max \{c_2, f(x)\}$ also satisfy Assumption 2.1 for any $c_1, c_2 \in [0, 1]$.

**(iv) Allows additive, multiplicative, and random reward transformations.** Concretely, if $X_{i,t} - \mu_i$ is 1-subGaussian, then all of the following choices of feedback $Y_t$ satisfy Assumption 2.1: (a) $Y_t = W_{A_t}(t) X_{A_t, t}$, (b) $Y_t = X_{A_t, t} + \mu_{A_t}(W_i(t) - 1)$, and (c) $Y_t = B_t X_{A_t, t}$ where $B_t$ is Bernoulli with mean $W_i(t)$ (conditionally) independent of $X_{A_t, t}$.

**(v) Allows dependence on initial biases.** The parameters $T_i^0$ correspond to the "initial bias" of the feedback system. These parameters can, for instance, make initial feedback for the optimal arm appear very small, and initial feedback for a suboptimal arm appear very large. Indeed, if $T_1^0 \gg T_2^0$ and $W_i(t) = \frac{T_i^{\text{bias}}(t-1)}{t^{\text{bias}}-1}$, then on average, the feedback for arm 1 will initially appear larger than that of arm 2, even if $\mu_2 > \mu_1$.

## 3. Why is the problem difficult?

**Ignoring the bias leads to linear regret.** One may wonder if naïvely ignoring the bias model and running a standard bandit algorithm such as UCB or EXP3 could still achieve sublinear regret in this setting. Unfortunately, these algorithms suffer linear regret. Indeed, we prove in Theorem E.1 that UCB suffers linear regret on a simple 2-armed Bernoulli bandit instance with constant suboptimality gaps. Empirically, we observe this same phenomenon for UCB (Auer et al., 2002), EXP3 (Auer et al., 1995), and EXP3-IX (Kocák et al., 2014) in Figure 3[4].

At a high level, these algorithms fail because the bias structure can make a suboptimal arm appear empirically optimal at early time steps. These standard algorithms will therefore start by favoring the suboptimal arm in the early stages. However, the more an arm is played, the better its observed mean appear, despite its real mean remaining unchanged. This leads the algorithms to continue to select the suboptimal arm, incurring linear regret and even failing at best-arm identification with constant probability. See Appendix F for more details and comparisons to other UCB variants.

**Exploding variance when "unbiasing" the feedback.** Even if the learner knew the feedback model exactly, it could obtain unbiased samples from the true reward distributions by multiplying the observed feedback by the inverse reweighting $W_{A_t}(t)^{-1}$. However, this operation scales up the variance by $W_{A_t}(t)^{-2} = f\left(T_{A_t}^{\text{bias}}(t-1)/(t^{\text{bias}}-1)\right)^{-2}$ by Assumption 2.1. Thus, for arms which have been played infrequently, obtaining an unbiased sample comes at the cost of potentially large variance (since $f(x)$ is nondecreasing in $x$). The fact that the variance (as well as, potentially,

---

[4]We report the results for a 2-armed Bernoulli bandit environment with $\mu_1 = .4 < .6 = \mu_2$ and with bias model $W_i(t) = T_i^{\text{bias}}(t-1)/(t^{\text{bias}}-1)$. Each datapoint is the average of 60 repeats with time horizon $n = 2 \cdot 10^4$, $T_2^{\text{bias}} = 10$, and $T_1^{\text{bias}}$ varying from 1 and 200.

[4]We remark that the standard UCB-V algorithm assumes all rewards are bounded on the interval $[0, b]$ for a known constant $b$. However, our implementation debiases the feedback $Y_t$ by obtaining unbiased estimates of the true rewards, $Z_{t,A_t} = Y_t W_{A_t}(t)^{-1}$, which has unbounded support. Our implementation of this algorithm adaptively estimates an upper bound for $Z_{t,A_t}$ based on the observed samples. See Appendix F for details.

---

**Algorithm 1** Elimination algorithm for unknown bias model

---
**Require:** Time horizon $n \in \mathbb{N}$, sampling schedule $m_r \approx \log(n)/\widetilde{\Delta}_r^2$, where $\widetilde{\Delta}_r = 2^{-r}$.
  Let $\tau_0 = 0$, $t = 1$ and $\mathcal{A}_1 = [K]$
  **for** $r = 1, 2, \ldots$ **do**
    **for** $\ell \in [\lfloor m_r \rfloor]$, $i \in \mathcal{A}_r$ in increasing order of index **do**
      Pull arm $i$, receive feedback $Y_t$, update $t \leftarrow t + 1$.
    **end for**
    Compute $\widehat{\mu}_i(r)$, the empirical average of the feedback for arm $i$ observed during round $r$
    Update active arms:
    $\mathcal{A}_{r+1} = \left\{ i \in \mathcal{A}_r : \max_{j \in \mathcal{A}_r} \widehat{\mu}_j(r) - \widehat{\mu}_i(r) \leq \widetilde{\Delta}_r \right\}$
    Mark $\tau_r$ as the end time of round $r$
  **end for**

---

the support of the debiased samples) is time-varying and can potentially scale polynomially in the time horizon invalidates or trivializes standard regret guarantees for many bandit algorithms (e.g., UCB-V (Audibert et al., 2007) and EXP3 (Auer et al., 1995)). Since it rescales the feedback by $W_{A_t}(t)^{-1}$ to obtain unbiased samples, UCB-V has significantly larger regret scaling and deviations than in the standard, unbiased stochastic feedback setting, as can be seen on Figure 4[5].

**Lower bound for *known* bias model but upper bound for *unknown*.** One notable feature of our lower bound is that it holds even when the bias model $f(\cdot)$ and initial biases $T_i^0$ from Assumption 2.1 are known exactly to an algorithm. Recall, however, that we aim to design an algorithm for settings where the bias model and initial biases are unknown. Theorem 5.2 thus gives us an ambitious (yet, as we show in Theorem 4.1 and Corollary 5.4, nearly-tight) regret scaling target.

## 4. Regret upper-bounds for unknown bias model

Here, we study the phased-elimination style algorithm (essentially the algorithm from (Auer & Ortner, 2010)) described in Algorithm 1. We show that, even when the bias model $f\left(T_i^{\text{bias}}(t-1)/(t^{\text{bias}}-1)\right)$ is unknown to the algorithm, logarithmic instance-dependent regret bounds are possible, assuming the time horizon is known and sufficiently large.

Before establishing the regret guarantee, let us first give some intuition for why the algorithm should work. Algorithm 1 proceeds in rounds $r \geq 1$. At each round, the objective of the algorithm is to eliminate all arms whose av-

---

[5]One can observe the impact of this scaling issue on a 2-armed Bernoulli instance, with $\mu_1 = .4 < .6 = \mu_2$ and bias model $W_i(t) = T_i^{\text{bias}}(t-1)/(t^{\text{bias}}-1)$. We show 40 sample paths for $n = 2 \cdot 10^5$ time steps. $T_1^{\text{bias}} = 100$, $T_2^{\text{bias}} = 10$.

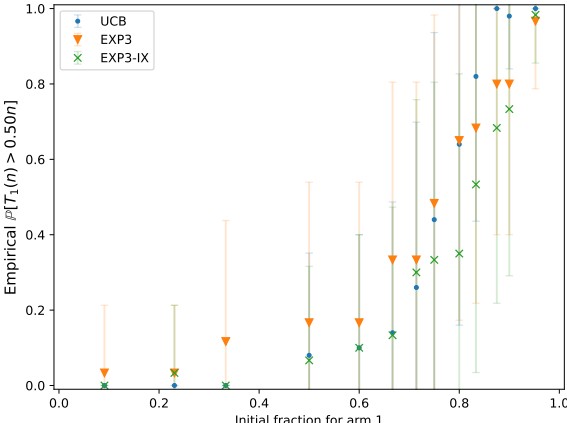

Figure 3: Empirical probability of the suboptimal arm being pulled by more than $1/2$ of the time horizon as a function of the initial bias. We show results for UCB, EXP3 and EXP3-IX. For high initial weight on the suboptimal arm, all three algorithms are more likely to pull it more than the optimal arm. Moreover, even for high weight on the optimal arm, the probability that the suboptimal arm is pulled more than the optimal arm can be bounded away from 0.

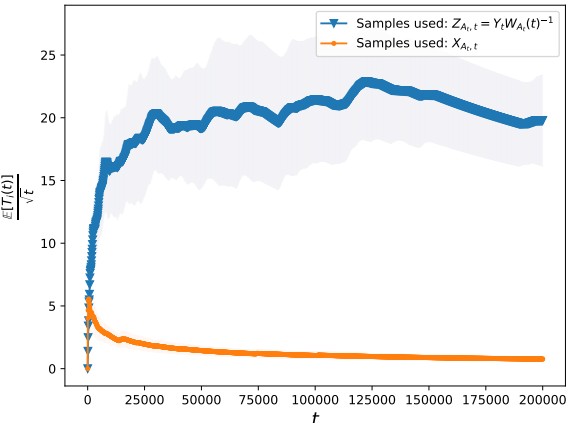

Figure 4: The number of times the suboptimal arm is pulled as a function of time for UCB-V[3] in two environments, normalized by $\sqrt{t}$. In the first, UCB-V receives the true rewards $X_{A_t,t}$ as samples. In the second environment, UCB-V receives "debiased" feedback $Y_t W_{A_t}(t)^{-1}$ as samples. While we cannot conclude whether the regret of UCB-V grows as $\sqrt{t}$ from this graph, it is unlikely it grows as $\log(t)$.

erage feedback is smaller than the largest average feedback by an additive factor of $\widetilde{\Delta}_r = 2^{-r}$. The main challenge is to guarantee that this rule, with sufficiently high probability, eliminates only the suboptimal arms. Notice that, for any two arms $i_*$ and $i$,

$$\mathbb{E}\left[\widehat{\mu}_{i_*}(r) - \widehat{\mu}_i(r) \mid \mathcal{F}_{r-1}\right] = \mu_{i_*}\bar{W}_{i_*}(r) - \mu_i\bar{W}_i(r)$$
$$= (\mu_{i_*} - \mu_i)\bar{W}_{i_*}(r) \quad (4)$$
$$+ \mu_i(\bar{W}_{i_*}(r) - \bar{W}_i(r)), \quad (5)$$

where in the above, we employ a slight abuse of notation by taking $\mathcal{F}_r$ to be the filtration of observations until the end of round $r$, and $\bar{W}_i(r) = 1/m_r \sum_{t=\tau_{r-1}+1}^{\tau_r} W_i(t)\mathbb{1}\{A_t = i\}$ is the average reweighting of arm $i$ over round $r$.

The first term in the above decomposition (4) is essentially a reweighted suboptimality gap between arms $i_*$ and $i$ with the same sign as the difference in true means between these two arms. The second term (5), however, is a non-zero bias term which can be positive or negative, and could potentially cause the optimal arm to be eliminated. Fortunately, one can show (see Lemma C.7) that, under Assumption 2.1, that (5) is bounded by:

$$|\mu_i(\bar{W}_{i_*}(r) - \bar{W}_i(r))| \lesssim \widetilde{\Delta}_r^2 L\left(1 + \frac{T_{\max}^0 - T_{\min}^0}{K}\right)\frac{\log(\log(n))}{\log(n)}$$

where $T_{\max}^0 - T_{\min}^0$ is the gap between the largest and smallest initial number of arm pulls. This establishes that, for sufficiently large $n$, the bias term (5) scales as $\widetilde{\Delta}_r^2 \ll \widetilde{\Delta}_r$, and hence is negligible relative to the elimination criterion of Algorithm 1. This is the key insight to proving that the al-

gorithm achieves a sublinear regret guarantee. In particular, we establish the following:

**Theorem 4.1** (Regret guarantee for Algorithm 1; Simplified version of Theorem C.4 and Corollary C.6)**. *Suppose that Algorithm 1 is run for $n$ time-steps in an environment $\boldsymbol{\nu}$ with bias model satisfying Assumption 2.1 with Lipschitz constant $L$ and $\mu_i \in [0,1]$ for all $i \in [K]$, using the sampling schedule $m_r = 2^{2r+6}\log(\frac{12}{\pi^2}K^2r^2n)$. Further, suppose $n$ is sufficiently large such that $\log(nK)/\log(\log(nK)) \gtrsim L(1 + (T_{\max}^0 - T_{\min}^0/K))$, and $T_{\max}^0 \lesssim \log(Kn)$. Then, the regret of Algorithm 1 satisfies the following two bounds:*

$$R_{\boldsymbol{\nu},\pi}(n) \lesssim f\left(1/15K\right)^{-2}\sum_{i:\Delta_i>0}\log(n)/\Delta_i$$

*and*

$$R_{\boldsymbol{\nu},\pi}(n) \lesssim f\left(1/15K\right)^{-1}\sqrt{Kn\log(n)}.$$

## 5. Asymptotic instance-dependent lower bound

To characterize the fundamental difficulties of our problem setting, we derive instance-dependent lower bounds on the performance of any "consistent" bandit policy. Our notion of consistency in Definition 5.1 is a finite-time adaptation of similar (asymptotic) notions of consistency from the bandits literature (e.g., (Lai & Robbins, 1985, Eq. (1.8)) and (Burnetas & Katehakis, 1996, UF Policy)).

**Definition 5.1** (Consistent policy)**. Let $\mathcal{E}$ be a set of unbiased bandit environments $\boldsymbol{\nu}$ with bias model following Assumption 2.1 with a *fixed and common* set of initial biases $\{T_i^0\}_{i\in[K]}$ and reweighting function $W_i(t) =$

$f\left(T_i^{\text{bias}}(t-1)/(t^{\text{bias}}-1)\right)$. We call a family of bandit policies $\{\pi_n\}_{n\geq1}$ *consistent* for an environment class $\mathcal{E}$ (with its associated bias model) if there are constants[6] $C > 0$ and $a \in (0,1)$ such that, for all $n \geq 1$ and $\boldsymbol{\nu} \in \mathcal{E}$, the regret of policy $\pi_n$ is bounded as: $R_{\boldsymbol{\nu},\pi_n}(n) \leq C \cdot n^a$.

When the dependence of $\pi_n$ on $n$ is clear from context, we adopt a slight abuse of notation and call the policy $\pi$ consistent.

Before continuing, we emphasize a couple key features of our notion of consistency, which differ slightly from those in prior literature. Consistent policies $\pi_n$ (as per Definition 5.1) may (potentially) know both the time horizon and bias model *exactly*. (i) The fact that the policy may know the time horizon and satisfy consistency is crucial – indeed, it is unclear if there exists *any* policy with sublinear regret when the time horizon is unknown.[7] (ii) Moreover, the fact that the lower bound holds against policies which know the bias model exactly makes the guarantee quite strong, and reflects the fact that our setting is fundamentally harder than a standard stochastic bandits problem, even when the bias model is exactly known. (iii) Finally, we emphasize that Definition 5.1 is non-vacuous, as *there exist policies satisfying the definition* (e.g, Algorithm 1 with regret guarantee in Theorem 4.1).

We state the following lower bound and proof sketch under the assumption that $\Delta_{\max}/\Delta_{\min} = \mathcal{O}(1)$ for simplicity of exposition. For the more general statement under less restrictive conditions, see the more general statement and proofs in Appendix D.

**Theorem 5.2** (Informal statement of Theorem D.1). *Fix any $K > 1$, time horizon $n$, and bias model satisfying Assumption 2.1. Let $\pi$ be a consistent policy for the class of Gaussian environments with suboptimality gaps $\Delta_i \leq 1$ per Definition 5.1. Then, for any such environment $\boldsymbol{\nu}$ for which $\Delta_{\max}/\Delta_{\min} = \mathcal{O}(1)$ and $-\log(f(4/K)) = \mathcal{O}(\log(K))$, the policy $\pi$ must suffer regret at least:*

$$R_{\boldsymbol{\nu},\pi}(n) \gtrsim f\left(\mathcal{O}(\log(K)/K)\right)^{-2} \sum_{i:\Delta_i>0} \log(n)/\Delta_i,$$

*Remark* 5.3 (Comparison to standard bandit regret lower bound). We note that in the standard, unbiased setting, Lai & Robbins (1985) established a lower bound for Gaussian bandits of the form $R_{\boldsymbol{\nu},\pi}(n) \geq \sum_{i:\Delta_i>0} \frac{2\log(n)}{\Delta_i} - \mathcal{O}(1)$. They

also gave an algorithm with regret asymptotically matching their lower bound. Our lower bound shows that, at least in the setting where the maximum ratio of (nonzero) suboptimality gaps is not too small, then the regret in the biased setting we study must be at least a factor (roughly) $f\left(\mathcal{O}(\log(K)/K)\right)^{-2}$ larger than in the standard setting.

As a consequence of Theorem 5.2, one can show that, under a mild additional condition on the bias function $f(\cdot)$, our regret guarantee for Algorithm 1 is optimal up to $\operatorname{poly}\log(K)$ factors.

**Corollary 5.4** (Comparison of Theorem 4.1 and Theorem 5.2). *Under the conditions in Theorems 4.1 and 5.2a, suppose additionally that the bias model satisfies the following: for any $x \in (0,1)$ and $\mu \in (1, 1/x)$, there is a constant $L' > 0$ such that $f(\mu x) \leq \mu^{L'} f(x)$. Then, for sufficiently large time horizons, the regret bound of Algorithm 1 in Theorem 4.1 matches Theorem 5.2 up to a multiplicative $\mathcal{O}(\log(K)^{2L'})$ factor.*

We briefly interpret Corollary 5.4 with some examples. When $f(x) = x^\alpha$ for some $\alpha \geq 1$, then Corollary 5.4 implies the regret bounds are tight up to an $\mathcal{O}(\log(K)^{2\alpha})$ factor. Moreover, any non-decreasing reweighting function $f(x)$ which is upper and lower bounded by constant degree polynomials in $x$ similarly satisfy optimality up to a $\operatorname{poly}\log(K)$ factor. Finally, notice that some reweighting functions such as the sigmoid function $f(x) = (1 + \exp(-x))^{-1}$ are tight up to constant factors, since $f(x) \in [1/2, 1]$.

### 5.1. Proof sketch

Here, we give a sketch of the proof of Theorem 5.2. A complete proof with all formal statements can be found in Appendix D. In the following proof sketch, to reduce clutter, we will use $\mathbb{E}[\cdot], \mathbb{E}^{(i)}[\cdot]$ to denote expectation w.r.t. the observations of a policy $\pi$ in environment $\boldsymbol{\nu}, \boldsymbol{\nu}^{(i)}$ (and similarly for probabilities). Further, for a measure $\mathbb{P}$ on the filtration $\mathcal{F}_n = \sigma(\mathcal{H}_n)$ generated by the $n$-round observation history $\mathcal{H}_n = (A_t, Y_t)_{t\in[n]}$, we will denote $\mathbb{P}^{\mathcal{H}_\tau}$ for $\tau \leq n$ to be the pushforward measure of the observation history until $\tau$, $\mathcal{H}_\tau$, under $\mathbb{P}$.

Consider any bandit policy $\pi$ interacting in an environment $\boldsymbol{\nu}$ with a bias model satisfying Assumption 2.1. To obtain a lower bound on the regret $R_{\boldsymbol{\nu},\pi}(n)$, we construct a set of alternative environments $\boldsymbol{\nu}^{(i)}$ for each suboptimal arm in $\boldsymbol{\nu}$, such that arm $i$ is optimal under $\boldsymbol{\nu}^{(i)}$. To obtain sublinear regret in environment $\boldsymbol{\nu}$ and $\boldsymbol{\nu}^{(i)}$ simultaneously (as is mandated by the consistency condition in Definition 5.1), the policy $\pi$ must pull arm $i$ sufficiently many times to distinguish between these two environments. Due to the feedback model from Assumption 2.1, however, pulling an arm $i$ decreases the mean of the feedback distribution for *every*

---

[6]Note that this constant $C$ may depend on the common environment parameters of $\mathcal{E}$ such as $K$, the initial biases $T_i^0$, and the bias function $f(\cdot)$.

[7]One might hope to apply the standard "doubling trick" (Auer et al., 1995) used to convert an algorithm which knows the time horizon to one which do not (while essentially preserving the regret guarantee of the known time-horizon algorithm). Unfortunately, this trick does not apply to our setting, since the feedback observed by the algorithm depends on the entire observation history.

*other arm*. We show that this feedback structure implies a significant strengthening of the stochastic bandit lower bound.

**Step 1: From lower-bounding regret to lower bounding the number of arm pulls.**

Recall that, by the standard regret decomposition (1), we may write: $R_{\boldsymbol{\nu},\pi}(n) = \sum_{i:\Delta_i > 0} \Delta_i \mathbb{E}\left[T_i(n)\right]$. Thus, to lower bound $R_{\boldsymbol{\nu},\pi}(n)$, it suffices to lower bound $\mathbb{E}\left[T_i(n)\right]$ for each suboptimal arm $i$. Since we assume for the simplicity of this proof sketch that $\Delta_{\max}/\Delta_{\min} = \mathcal{O}(1)$, it thus suffices to find a set of arms $S \subset [K]$ such that $|S| = \Omega(K)$ and (roughly), for some $\beta \lesssim \log(K)$:

$$\mathbb{E}\left[T_i(n)\right] \gtrsim f\left(\beta/K\right)^{-2}\log(n)/\Delta_{\max}^2 \ \forall i \in S. \qquad (6)$$

**Step 2: Relating the divergence decomposition to a "biased proxy" for $\mathbb{E}\left[T_i(n)\right]$.**

A standard technique for lower-bounding $\mathbb{E}\left[T_i(n)\right]$ is via the KL-divergence decomposition. Indeed, in the standard unbiased stochastic setting, the KL-divergence between the observation histories in two environments $\boldsymbol{\nu}, \boldsymbol{\nu}^{(i)}$ differing only in arm $i$ takes the following convenient form:

$$D_{\mathrm{KL}}(\mathrm{Pr}^{\mathcal{H}_n} \| \mathrm{Pr}^{(i),\mathcal{H}_n}) = D_{\mathrm{KL}}(\nu_i \| \nu_i^{(i)}) \mathbb{E}\left[T_i(n)\right].$$

See, e.g., (Auer et al., 1995, Eq. (17)) or (Lattimore & Szepesvári, 2020, Lemma 15.1) for a reference. Under the feedback model of Assumption 2.1, however, the KL-divergence depends on a "biased proxy" $\mathbb{E}\left[\sum_t W_i(t)^2 \mathbb{1}\{A_t = i\}\right]$ instead of directly on $\mathbb{E}\left[T_i(n)\right]$. In particular, we have the following:

**Lemma 5.5** (A divergence decomposition for biased environments; Informal statement of Lemma D.3). *Fix a time horizon $n \geq 1$, a Gaussian bandit environment with suboptimal arm $i$ satisfying Assumption 2.1 and a policy $\pi$ satisfying Definition 5.1. Let $\boldsymbol{\nu}^{(i)}$ be a Gaussian bandit environment (with the same bias model) such that $\nu_j = \nu^{(j)}$ for all $j \neq i$, and $\boldsymbol{\nu}^{(i)}$ has mean $\mu_i^{(i)} = \mu_i + (1 + \varepsilon)\Delta_i$ for some $\varepsilon > 0$. Then, for any stopping time $\tau_i \leq n$,*

$$\frac{(1+\varepsilon)^2 \Delta_i^2}{2} \mathbb{E}\left[\sum_{t \in [\tau_i]} W_i(t)^2 \mathbb{1}\{A_t = i\}\right]$$
$$= D_{\mathrm{KL}}(\mathrm{Pr}^{\mathcal{H}_{\tau_i}} \| \mathrm{Pr}^{(i),\mathcal{H}_{\tau_i}})$$
$$\gtrsim \frac{\mathbb{E}\left[\tau_i\right]}{n}\log(n) - \mathcal{O}(1).$$

Lemma 5.5 has two parts. The equality in Lemma 5.5 is the generalization of the divergence decomposition to the biased feedback setting. As discussed above, instead of directly equating the KL-divergence to $\mathbb{E}\left[T_i(\tau_i)\right]$, this identity relates the KL-divergence to a "biased proxy" for this quantity, depending on the multiplicative biases $W_i(t)$.

The inequality in Lemma 5.5 is the consequence of a data-processing inequality on the KL-divergence (in a similar spirit to the argument in (Garivier et al., 2019, Eq. (8))). For more details, refer to the proof in Appendix D.

**Step 3: Relating the "biased proxy" with $\mathbb{E}\left[T_i(\tau_i)\right]$ via stopping times**

Inspecting Lemma 5.5, we observe that *if* we could find a stopping time $\tau_i$ such that, simultaneously (i) the multiplicative reweighting $W_i(t) \leq f\left(\beta/K\right)$ for all $t < \tau_i$ and (ii) $\mathbb{E}[\tau_i]/n = \Omega(1)$, then we could conclude that:

$$\frac{(1+\varepsilon)^2 \Delta_i^2}{2} f\left(\frac{\beta}{K}\right)^2 \mathbb{E}\left[T_i(\tau_i)\right] \gtrsim \log(n) - \mathcal{O}(1).$$

The above would immediately imply our claimed regret lower bound, through the regret decomposition (1). The following claim gives a construction for $\tau_i$ which will (essentially) satisfy (i). We will soon see that this construction also satisfies (ii).

**Claim 5.6** (Consequence of the Divergence Decomposition; Simplified version of Claim D.4). *Consider the same setting as in Lemma 5.5, where arm $i$ is suboptimal in $\boldsymbol{\nu}$. Fix $n_0 \approx \log(n)/12\Delta_{\max}^2$, $\beta \approx \log(K)$, and define:*

$$\tau_i = \min\left\{t \geq n_0 : T_i^{\mathrm{bias}}(t)/t^{\mathrm{bias}} \gtrsim \beta/K \text{ or } t = n\right\}.$$

*Then, denoting $T_i(a,b) = T_i(b) - T_i(a)$, we have that:*

$$\mathbb{E}\left[\sum_{t \in [\tau_i]} W_i(t)^2 \mathbb{1}\{A_t = 1\}\right] \leq n_0 + f\left(\frac{\beta}{K}\right)^2 \mathbb{E}\left[T_i(n_0, \tau_i)\right].$$

*In particular, this implies that for any policy satisfying $\mathbb{E}\left[\tau_i\right] = \Omega(n)$:*

$$\mathbb{E}\left[T_i(n_0, \tau_i)\right] \gtrsim f\left(\beta/K\right)^{-2}\left(\log(n)/\Delta_i^2 - \mathcal{O}(1)\right).$$

Claim 5.6 follows from the definition of the stopping time $\tau_i$, together with the conditions on $W_i(t)$ from Assumption 2.1. To utilize this claim, however, we must show that $\mathbb{E}\left[\tau_i\right] = \Omega(n)$.

From Claim 5.6, we see that we can obtain a refined upper-bound on the "biased proxy" using stopping times. Thus, recall that we choose $n_0 \approx \log(n)/12\Delta_{\max}^2$. If we can show that $\mathbb{E}\left[\tau_i\right] \geq n/6$ for a constant fraction of arms, then Claim 5.6 would give our desired regret lower bound.

**Step 4: Lower-bounding $\mathbb{E}\left[\tau_i\right]$ for many arms**

Recall from Claim 5.6 that $\tau_i$ is the first time after $n_0$ when arm $i$ is played $> \beta/K$ fraction of the time (or $n$, in the case that this event does not occur). To complete our lower bound argument, by Claim 5.6, it suffices to show that, for *any* policy $\pi$ (satisfying Definition 5.1), $\mathbb{E}\left[\tau_i\right] = \Omega(n)$. Notice that, by definition of $\tau_i$, the event that $\tau_i = n$ is

equivalent to $T_i^{\mathrm{bias}}(t)/t^{\mathrm{bias}} \leq \beta/K$ for all $t \in [n_0, n)$. Thus, we have:

$$
\begin{aligned}
\mathbb{E}\left[\tau_i\right] &\geq n\Pr\left[\tau_i = n\right] \\
&= n\Pr\left[T_i^{\mathrm{bias}}(t)/t^{\mathrm{bias}} \leq \beta/K \; \forall t \in [n_0, n)\right]. \quad (7)
\end{aligned}
$$

Hence, it suffices to show that the above probability is constant.

Unfortunately, since the quantity $T_i^{\mathrm{bias}}(t)/t^{\mathrm{bias}}$ is inherently policy-specific, it is infeasible to make deterministic statements about this quantity for every arm. Fortunately, however, obtaining our lower bound requires bounding $\mathbb{E}\left[\tau_i\right]$ only for a constant fraction of suboptimal arms $i$. Thus, through deterministic pigeonholing arguments, we identify properties on the quantities $T_i^{\mathrm{bias}}(t-1)/(t^{\mathrm{bias}}-1)$ (for a sufficiently large subset of arms) which hold deterministically for any algorithm. The first is the observation that, for any time $t$, many arms must have small $T_i^{\mathrm{bias}}(t-1)/(t^{\mathrm{bias}}-1)$.

**Lemma 5.7** (Size of the small bias set). *Consider a bandit policy $\pi$ interacting in an environment $\nu$. Let us denote, for any $t \in [n]$ and $\beta > 1$,*

$$
S_t(\beta) = \left\{i \in [K] : T_i^{\mathrm{bias}}(t)/t^{\mathrm{bias}} \leq \beta/K\right\}.
$$

*Then, $|S_t(\beta)| \geq (1 - 1/\beta)K$.*

While Lemma 5.7 guarantees that $|S_t(\beta)| = \Omega(K)$ for every fixed $t$, it does not guarantee that the arms in $S_t(\beta)$ stay the same as $t$ changes. Moreover, the set $S_t(\beta)$ is random. Ideally, we would like to use Lemma 5.7 to conclude that some fixed set $\tilde{S}$ of $\Omega(K)$ arms such that, for every $i \in \tilde{S}$:

$$
\Pr\left[T_i^{\mathrm{bias}}(t-1)/(t^{\mathrm{bias}}-1) \leq \beta/K \; \forall t \in [n_0, n)\right] = \Omega(1).
$$

Indeed, together with Claim 5.6 with (7), this would imply our desired lower bound.

There is, however, a flaw above – we cannot give any unconditional nontrivial guarantee on the number of arms for $\tilde{S}$. Indeed, at time $n_0 \approx \log(n)/12\Delta_{\max}^2$, it might be the case that the policy $\pi$ identifies all arms $i \in S_{n_0}(\beta)$, pulling them until $T_i^{\mathrm{bias}}(t-1)/(t^{\mathrm{bias}}-1) > \beta/K$. In this way, an algorithm could guarantee that $\tilde{S} = \emptyset$. However, there is a cost to the policy removing many arms from $S_{n_0}(\beta)$. Indeed, playing any fixed arm decreases the fraction of times that all other arms have been played. Thus, one can show that removing each successive arm from $S_{n_0}(\beta)$ requires playing the arm more than the previously removed one, and this cost scales with $n_0$ (see Lemma D.8 for a precise statement).

Motivated the above discussion, we show that, for any policy $\pi$, there is a (random) set of arms $B$ with $|B| = \Omega(K)$ such that, for each arm $i \in B$, one of two conditions must be satisfied: either (i) the arm has been played less than a $\mathcal{O}(1/K)$ fraction of time, or (ii) it has been pulled more than

our desired lower bound on this quantity. Notice that, if $B$ were not random, then we could appeal to Claim 5.6 to conclude with a stronger lower bound on $\mathbb{E}\left[T_i(n)\right]$ for each of the $\Omega(K)$ arms.

**Lemma 5.8** (A small bias set which is stable over time; informal statement of Lemma D.6). *Let $\pi$ be any bandit policy interacting in an environment $\nu$ satisfying Assumption 2.1 with the reweighting function $-\log(f(4/K)) = \mathcal{O}(\log(K))$ and suboptimality gaps satisfying $\Delta_{\max}/\Delta_{\min} = \mathcal{O}(1)$. Let $n_0 \approx \log(n)/\Delta_{\max}^2$ and $\beta \approx \log(K)$. Then, there exists a set of arms $B \subseteq S_{n_0}(\beta)$ such that $|B| \geq K/2$, and each arm $i \in B$ satisfies one of the following:*

***Case 1.*** *$\frac{T_i^{\mathrm{bias}}(t)}{t^{\mathrm{bias}}} \lesssim \frac{\beta}{K} \; \forall t \in [n_0, n)$.*
***Case 2.*** *$T_i(n_0, n) \gtrsim f\left(\frac{4}{K}\right)^{-2} \frac{\log(n)}{\Delta_i^2}$.*

Intuitively, Lemma 5.8 tells us that, for any bandit policy $\pi$, for a large (possibly random) set of arms $B$, one of two things can happen. In Case 1, an arm $i \in B$ is played roughly $\widetilde{\mathcal{O}}(1/K)$ times. Ignoring the fact that $B$ is random, these are the arms for which the stronger lower bound from Claim 5.6 applies. In Case 2, $T_i(n)$ must already be larger than the desired regret lower-bound (recall that we assume $\Delta_{\max}/\Delta_{\min} = \mathcal{O}(1)$ for this proof sketch.

In order to apply the arguments from above, we need to translate the guarantees from Lemma 5.8, which hold for a *random* set of arms $B$, to a guarantee that either Case 1 or 2 happens with constant probability for a deterministic set of arms $B'$ of a similar size. As it turns out, this can be accomplished via a pigeonholing argument. In particular, we can obtain the following "derandomization" of Lemma 5.8:

**Lemma 5.9** (A derandomization of Lemma 5.8; Informal statement of Lemma D.11). *There exists a (deterministic) set $B' \subseteq [K]$ such that $|B'| \geq K/4$ and, for each $i \in B'$, one of the following holds:*

***Case 1'.*** *$\mathbb{E}\left[\tau_i\right] \geq \frac{n}{6}$.*
***Case 2'.*** *$\mathbb{E}\left[T_i(n_0, n)\right] \gtrsim f\left(\frac{4}{K}\right)^{-2} \frac{\log(n)}{\Delta_i^2}$.*

At a high level, Lemma 5.9 follows from Lemma 5.7 as follows: We show, via a pigeonholing argument, that since Lemma 5.7 is true for a random set $B$, there exists a deterministic set $B'$ (of size roughly half of $B$) such that either Case 1 or Case 2 happens with constant probability for each arm $i \in B'$. Since one of these two events happens with constant probability, we can translate each of these cases into a corresponding in-expectation condition (see Cases 1' and 2').

With this deterministic guarantee, the proof is immediate: for each arm in $B'$, if the first case holds with constant probability, then, by definition of $\tau_i$, we have $\mathbb{E}\left[\tau_i\right] = \Omega(n)$. In this case, our lower bound follows from Claim 5.6. In the other case, we directly have our desired lower bound

on $\mathbb{E}\left[T_i(n)\right]$. For further details, refer to the full proof in Appendix D.

## Acknowledgements

We thank Lucas Dixon and Nithum Thain for helpful comments, and Sanjay Shakkottai for insightful discussions on the mathematical aspects of the model and analysis.

## Impact Statement

We introduce a simplified model to mathematically study the feedback loop created by affinity bias in hiring, and how to mitigate it. We prove a lower bound on the regret of *any* algorithm (even one that knows the exact bias model), and provide an algorithm nearly matching this lower bound despite not knowing the bias. Perhaps the biggest insight we gain from this analysis comes from the fact that this almost-optimal algorithm is a variant of the elimination algorithm (in which we pick each group one after the other in a round-robin fashion, until we're confident we have gathered enough information), which shows giving every group a chance is essentially the best policy in this setting.

Since we aim to minimize regret with respect to fixed arm means, underlining assumptions of this work are that (i) some groups perform better than others, and (ii) the underlying qualities of each group do not change with time. In the case where groups are defined by skill-set and we aim to pinpoint the most relevant skill-set for a job, assumption (i) may come at no cost. However, if we define groups based on sensitive attributes (which would allow to mitigate legally-relevant discrimination), assuming that the different groups have different expectations become problematic. This setting could be better modeled, e.g., by a variant of contextual bandits rather than our variant of the traditional bandits. Moreover, if the decision-making process has downstream impacts on the groups, assumption (ii) may also be unrealistic. However, generalizing our model to accommodate both time-varying feedback and underlying rewards appears to be a challenging direction. We hope the proof techniques here pave the way towards these setting, which we leave for future work.

Finally, another assumption of the simplified model is that we always consider the whole history of the algorithm in our bias model, which would correspond to an ever-growing hiring committee. In practice, the number of people in a hiring committee is bounded. One could address this limitation by allowing the algorithm to depend only on the last $M$ hires. This is also left for future work.

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

# A. Extended related works

**Fairness and sequential decision making** The study of fairness in the context of multi-armed bandit problems was first studied in (Joseph et al., 2016). Since then, a number of works, including (Liu et al., 2017; Gillen et al., 2018; Khalili et al., 2021; Wang et al., 2021), have considered a variety of fairness notions for bandits and sequential decision making problems. In these settings, the goal is typically to simultaneously minimize regret measured with respect to the observed rewards, while satisfying some notion of fairness. By contrast, in our setting, the decision maker receives biased feedback, and the goal is to minimize regret relative to the unbiased reward distribution. There is otherwise no reward or penalty for fairness (or lack thereof), diversity in action selection, etc., beyond the goal of minimizing regret.

**Bandits with biased feedback** A number of recent works have studied bandit models where the observed feedback is biased. (Tang & Ho, 2019) considers a Bernoulli bandit problem where the observed rewards for arm $i$ are biased as a function of (i) the number of times arm $i$ has been played and (ii) the empirical average of past rewards from arm $i$. (Gaucher et al., 2022; Schumann et al., 2022) considers the problem of linear bandits where the observed feedback is biased by a linear function which depends on the current action and reward of the selected arm. None of these models, however, can capture the setting which we consider, where (a) the arm mean depends on the *fraction* of times an arm is played, and (b) playing one arm decreases the biased means of every other arm.

**Non-stationary bandits** Multi-armed bandit problems with time-varying rewards have a long history. One of the earliest models is the so-called rested bandit (Gittins, 1979), where the reward distribution of an arm changes (in some structured way) when it is pulled. A related setting is the restless bandits problem, introduced in (Whittle, 1988), where the reward distributions change with time, independently of the chosen arm. The rotting bandits and rising bandits problems (Heidari et al., 2016; Levine et al., 2017; Li et al., 2020) consider settings where the means of an arm's reward is decreasing or increasing (respectively) as a function of the number of times it is played. The tallying bandit problem (Malik et al., 2022; 2023) is a generalization of the rested, rotting, and increasing bandits settings which allows the mean of an arm's reward to vary as a function of the number of times an arm was played over the last $m$ time-steps. The recharging bandits problem (Kleinberg & Immorlica, 2018) is a setting where the means vary as an increasing concave function of the time since they were last played.

The measure of regret in each of these settings is with respect to the *observed* rewards with potentially changing distributions. By contrast, in our model, the observed rewards are non-stationary, but the distributions of unbiased rewards (against which we measure our regret) do not change with time.

**Partial Monitoring** Partial monitoring is a general sequential decision-making setting introduced by (Rustichini, 1999) which encompasses both bandit and full-information problems, and allows the feedback observed by the learner after playing an action to be different than the reward associated with that action. In the standard $K$-arm, $m$-outcome setting, there is a loss matrix $L \in \mathbb{R}^{K \times m}$ and feedback matrix $\Phi \in \Sigma^{K \times m}$, where $\Sigma$ is the set of $m$ outcomes. At each round, the learner selects an arm $A_t \in [K]$, simultaneously the environment selects an outcome $i_t$, then the learner suffers (*but does not observe*) loss $L_{A_t i_t}$, and observes feedback $\Phi_{A_t i_t}$. The goal is to minimize regret *with respect to the true losses*, not the observed feedback. While our setting can be modelled as an adversarial partial monitoring problem (where the number of outcomes scales with the number of possible bias configurations for each arm), the regret guarantees do not transfer meaningfully. Indeed, the regret classification theorem (Bartók et al., 2014; Lattimore & Szepesvári, 2019) implies linear regret in the worst case (since the guarantees assume adversarial noise). Moreover, guarantees for stochastic partial monitoring (e.g., (Bartók et al., 2011)) are not applicable, as the feedback distributions are not i.i.d. in our model.

**Techniques for bandit lower bounds** Our lower bound techniques build upon a long line of works which characterized the fundamental limits of bandit problems. Asymptotic instance-dependent bandit lower bounds were first given in (Lai & Robbins, 1985), and later generalized in (Burnetas & Katehakis, 1996). (Garivier et al., 2019) gave a simple yet powerful framework for obtaining lower bounds by combining the standard "divergence decomposition" for KL divergences with a data-processing inequality. (Kaufmann et al., 2016) exploited the fact that the divergence decomposition holds also until any finite stopping time to obtain lower bounds for the Best-Arm Identification problem. Our work generalizes this framework to handle the challenging setting where observed feedback is time-varying and dependent on the decisions of a policy which potentially knows the bias model exactly.

---

**Algorithm 2** The Elimination-style algorithm for unknown bias model, with added notations

---

**Require:** Time horizon $n \in \mathbb{N}$, sampling schedule $m_r$.

  Pull each arm $i \in [K]$ once (in arbitrary order) and discard the sample {Ensure $T_i^0 \geq 1$ for all $i \in [K]$}

  Let $\tau_0 = 0, t = 1$, and $\mathcal{A}_1 = [K]$

  **for** $r = 1, 2, \ldots$ **do**

    Set $b_i(r) = \sum_{j \in \mathcal{A}_r} \mathbb{1}\{j < i\}$ {The number of arms played before $i$ at each iteration $\ell$ during round $r$.}

    **for** $\ell \in [|m_r|], i \in \mathcal{A}_r$ in increasing order of index **do**

      Pull arm $i$ and receive feedback $Y_t$ (we will sometimes refer to this sample as $Y_{i,r,\ell}$).

      Set $\text{time}_i(r, \ell) = t$ and update $t \leftarrow t + 1$

    **end for**

    Update:

$$\widehat{\mu}_i(r) = \frac{\sum_{\ell \in [|m_r|]} Y_{i,r,\ell}}{m_r} \quad \text{and} \quad \mathcal{A}_{r+1} = \left\{ i \in \mathcal{A}_r : \max_{j \in \mathcal{A}_r} \widehat{\mu}_j(r) - \widehat{\mu}_i(r) \leq 2^{-r} \right\} \quad \text{and} \quad \tau_r = t - 1.$$

  **end for**

---

## B. Relationship to partial monitoring literature

Here, we expand upon the comment in Section 1 regarding partial monitoring algorithms suffering linear regret in our setting.

Let us consider a 2-armed Bernoulli bandit instance with means $1 > \mu_1 > \mu_2 > 0$, under a bias model $W_i(t)$. At each round, $Y_t$ is generated as follows: For each arm $i \in \{1, 2\}$ at time $t$, let $X_{i,t} \sim \text{Bernoulli}(\mu_i)$, let $F_{i,t} \sim \text{Bernoulli}(W_i(t))$, let $Y_{i,t} = X_{i,t} F_{i,t}$, and take $Y_t = Y_{A_t,t}$. Clearly, this construction satisfies Assumption 2.1.

We can model this setting as a partial monitoring problem as follows: let $\mathcal{L}, \Phi \in \mathbb{R}^{K \times m}$, where $K = 2$ is the number of arms and $m = 16$ is the number of outcomes (representing the $2^4$ possible configurations of $(X_{1,t}, X_{2,t}, F_{1,t}, F_{2,t})$ for each $t$). Then, the loss and feedback matrices at each row $i \in [K]$ and each column $(X_{1,t}, X_{2,t}, F_{1,t}, F_{2,t}) \in \{0, 1\}^4$ is:

$$\mathcal{L}[i, (X_{1,t}, X_{2,t}, F_{1,t}, F_{2,t})] = 1 - X_{i,t}$$
$$\Phi[i, (X_{1,t}, X_{2,t}, F_{1,t}, F_{2,t})] = 1 - X_{i,t} F_{i,t}$$

An adversary could simulate our setting by sampling $X_{i,t} \sim \text{Bernoulli}(\mu_i)$ and $F_{i,t} \sim \text{Bernoulli}(W_i(t))$, then selecting the outcome $O(t) = (X_{1,t}, X_{2,t}, F_{1,t}, F_{2,t})$.

However, it is straightforward to observe that, when the adversary is allowed to choose the outcomes arbitrarily, then linear regret is inevitable. Indeed, consider the outcomes $O_1 = (0, 1, 0, 0)$ and $O_2 = (1, 0, 0, 0)$. Notice that $\mathcal{L}[i, O_1] = \mathbb{1}\{i = 1\}$ and $\mathcal{L}[i, O_2] = \mathbb{1}\{i = 2\}$, while $\Phi[i, O_1] = 0 = \Phi[i, O_2]$. Consider two environments: in the first, the adversary chooses $O_1$ for each time $t \in [n]$ (hence, action 2 is optimal); in the second, the adversary chooses $O_2$ for each time $t \in [n]$ (hence, action 1 is optimal). However, since the feedback is deterministically 0 at every time-step, (the distribution of) any policy is the same in both environments. Thus, any policy must suffer regret at least $n/2$ in one of these two environments.

## C. A phased elimination-style algorithm for unknown bias model

Here, we analyze the regret of Algorithm 2. Before stating the bound, let us first introduce a useful decomposition of $\frac{T_{A_t}^{\text{bias}}(t-1)}{t^{\text{bias}} - 1}$ for Algorithm 2:

**Lemma C.1.** *In the context of Algorithm 2, let $t = \text{time}_i(r, \ell)$ be the time when the algorithm plays an active arm $i \in \mathcal{A}_r$ for the $\ell$th time in round $r$. Then,*

$$\frac{T_i^{\text{bias}}(t-1)}{t^{\text{bias}} - 1} = \frac{T_i^0 + \sum_{r'=1}^{r-1} m_{r'} + (\ell - 1)}{t_0^{\text{bias}} + \sum_{r'=1}^{r-1} |\mathcal{A}_{r'}| m_{r'} + |\mathcal{A}_r|(\ell - 1) + b_i(r)},$$

*where $m_r$ is the number of times each active arm is played in round $r$, and $b_i(r) = \sum_{j \in \mathcal{A}_r} \mathbb{1}\{j < i\}$ is the number of arms played before $i$ in each iteration $\ell$ of round $r$.*

*Proof.* This decomposition follows immediately from the definition of Algorithm 2, noting that (i) at each round $r'$, each arm is played $m_{r'}$ times (for a total of $|\mathcal{A}_{r'}|m_{r'}$ time steps), (ii) before arm $i$ is played for the $\ell$th time in round $r$, every arm has been played for $(\ell-1)|\mathcal{A}_r|$ times, and (iii) additionally the arms in $\mathcal{A}_r$ which have smaller index than $i$ ($b_i(r)$ in total) have been played one additional time. Similarly, since $\mathcal{A}_{r+1} \subseteq \mathcal{A}_r$, an active arm at the $\ell$th iteration of round $r$ was played $\sum_{r'=1}^{r-1} m_{r'}$ times in the previous rounds (corresponding to the plays in rounds 1 to $r-1$), plus $\ell-1$ times in round $r$ before the current iteration. $\square$

The regret guarantee for Algorithm 2 will also make use of the following notation:

**Definition C.2** (Active arm upper confidence sets). Let us define the following sets $U_r$:

$$U_1 = [K] \quad \text{and} \quad U_{r+1} = \left\{ i \in U_r : \Delta_i \bar{f}_i^{\min}(r) \leq 2^{-r+1} \right\},$$

where

$$\bar{f}_i^{\min}(r) = \bar{f}_i(r \mid U_1, \ldots, U_r),$$

and $\bar{f}_i(r \mid \mathcal{A}_1, \ldots, \mathcal{A}_r)$ is the average reweighting of arm $i$ during round $r$, i.e.,

$$\bar{f}_i(r \mid \mathcal{A}_1, \ldots, \mathcal{A}_r) = \frac{1}{m_r} \sum_{t=\text{time}_i(r,1)}^{\text{time}_i(r,m_r)} f\left( \frac{T_i^{\text{bias}}(t-1)}{t^{\text{bias}}-1} \right) \mathbb{1}\{A_t = i\} \tag{8}$$

$$= \frac{1}{m_r} \sum_{\ell \in [m_r]} f\left( \frac{T_i^0 + \sum_{r'=1}^{r-1} m_{r'} + (\ell-1)}{t_0^{\text{bias}} + \sum_{r'=1}^{r-1} |\mathcal{A}_{r'}|m_{r'} + |\mathcal{A}_r|(\ell-1) + b_i(\mathcal{A}_r)} \right). \tag{9}$$

Further, let us denote $u_i$ as the last round when $i$ is in $U_r$, i.e.,

$$u_i = \min\{r \geq 1 : i \notin U_{r+1}\}.$$

Before stating our main algorithmic guarantee, we establish some important properties of Definition C.2:

**Lemma C.3.** *For every $r \geq 1$, the sets $U_r$ and associated functions $\bar{f}_i^{\min}(r)$ from Definition C.2 satisfy the following: Let $i_* = \arg\max \mu_i$ be the index of an optimal arm. Then, for any arm $i \in U_r$, under Assumption 2.1,*

$$i_* \in U_r \quad \text{and} \quad \bar{f}_i^{\min}(r) \leq \bar{f}_i(r \mid \mathcal{A}_1, \ldots, \mathcal{A}_r) \quad \forall i \in [K], \mathcal{A}_\ell \subseteq U_\ell \; \forall \ell \in [r].$$

Before proving Lemma C.3, we first briefly give some intuition for Definition C.2 in light of this result. Recall that Algorithm 1 maintains a set of "active" arms $\mathcal{A}_r$ during each round $r$. At the end of each round $r$, the algorithm eliminates arms whose empirically averaged feedback is sufficiently smaller than the largest observed feedback. More specifically, it eliminates all arms $i$ such that:

$$\max_{j \in \mathcal{A}_r} \widehat{\mu}_j(r) - \widehat{\mu}_i(r) > 2^{-r},$$

By definition, the expected feedback averaged over round $r$ (and conditioned on the observations from previous rounds) for an active arm $i \in \mathcal{A}_r$ is:

$$\widetilde{\mu}_i(r) = \mu_i \bar{f}_i(r \mid \mathcal{A}_1, \ldots, \mathcal{A}_r).$$

Therefore, the expected gap between an optimal arm $i_*$'s feedback and any other active arm $i$'s feedback averaged over round $r$ is given by:

$$\widetilde{\mu}_{i_*}(r) - \widetilde{\mu}_i(r) = \mu_{i_*} \bar{f}_{i_*}(r \mid \mathcal{A}_1, \ldots, \mathcal{A}_r) - \mu_i \bar{f}_i(r \mid \mathcal{A}_1, \ldots, \mathcal{A}_r)$$
$$= \Delta_i \bar{f}_i(r \mid \mathcal{A}_1, \ldots, \mathcal{A}_r) + \mu_{i_*}(\bar{f}_i(r \mid \mathcal{A}_1, \ldots, \mathcal{A}_r) - \bar{f}_{i_*}(r \mid \mathcal{A}_1, \ldots, \mathcal{A}_r)).$$

In Lemma C.7, we show that, for sufficiently large time horizons $T$ and by the choice of sampling schedule $m_r$, the second term above is negligible (i.e., sufficiently smaller than $2^{-r}$). Hence, the gap above is dominated by the first term, $\Delta_i \bar{f}_i(r \mid \mathcal{A}_1, \ldots, \mathcal{A}_r)$. Now, we show in Lemma C.8 that, with high probability, $\mathcal{A}_r \subseteq U_r$ and $i_* \in \mathcal{A}_r$ for all rounds $r$. Hence, with high probability, by Lemma C.3 it holds that $\bar{f}_i(r \mid \mathcal{A}_1, \ldots, \mathcal{A}_r) \geq \bar{f}_i^{\min}(r)$ for every active arm $i \in \mathcal{A}_r$, and the suboptimal arm is not eliminated. Thus, we can interpret $\Delta_i \bar{f}_i^{\min}(r)$ as (essentially) a high probability lower bound on the "reweighted" suboptimality gap $\widetilde{\mu}_{i_*}(r) - \widetilde{\mu}_i(r)$. Thus, the sets $U_{r+1}$ mimic the definition of the active sets $\mathcal{A}_{r+1}$ in Algorithm 1, replacing the empirical gap $\max_{j \in \mathcal{A}_r} \widehat{\mu}_j(r) - \widehat{\mu}_i(r)$ with $\Delta_i \bar{f}_i^{\min}(r) \lesssim \max_{j \in U_r} \widetilde{\mu}_j(r) - \widetilde{\mu}_i(r)$.

*Proof of Lemma C.3.* We first notice that, since $\Delta_{i_*} = 0$ for any optimal arm $i_*$, trivially we have that $\Delta_{i_*} \bar{f}_{i_*}^{\min}(r) = 0 \leq 2^{-r+1}$ for every $r$. Hence, by definition of the upper confidence sets from Definition C.2, $i_* \in U_r$ for all $r$.

To establish the remaining claim, we notice that, for any round $r$ such that $i \in \mathcal{A}_r$ (hence also $i \in U_r$ since $\mathcal{A}_r \subseteq U_r$):

$$
\begin{aligned}
\bar{f}_i^{\min}(r) &= \bar{f}_i(r \mid U_1, \ldots U_r) \\
&= \frac{1}{m_r} \sum_{\ell=1}^{m_r} f \left( \frac{T_i^0 + \sum_{r'=1}^{r-1} m_{r'} + (\ell-1)}{t_0^{\text{bias}} + \sum_{r'=1}^{r-1} |U_{r'}| m_{r'} + |U_r|(\ell-1) + b_i(U_r)} \right) \\
&= \frac{1}{m_r} \sum_{\ell=1}^{m_r} f \left( \frac{T_i^0 + \sum_{r'=1}^{r-1} m_{r'} + (\ell-1)}{t_0^{\text{bias}} + \sum_{r'=1}^{r-1} |U_{r'}| m_{r'} + |U_r|(\ell-1) + \sum_{j \in U_r} \mathbb{1}\{j < i\}} \right) \\
&\leq \frac{1}{m_r} \sum_{\ell=1}^{m_r} f \left( \frac{T_i^0 + \sum_{r'=1}^{r-1} m_{r'} + (\ell-1)}{t_0^{\text{bias}} + \sum_{r'=1}^{r-1} |\mathcal{A}_{r'}| m_{r'} + |\mathcal{A}_r|(\ell-1) + \sum_{j \in \mathcal{A}_r} \mathbb{1}\{j < i\}} \right) \\
&= \frac{1}{m_r} \sum_{\ell=1}^{m_r} f \left( \frac{T_i^0 + \sum_{r'=1}^{r-1} m_{r'} + (\ell-1)}{t_0^{\text{bias}} + \sum_{r'=1}^{r-1} |\mathcal{A}_{r'}| m_{r'} + |\mathcal{A}_r|(\ell-1) + b_i(\mathcal{A}_r)} \right) \\
&= \bar{f}_i(r \mid \mathcal{A}_1, \ldots, \mathcal{A}_r),
\end{aligned}
$$

where in the inequality above, we used the facts that (i) $\mathcal{A}_{r'} \subseteq U_{r'}$ for every $r' \in [r]$ by definition, and (ii) the function $f(\cdot)$ is non-decreasing by Assumption 2.1. Notice that this inequality becomes an equality in the case that $r = 1$, since $\mathcal{A}_1 = U_1$. $\square$

**Theorem C.4** (Generalized version of regret guarantee from Theorem 4.1). *Suppose that Algorithm 2 is run using the sampling schedule:*

$$
m_r = 2^{2r+5} \log \left( \frac{12}{\pi^2} K^2 r^2 n \right), \tag{10}
$$

*for a time horizon $n$ sufficiently large such that:*

$$
\log(nK) \geq \frac{\mu_1 L}{2^5} \left( 1 + \max \left\{ \left( 1 + \frac{T_{max}^0 - T_{min}^0}{K} \right) \log \left( 1 + 2^8 \log \left( \frac{12}{\pi^2} K^2 n \right) \right), \left( 1 + T_{max}^0 - T_{min}^0 \right) \log(13) \right\} \right). \tag{11}
$$

*Then, assuming the environment satisfies (2) and Assumption 2.1 with $\mu_i \in [0,1]$ for all $i \in [0,1]$, the regret of Algorithm 2 is bounded as:*

$$
R_{\boldsymbol{\nu}, \pi}(n) = \sum_{i:\Delta_i > 0} \Delta_i \mathbb{E}[T_i(n)] \leq \sum_{i:\Delta_i > 0} \Delta_i + \frac{2^{11}}{3} \log \left( \frac{12}{\pi^2} K^2 n^3 \right) \frac{1}{\Delta_i \bar{f}_i^{\min}(u_i - 1)^2}. \tag{12}
$$

*Further, we also have the bound:*

$$
R_{\boldsymbol{\nu}, \pi}(n) \leq K + 2 \sqrt{\frac{2^{11} nK \log \left( \frac{12}{\pi^2} K^2 n^3 \right)}{3 f \left( \frac{T_{min}^0}{t_0^{\text{bias}} + K - 1} \right)^2}}. \tag{13}
$$

Let us briefly comment on how to interpret the regret guarantee of Theorem C.4. Recall from Lemma C.3 that the average multiplicative reweighting of arm $i$ during round $r$ can be written as:

$$
\bar{f}_i(r \mid \mathcal{A}_1, \ldots, \mathcal{A}_r) = \frac{1}{m_r} \sum_{\ell=1}^{m_r} f \left( \frac{T_i^0 + \sum_{r'=1}^{r-1} m_{r'} + (\ell-1)}{t_0^{\text{bias}} + \sum_{r'=1}^{r-1} |\mathcal{A}_{r'}| m_{r'} + |\mathcal{A}_r|(\ell-1) + b_i(r)} \right),
$$

where $b_i(r) = \sum_{j \in \mathcal{A}_r} \mathbb{1}\{j < i\}$ is the number of arms played before $i$ during each iteration of round $r$. Now, one can show that $L_{r'} \subseteq \mathcal{A}_{r'} \subseteq U_{r'}$ holds for all $r' \leq n$ with high probability (see the proof of Lemma C.8 for details). Hence, with high probability, $\bar{f}_{\min}(r) \leq \bar{f}_i(r \mid \mathcal{A}_1, \ldots, \mathcal{A}_r)$ for every $i \in \mathcal{A}_r$.

Now, by definition, $u_i$ is the smallest $r$ such that $i \notin U_{r+1}$. Since $\mathcal{A}_{r'} \subset U_{r'}$ for all $r'$ with high probability, it follows that $i \notin \mathcal{A}_{u_i+1}$ with high probability. Also, notice that $i \in S_{u_i}$, so $\Delta_i \bar{f}_{\min}(u_i - 1) \leq 2^{-(u_i-1)+1}$, or equivalently, $2^{u_i} \leq \frac{4}{\bar{f}_{\min}(u_i-1)\Delta_i}$. Recalling our choice of sampling schedule $m_r$, this impies that, with high probability, each suboptimal arm will be played no more than $\sum_{r' \leq u_i} m_{r'} \lesssim m_{u_i} \lesssim \frac{\log(n)}{\bar{f}_i^{\min}(u_i-1)^2\Delta_i^2}$ times.

Because of the above observations, we can interpret $\bar{f}_i^{\min}(u_i - 1)$ to be a high-probability lower bound on the average reweighting on arm $i$ over round $u_i - 1$, i.e., the round before $i$ is eliminated (with high probability) by Algorithm 1.

Before proving Theorem C.4, we interpret this result by giving bounds on $\bar{f}_i^{\min}(u_i - 1)$.

**Lemma C.5** (Typical scaling). *Recall the function $\bar{f}_i^{\min}(r)$ from Definition C.2 defined for each round $r \geq 1$ and $i \in U_r$. For any round $r \geq 1$,*

$$\bar{f}_i^{\min}(r) \geq f\left(\min\left\{\frac{T_i^0}{t_0^{\text{bias}} + i - 1}, \frac{1}{K}\right\}\right) \geq f\left(\frac{T_{\min}^0}{t_0^{\text{bias}} + K - 1}\right).$$

*Further, if $T_{\max}^0 \leq 2^7 \log\left(12K^2 n/\pi^2\right)$ and $n \geq r \geq 2$, then*

$$\bar{f}_i^{\min}(r) \geq f\left(\frac{1}{15K}\right).$$

*Proof.* Begin by recalling, by Definition C.2,

$$\bar{f}_i^{\min}(r) = \bar{f}_i(r \mid U_1, \ldots, U_r) = \frac{1}{m_r}\sum_{\ell \in [m_r]} f\left(\frac{T_i^0 + \sum_{r'=1}^{r-1} m_{r'} + (\ell - 1)}{t_0^{\text{bias}} + \sum_{r'=1}^{r-1}|U_{r'}|m_{r'} + |U_r|(\ell-1) + b_i(U_r)}\right).$$

Recall that, by Assumption 2.1, $f(\cdot)$ is nondecreasing. Therefore, to establish our claims, it suffices to lower bound, for each $\ell \in [m_r]$, the fraction of times arm $i$ is played at the $\ell$th iteration of round $r$:

$$\frac{T_i^0 + \sum_{r'=1}^{r-1} m_{r'} + (\ell - 1)}{t_0^{\text{bias}} + \sum_{r'=1}^{r-1}|U_{r'}|m_{r'} + |U_r|(\ell-1) + b_i(U_r)}.$$

Now, we can decompose the fraction of times an arm $i$ is played as:

$$\frac{T_i^0 + \sum_{r'=1}^{r-1} m_{r'} + (\ell - 1)}{t_0^{\text{bias}} + \sum_{r'=1}^{r-1}|U_{r'}|m_{r'} + |U_r|(\ell-1) + b_i(U_r)} = \frac{T_i^0}{t_0^{\text{bias}} + b_i(U_r)}\frac{t_0^{\text{bias}} + b_i(U_r)}{t_0^{\text{bias}} + \sum_{r'=1}^{r-1}|U_{r'}|m_{r'} + |U_r|(\ell-1) + b_i(U_r)} \quad (14)$$

$$+ \sum_{r'=1}^{r-1} \frac{1}{|U_{r'}|}\frac{|U_{r'}|m_{r'}}{t_0^{\text{bias}} + \sum_{r'=1}^{r-1}|U_{r'}|m_{r'} + |U_r|(\ell-1) + b_i(U_r)} \quad (15)$$

$$+ \frac{\ell - 1}{|U_r|(\ell-1)}\frac{|U_r|(\ell-1)}{t_0^{\text{bias}} + \sum_{r'=1}^{r-1}|U_{r'}|m_{r'} + |U_r|(\ell-1) + b_i(U_r)}. \quad (16)$$

Using the fact that $U_{r+1} \subseteq U_r$ for all $r$ and $U_1 = [K]$, we thus conclude that:

$$\frac{T_i^0 + \sum_{r'=1}^{r-1} m_{r'} + (\ell - 1)}{t_0^{\text{bias}} + \sum_{r'=1}^{r-1}|U_{r'}|m_{r'} + |U_r|(\ell-1) + b_i(U_r)} \geq \min\left\{\frac{T_i^0}{t_0^{\text{bias}} + b_i(U_1)}, \frac{1}{|U_1|}\right\} = \min\left\{\frac{T_i^0}{t_0^{\text{bias}} + i - 1}, \frac{1}{K}\right\} \geq \frac{T_{\min}^0}{t_0^{\text{bias}} + K - 1}.$$

This establishes the first claim.

We now focus on establishing the refined claim for rounds $n \geq r \geq 2$. Using the decomposition from (14), it follows that

$$\frac{T_i^0 + \sum_{r'=1}^{r-1} m_{r'} + (\ell - 1)}{t_0^{\text{bias}} + \sum_{r'=1}^{r-1}|U_{r'}|m_{r'} + |U_r|(\ell-1) + b_i(U_r)} \geq \sum_{r'=1}^{r-1} \frac{1}{|U_{r'}|}\frac{|U_{r'}|m_{r'}}{t_0^{\text{bias}} + \sum_{r'=1}^{r-1}|U_{r'}|m_{r'} + |U_r|(\ell-1) + b_i(U_r)}.$$

Since $r \geq 2$ and $K \geq |U_{r'}| \geq |U_r|$ for $r' \leq r$, we can further bound the RHS above as:

$$\sum_{r'=1}^{r-1} \frac{1}{|U_{r'}|} \frac{|U_{r'}|m_{r'}}{t_0^{\text{bias}} + \sum_{r'=1}^{r-1}|U_{r'}|m_{r'} + |U_r|(\ell-1) + b_i(U_r)} \geq \frac{1}{K} \frac{\sum_{r'=1}^{r-1}|U_{r'}|m_{r'}}{t_0^{\text{bias}} + \sum_{r'=1}^{r-1}|U_{r'}|m_{r'} + |U_r|(\ell-1) + b_i(U_r)}$$

$$= \frac{1}{K} \frac{1}{\frac{t_0^{\text{bias}}+b_i(U_r)}{\sum_{r'=1}^{r-1}|U_{r'}|m_{r'}} + 1 + \frac{|U_r|(\ell-1)}{\sum_{r'=1}^{r-1}|U_{r'}|m_{r'}}}$$

$$\geq \frac{1}{K} \frac{1}{\frac{t_0^{\text{bias}}+K-1}{\sum_{r'=1}^{r-1}|U_{r'}|m_{r'}} + 1 + \frac{|U_r|(\ell-1)}{\sum_{r'=1}^{r-1}|U_{r'}|m_{r'}}}$$

$$\geq \frac{1}{K} \frac{1}{\frac{t_0^{\text{bias}}+K-1}{Km_1} + 1 + \frac{m_r}{\sum_{r'=1}^{r-1}m_{r'}}},$$

Recalling our choice of $m_r$,

$$\sum_{r'=1}^{r-1} m_{r'} = \sum_{r'=1}^{r-1} 2^{2r+5} \log\left(\frac{12}{\pi^2}K^2(r')^2 n\right)$$

$$\geq 2^5 \log\left(\frac{12}{\pi^2}K^2 n\right) \sum_{r'=1}^{r-1} 2^{2r}$$

$$= 2^7 \log\left(\frac{12}{\pi^2}K^2 n\right) \frac{2^{2(r-1)}-1}{3},$$

so, since $r \leq n$,

$$\frac{m_r}{\sum_{r'=1}^{r-1}m_{r'}} \leq \frac{2^{2r+5}\log\left(\frac{12}{\pi^2}K^2 r^2 n\right)}{2^7 \log\left(\frac{12}{\pi^2}K^2 n\right)\frac{2^{2(r-1)}-1}{3}}$$

$$\leq \frac{\log\left(\frac{12}{\pi^2}K^2 n^3\right)}{\log\left(\frac{12}{\pi^2}K^2 n\right)} \frac{2^{2r+5}}{2^7 \frac{2^{2(r-1)}-1}{3}}$$

$$\leq \frac{\log\left(\left(\frac{12}{\pi^2}K^2 n\right)^3\right)}{\log\left(\frac{12}{\pi^2}K^2 n\right)} \frac{2^{2r+5}}{2^7 \frac{2^{2(r-1)}-1}{3}}$$

$$= 9\frac{2^{2r}}{2^{2r}-4}$$

$$= \frac{9}{1-2^{-2(r-1)}}$$

$$\leq \frac{9}{1-2^{-2(2-1)}}$$

$$= 12.$$

Collecting our results, we thus have:

$$\sum_{r'=1}^{r-1} \frac{1}{|U_{r'}|} \frac{|U_{r'}|m_{r'}}{t_0^{\text{bias}} + \sum_{r'=1}^{r-1}|U_{r'}|m_{r'} + |U_r|(\ell-1) + b_i(U_r)} \geq \frac{1}{K} \frac{1}{\frac{t_0^{\text{bias}}+K-1}{Km_1} + 1 + \frac{m_r}{\sum_{r'=1}^{r-1}m_{r'}}}$$

$$\geq \frac{1}{K} \frac{1}{\frac{t_0^{\text{bias}}+K-1}{Km_1} + 13}$$

$$\geq \frac{1}{K} \frac{1}{\frac{t_0^{\text{bias}}}{Km_1} + 14},$$

where in the last line, we used the fact that $m_1 \geq 1$. Therefore, if $m_1 \geq T_{\max}^0$, i.e.,

$$T_{\max}^0 \leq 2^7 \log\left(\frac{12}{\pi^2} K^2 n\right) = m_1,$$

then since $t_0^{\text{bias}}/K \leq T_{\max}^0$, we conclude that:

$$
\begin{aligned}
\frac{T_i^0 + \sum_{r'=1}^{r-1} m_{r'} + (\ell-1)}{t_0^{\text{bias}} + \sum_{r'=1}^{r-1}|U_{r'}|m_{r'} + |U_r|(\ell-1) + b_i(U_r)} &\geq \sum_{r'=1}^{r-1} \frac{1}{|U_{r'}|} \frac{|U_{r'}|m_{r'}}{t_0^{\text{bias}} + \sum_{r'=1}^{r-1}|U_{r'}|m_{r'} + |U_r|(\ell-1) + b_i(U_r)} \\
&\geq \frac{1}{K} \frac{1}{\frac{t_0^{\text{bias}}}{Km_1} + 14} \\
&\geq \frac{1}{K} \frac{1}{\frac{T_{\max}^0}{m_1} + 14} \\
&\geq \frac{1}{15K}.
\end{aligned}
$$

This establishes the second claim. $\qquad\square$

**Corollary C.6** (Simplified regret upper bounds)**.** *In the same setting as Theorem C.4, we have the following regret bound for Algorithm 1:*

$$
\begin{aligned}
R_{\boldsymbol{\nu},\pi}(n) &\leq \sum_{i:\Delta_i>0} \Delta_i + \frac{2^{11}}{3} \log\left(\frac{12}{\pi^2} K^2 n^3\right) \frac{1}{\Delta_i f\left(\min\left\{\frac{T_i^0}{t_0^{\text{bias}}+i-1}, \frac{1}{K}\right\}\right)^2} \\
&\leq \sum_{i:\Delta_i>0} \Delta_i + \frac{2^{11}}{3} \log\left(\frac{12}{\pi^2} K^2 n^3\right) \frac{1}{\Delta_i f\left(\frac{T_{\min}^0}{t_0^{\text{bias}}+K-1}\right)^2}.
\end{aligned}
$$

*Further, assuming $T_{\max}^0 \leq 2^7 \log\left(12K^2 n/\pi^2\right)$, then*

$$R_{\boldsymbol{\nu},\pi}(n) \leq \sum_{i:\Delta_i>0} \Delta_i + \frac{2^{11}}{3} \log\left(\frac{12}{\pi^2} K^2 n^3\right) \frac{1}{\Delta_i f\left(\frac{1}{15K}\right)^2}.$$

*Proof of Corollary C.6.* The first set of regret bounds follows immediately from Theorem C.4 and the first set of inequalities in Lemma C.5.

For the second claim, we consider two cases: (i) $u_i \leq 2$, and (ii) $u_i > 2$. Recalling the regret decomposition from (1), we have that:

$$R_{\boldsymbol{\nu},\pi}(n) = \sum_{i:\Delta_i>0} \Delta_i \mathbb{E}\left[T_i(n)\right] = \sum_{i:\Delta_i>0} \Delta_i \left(\mathbb{E}\left[T_i(n)\mathbb{1}\{i \notin \mathcal{A}_{u_i+1}\}\right] + \mathbb{E}\left[T_i(n)\mathbb{1}\{i \in \mathcal{A}_{u_i+1}\}\right]\right).$$

Using Lemma C.8 and the fact that $T_i(n) \leq n$ by definition,

$$R_{\boldsymbol{\nu},\pi}(n) \leq \sum_{i:\Delta_i>0} \Delta_i \left(\mathbb{E}\left[T_i(n)\mathbb{1}\{i \notin \mathcal{A}_{u_i+1}\}\right] + n\Pr\left[i \in \mathcal{A}_{u_i+1}\right]\right) \leq \sum_{i:\Delta_i>0} \Delta_i + \Delta_i \mathbb{E}\left[T_i(n)\mathbb{1}\{i \notin \mathcal{A}_{u_i+1}\}\right].$$

Since $u_i \leq 2$ by assumption of case (i), and by definition of Algorithm 1:

$$\mathbb{E}\left[T_i(n)\mathbb{1}\{i \notin \mathcal{A}_{u_i+1}\}\right] \leq \sum_{r=1}^{u_i} m_r \leq \sum_{r=1}^{2} m_r.$$

Plugging in our choice of $m_r$ from Theorem C.4,

$$\mathbb{E}\left[T_i(n)\mathbb{1}\{i \notin \mathcal{A}_{u_i+1}\}\right] \leq \sum_{r=1}^{2} 2^{2r+5} \log\left(\frac{12}{\pi^2}K^2 r^2 n\right)$$

$$\leq 2^7 \frac{2^4 - 1}{3} \log\left(\frac{12}{\pi^2}K^2 n^3\right)$$

$$\leq \frac{2^{11}}{3} \log\left(\frac{12}{\pi^2}K^2 n^3\right).$$

Finally, recalling that $\Delta_i \in (0, 1]$ and $f(\cdot) \in (0, 1]$ by assumption, the above implies that

$$\mathbb{E}\left[T_i(n)\mathbb{1}\{i \notin \mathcal{A}_{u_i+1}\}\right] \leq \frac{2^{11}}{3} \log\left(\frac{12}{\pi^2}K^2 n^3\right)\left(\frac{1}{\Delta_i f\left(\frac{1}{15K}\right)}\right)^2.$$

Otherwise, in case (ii) that $u_i > 2$, by the second bound in Lemma C.5 we have that:

$$\bar{f}_i^{\min}(u_i - 1) \geq f\left(\frac{1}{15K}\right).$$

Plugging in this bound to Lemma C.9, we obtain:

$$\mathbb{E}\left[T_i(n)\right] \leq 1 + \frac{2^{11}}{3} \log\left(\frac{12}{\pi^2}K^2 n^3\right)\left(\frac{1}{\Delta_i \bar{f}_i^{\min}(u_i - 1)}\right)^2$$

$$\leq 1 + \frac{2^{11}}{3} \log\left(\frac{12}{\pi^2}K^2 n^3\right)\left(\frac{1}{\Delta_i f\left(\frac{1}{15K}\right)}\right)^2.$$

Collecting these two cases, we conclude that

$$R_{\boldsymbol{\nu},\pi}(n) \leq \sum_{i:\Delta_i > 0} \Delta_i + \frac{2^{11}}{3} \log\left(\frac{12}{\pi^2}K^2 n^3\right) \frac{1}{\Delta_i f\left(\frac{1}{15K}\right)^2},$$

as claimed. $\qquad\square$

We now turn our attention to the proof of Theorem C.4. The first step is a key stability result, showing that the error in suboptimality gap estimates is sufficiently small to prevent Algorithm 2 from mistakenly removing an arm too early.

**Lemma C.7.** *Let $\{Y_t\}_{t \geq 1}$ be feedback satisfying (2) and Assumption 2.1, which is observed by Algorithm 2. Let $\mathcal{A}_r$ be the set of active arms at round $r$, $m_r$ be the number of times each active arm $i \in \mathcal{A}_r$ is pulled during round $r$, and $\tau_r$ be the last time index of the $r$th round. Denote the mean of the samples for an arm $i \in \mathcal{A}_r$ observed by the algorithm during round $r \geq 1$ as:*

$$\widetilde{\mu}_i(r) = \frac{1}{m_r} \sum_{t=\tau_{r-1}+1}^{\tau_r} \mathbb{E}\left[Y_t \mathbb{1}\{A_t = i\} \mid \mathcal{F}_{\tau_{r-1}}\right].$$

*Then, for any active arms $i, j \in \mathcal{A}_r$ in round $r$,*

$$\widetilde{\mu}_i(r) - \widetilde{\mu}_j(r) = \Delta_{i,j}\bar{f}_i(r \mid \mathcal{A}_1, \ldots, \mathcal{A}_r) + \mu_j \xi_{i,j}(r \mid \mathcal{A}_1, \ldots, \mathcal{A}_r)$$
$$= \Delta_{i,j}\bar{f}_j(r \mid \mathcal{A}_1, \ldots, \mathcal{A}_r) + \mu_i \xi_{i,j}(r \mid \mathcal{A}_1, \ldots, \mathcal{A}_r)$$

*where $\bar{f}_i(r \mid \mathcal{A}_1, \ldots, \mathcal{A}_r)$ is as defined in (8), and*

$$\max_{i' \neq j' \in [K]} |\xi_{i',j'}(r \mid \mathcal{A}_1, \ldots, \mathcal{A}_r)| \leq \frac{L}{m_r}\left(1 + \left(1 + \frac{T_{max}^0 - T_{min}^0}{|\mathcal{A}_r|}\right)\log\left(1 + \frac{|\mathcal{A}_r|m_r}{t_0^{\text{bias}} + \sum_{r'=1}^{r-1}|\mathcal{A}_{r'}|m_{r'}}\right)\right)$$

*Proof.* Following the notation of Algorithm 2, take $\tau_r = \tau_{r-1} + |\mathcal{A}_r| m_r$ to be the time at the end of the $r$th round. Consider any time in the $r$th round $t = \tau_{r-1} + |\mathcal{A}_r|(\ell - 1) + b_i(\mathcal{A}_r) + 1$, where $\ell \in [|\mathcal{A}_r|]$ and $b_i(\mathcal{A}_r) = \sum_{j \in \mathcal{A}_r} \mathbb{1}\{j < i\}$ denotes the number of arms played before $i$ in round $\ell$. Then the fraction of times arm $i$ was played before $t$ is:

$$\frac{T_i^{\text{bias}}(t-1)}{t^{\text{bias}} - 1} = \frac{T_i^{\text{bias}}(\tau_{r-1}) + (\ell - 1)}{\tau_{r-1}^{\text{bias}} + |\mathcal{A}_r|(\ell - 1) + b_i(\mathcal{A}_r)} = \frac{T_i^0 + \sum_{r'=1}^{r-1} m_{r'} + (\ell - 1)}{T^0 + \sum_{r'=1}^{r-1}|\mathcal{A}_r| m_{r'} + |\mathcal{A}_r|(\ell - 1) + b_i(\mathcal{A}_r)}$$

Now, the empirical average of the feedback from round $r$ for arm $i$ is:

$$\begin{aligned}
\widetilde{\mu}_i(r) &= \frac{1}{m_r} \sum_{t=\tau_{r-1}+1}^{\tau_r} \mathbb{E}\left[Y_t \mathbb{1}\{A_t = i\} \mid \mathcal{F}_{\tau_{r-1}}\right] \\
&= \frac{1}{m_r} \sum_{\ell=1}^{m_r} \mathbb{E}\left[Y_{i,r,\ell} \mid \mathcal{F}_{\tau_{r-1}}\right] \\
&= \frac{1}{m_r} \sum_{\ell \in [m_r]} \mu_i W_i(\tau_{r-1} + |\mathcal{A}_r|(\ell - 1) + b_i(\mathcal{A}_r) + 1) \\
&= \frac{1}{m_r} \sum_{\ell \in [m_r]} \mu_i f\left(\frac{T_i^{\text{bias}}(\tau_{r-1}) + (\ell - 1)}{\tau_{r-1}^{\text{bias}} + |\mathcal{A}_r|(\ell - 1) + b_i(\mathcal{A}_r)}\right)
\end{aligned}$$

Thus, for any arms $i, j$, and taking $\Delta_{i,j} = \mu_i - \mu_j$,

$$\begin{aligned}
\widetilde{\mu}_i(r) - \widetilde{\mu}_j(r) &= \frac{1}{m_r} \sum_{\ell \in [m_r]} \mu_i f\left(\frac{T_i^{\text{bias}}(\tau_{r-1}) + (\ell - 1)}{\tau_{r-1}^{\text{bias}} + |\mathcal{A}_r|(\ell - 1) + b_i(\mathcal{A}_r)}\right) - \mu_j f\left(\frac{T_j^{\text{bias}}(\tau_{r-1}) + (\ell - 1)}{\tau_{r-1}^{\text{bias}} + |\mathcal{A}_r|(\ell - 1) + b_j(r)}\right) \\
&= \frac{\Delta_{i,j}}{m_r} \sum_{\ell \in [m_r]} f\left(\frac{T_i^{\text{bias}}(\tau_{r-1}) + (\ell - 1)}{\tau_{r-1}^{\text{bias}} + |\mathcal{A}_r|(\ell - 1) + b_i(\mathcal{A}_r)}\right) \\
&\quad + \frac{\mu_j}{m_r} \sum_{\ell \in [m_r]} f\left(\frac{T_i^{\text{bias}}(\tau_{r-1}) + (\ell - 1)}{\tau_{r-1}^{\text{bias}} + |\mathcal{A}_r|(\ell - 1) + b_i(\mathcal{A}_r)}\right) - f\left(\frac{T_j^{\text{bias}}(\tau_{r-1}) + (\ell - 1)}{\tau_{r-1}^{\text{bias}} + |\mathcal{A}_r|(\ell - 1) + b_j(r)}\right) \\
&= \Delta_{i,j} \bar{f}_i\left(r \mid \mathcal{A}_1, \ldots, \mathcal{A}_r\right) + \mu_j \xi_{i,j}\left(r \mid \mathcal{A}_1, \ldots, \mathcal{A}_r\right),
\end{aligned}$$

where

$$\bar{f}_i\left(r \mid \mathcal{A}_1, \ldots, \mathcal{A}_r\right) = \frac{1}{m_r} \sum_{\ell \in [m_r]} f\left(\frac{T_i^{\text{bias}}(\tau_{r-1}) + (\ell - 1)}{\tau_{r-1}^{\text{bias}} + |\mathcal{A}_r|(\ell - 1) + b_i(\mathcal{A}_r)}\right)$$

and

$$\xi_{i,j}\left(r \mid \mathcal{A}_1, \ldots, \mathcal{A}_r\right) = \frac{1}{m_r} \sum_{\ell \in [m_r]} f\left(\frac{T_i^{\text{bias}}(\tau_{r-1}) + (\ell - 1)}{\tau_{r-1}^{\text{bias}} + |\mathcal{A}_r|(\ell - 1) + b_i(\mathcal{A}_r)}\right) - f\left(\frac{T_j^{\text{bias}}(\tau_{r-1}) + (\ell - 1)}{\tau_{r-1}^{\text{bias}} + |\mathcal{A}_r|(\ell - 1) + b_j(r)}\right).$$

Similarly, we also have that:

$$\begin{aligned}
\widetilde{\mu}_i(r) - \widetilde{\mu}_j(r) &= \frac{1}{m_r} \sum_{\ell \in [m_r]} \mu_i f\left(\frac{T_i^{\text{bias}}(\tau_{r-1}) + (\ell - 1)}{\tau_{r-1}^{\text{bias}} + |\mathcal{A}_r|(\ell - 1) + b_i(\mathcal{A}_r)}\right) - \mu_j f\left(\frac{T_j^{\text{bias}}(\tau_{r-1}) + (\ell - 1)}{\tau_{r-1}^{\text{bias}} + |\mathcal{A}_r|(\ell - 1) + b_j(r)}\right) \\
&= \frac{\mu_i}{m_r} \sum_{\ell \in [m_r]} f\left(\frac{T_i^{\text{bias}}(\tau_{r-1}) + (\ell - 1)}{\tau_{r-1}^{\text{bias}} + |\mathcal{A}_r|(\ell - 1) + b_i(\mathcal{A}_r)}\right) - f\left(\frac{T_j^{\text{bias}}(\tau_{r-1}) + (\ell - 1)}{\tau_{r-1}^{\text{bias}} + |\mathcal{A}_r|(\ell - 1) + b_j(r)}\right) \\
&\quad + \frac{\Delta_{i,j}}{m_r} \sum_{\ell \in [m_r]} f\left(\frac{T_j^{\text{bias}}(\tau_{r-1}) + (\ell - 1)}{\tau_{r-1}^{\text{bias}} + |\mathcal{A}_r|(\ell - 1) + b_j(\mathcal{A}_r)}\right) \\
&= \mu_i \xi_{i,j}\left(r \mid \mathcal{A}_1, \ldots, \mathcal{A}_r\right) + \Delta_{i,j} \bar{f}_j\left(r \mid \mathcal{A}_1, \ldots, \mathcal{A}_r\right),
\end{aligned}$$

It suffices, thus, to bound $\xi_{i,j}\left(r \mid \mathcal{A}_1, \ldots, \mathcal{A}_r\right)$. To begin, we use Assumption 2.1 to bound this term as follows:

$$
\begin{aligned}
\left|\xi_{i,j}\left(r \mid \mathcal{A}_1, \ldots, \mathcal{A}_r\right)\right| &\leq \frac{1}{m_r} \sum_{\ell \in [m_r]} \left| f\left(\frac{T_i^{\text{bias}}(\tau_{r-1}) + (\ell-1)}{\tau_{r-1}^{\text{bias}} + |\mathcal{A}_r|(\ell-1) + b_i(\mathcal{A}_r)}\right) - f\left(\frac{T_j^{\text{bias}}(\tau_{r-1}) + (\ell-1)}{\tau_{r-1}^{\text{bias}} + |\mathcal{A}_r|(\ell-1) + b_j(r)}\right) \right| \\
&\leq \frac{L}{m_r} \sum_{\ell \in [m_r]} \left| \frac{T_i^{\text{bias}}(\tau_{r-1}) + (\ell-1)}{\tau_{r-1}^{\text{bias}} + |\mathcal{A}_r|(\ell-1) + b_i(\mathcal{A}_r)} - \frac{T_j^{\text{bias}}(\tau_{r-1}) + (\ell-1)}{\tau_{r-1}^{\text{bias}} + |\mathcal{A}_r|(\ell-1) + b_j(r)} \right|.
\end{aligned}
\tag{17}
$$

Next, we bound each term in (17) as follows:

$$
\begin{aligned}
&\left| \frac{T_i^{\text{bias}}(\tau_{r-1}) + (\ell-1)}{\tau_{r-1}^{\text{bias}} + |\mathcal{A}_r|(\ell-1) + b_i(\mathcal{A}_r)} - \frac{T_j^{\text{bias}}(\tau_{r-1}) + (\ell-1)}{\tau_{r-1}^{\text{bias}} + |\mathcal{A}_r|(\ell-1) + b_j(r)} \right| \\
&= \left| \frac{(T_i^{\text{bias}}(\tau_{r-1}) + (\ell-1))(\tau_{r-1}^{\text{bias}} + |\mathcal{A}_r|(\ell-1) + b_j(r)) - (T_j^{\text{bias}}(\tau_{r-1}) + (\ell-1))(\tau_{r-1}^{\text{bias}} + |\mathcal{A}_r|(\ell-1) + b_i(\mathcal{A}_r))}{(\tau_{r-1}^{\text{bias}} + |\mathcal{A}_r|(\ell-1) + b_i(\mathcal{A}_r))(\tau_{r-1}^{\text{bias}} + |\mathcal{A}_r|(\ell-1) + b_j(r))} \right| \\
&\leq \left| \frac{(T_i^{\text{bias}}(\tau_{r-1}) + (\ell-1))b_j(r) - (T_j^{\text{bias}}(\tau_{r-1}) + (\ell-1))b_i(\mathcal{A}_r)}{(\tau_{r-1}^{\text{bias}} + |\mathcal{A}_r|(\ell-1) + b_i(\mathcal{A}_r))(\tau_{r-1}^{\text{bias}} + |\mathcal{A}_r|(\ell-1) + b_j(r))} \right| \tag{18} \\
&\quad + \left| \frac{(T_i^{\text{bias}}(\tau_{r-1}) - T_j^{\text{bias}}(\tau_{r-1}))(\tau_{r-1}^{\text{bias}} + |\mathcal{A}_r|(\ell-1))}{(\tau_{r-1}^{\text{bias}} + |\mathcal{A}_r|(\ell-1) + b_i(\mathcal{A}_r))(\tau_{r-1}^{\text{bias}} + |\mathcal{A}_r|(\ell-1) + b_j(r))} \right|. \tag{19}
\end{aligned}
$$

For (18), we use the facts that $0 \leq b_i(\mathcal{A}_r) \leq |\mathcal{A}_r|$, $|\mathcal{A}_r| \geq 1$, and $T_i^{\text{bias}}(\tau_{r-1}) \leq \tau_{r-1}^{\text{bias}}$ to further bound (18) as:

$$
\left| \frac{(T_i^{\text{bias}}(\tau_{r-1}) + (\ell-1))b_j(r) - (T_j^{\text{bias}}(\tau_{r-1}) + (\ell-1))b_i(\mathcal{A}_r)}{(\tau_{r-1}^{\text{bias}} + |\mathcal{A}_r|(\ell-1) + b_i(\mathcal{A}_r))(\tau_{r-1}^{\text{bias}} + |\mathcal{A}_r|(\ell-1) + b_j(r))} \right| \leq \frac{|\mathcal{A}_r|}{\tau_{r-1}^{\text{bias}} + |\mathcal{A}_r|(\ell-1)}.
$$

For (19), we use the fact that, by definition of Algorithm 2, all active arms at the end of each round have been pulled the same number of times (modulo their initial biases), i.e., $T_i^{\text{bias}}(\tau_{r-1}) - T_j^{\text{bias}}(\tau_{r-1}) = T_i^0 - T_j^0$. Thus, we further bound (19) as:

$$
\left| \frac{(T_i^{\text{bias}}(\tau_{r-1}) - T_j^{\text{bias}}(\tau_{r-1}))(\tau_{r-1}^{\text{bias}} + |\mathcal{A}_r|(\ell-1))}{(\tau_{r-1}^{\text{bias}} + |\mathcal{A}_r|(\ell-1) + b_i(\mathcal{A}_r))(\tau_{r-1}^{\text{bias}} + |\mathcal{A}_r|(\ell-1) + b_j(r))} \right| \leq \frac{T_{\max}^0 - T_{\min}^0}{\tau_{r-1}^{\text{bias}} + |\mathcal{A}_r|(\ell-1)}.
$$

Plugging in these bounds to (17), we conclude that:

$$
\begin{aligned}
\left|\xi_{i,j}\left(r \mid \mathcal{A}_1, \ldots, \mathcal{A}_r\right)\right| &\leq \frac{L(|\mathcal{A}_r| + (T_{\max}^0 - T_{\min}^0))}{m_r} \sum_{\ell \in [m_r]} \frac{1}{\tau_{r-1}^{\text{bias}} + |\mathcal{A}_r|(\ell-1)} \\
&\leq \frac{L(|\mathcal{A}_r| + (T_{\max}^0 - T_{\min}^0))}{m_r} \left( \frac{1}{\tau_{r-1}^{\text{bias}}} + \int_0^{m_r - 1} \frac{1}{\tau_{r-1}^{\text{bias}} + |\mathcal{A}_r| x} \mathrm{d}x \right) \\
&= \frac{L(|\mathcal{A}_r| + (T_{\max}^0 - T_{\min}^0))}{m_r} \left( \frac{1}{\tau_{r-1}^{\text{bias}}} + \frac{\log\left( \frac{\tau_{r-1}^{\text{bias}} + |\mathcal{A}_r|(m_r - 1)}{\tau_{r-1}^{\text{bias}}} \right)}{|\mathcal{A}_r|} \right) \\
&\leq \frac{L(|\mathcal{A}_r| + (T_{\max}^0 - T_{\min}^0))}{m_r} \left( \frac{1}{\tau_{r-1}^{\text{bias}}} + \frac{\log\left( \frac{\tau_r^{\text{bias}} - 1}{\tau_{r-1}^{\text{bias}}} \right)}{|\mathcal{A}_r|} \right).
\end{aligned}
$$

Using the fact that $\tau_{r-1}^{\text{bias}} \geq t_0^{\text{bias}} = K T_{\min}^0 + \sum_{i \in [K]} T_i^0 - T_{\min}^0 \geq |\mathcal{A}_r| + T_{\max}^0 - T_{\min}^0$, we thus obtain:

$$
\left|\xi_{i,j}\left(r \mid \mathcal{A}_1, \ldots, \mathcal{A}_r\right)\right| \leq \frac{L}{m_r} \left( 1 + \left( 1 + \frac{T_{\max}^0 - T_{\min}^0}{|\mathcal{A}_r|} \right) \log\left( \frac{\tau_r^{\text{bias}} - 1}{\tau_{r-1}^{\text{bias}}} \right) \right),
$$

which, after using the fact that, by definition of Algorithm 2, $\tau_r^{\text{bias}} = t_0^{\text{bias}} + \sum_{r'=1}^r |\mathcal{A}_{r'}| m_{r'}$, we obtain the claimed result. $\qquad \square$

Using Lemma C.7, we are now ready to show that, with high probability, arm $i$ is removed after at most $u_i$ rounds, where $u_i$ is the round defined in Definition C.2.

**Lemma C.8.** *Suppose that Algorithm 2 is run with $m_r$ as in (10). and suppose that $n$ is sufficiently large such that it satisfies (11). Then, Algorithm 2 satsifies the following:*

$$\Pr\left[i \notin \mathcal{A}_{u_i+1}\right] \geq 1 - \frac{1}{n},$$

*where $u_i$ is the round defined in Definition C.2.*

*Proof.* For ease of notation, let us assume without loss of generality that $\mu_i \geq \mu_{i+1}$ for all $i \in [K-1]$ in this proof (hence, arm 1 is optimal). We prove the following stronger claim:

$$\Pr\left[L_* \subseteq \mathcal{A}_r \subseteq U_r \; \forall r \in [u_i] \text{ and } i \notin \mathcal{A}_{u_i+1}\right] \geq 1 - \frac{1}{n} \quad \text{where} \quad L_* = \{i \in [K] : \Delta_i = 0\}$$

To prove the claim, it is equivalent to show that

$$\Pr\left[i \in \mathcal{A}_{u_i+1} \text{ or } \exists r \in [u_i] : L_* \not\subseteq \mathcal{A}_r \text{ or } \mathcal{A}_r \not\subseteq U_r\right] \leq \frac{1}{n}.$$

To begin, first notice that, by definition of $u_i$, $i \notin U_{u_i+1}$, so since $\mathcal{A}_1 = U_1 = [K]$:

$$\Pr\left[i \in \mathcal{A}_{u_i+1} \text{ or } \exists r \in [u_i] : L_* \not\subseteq \mathcal{A}_r \text{ or } \mathcal{A}_r \not\subseteq U_r\right]$$
$$\leq \Pr\left[\exists r \in [u_i+1] : L_* \not\subseteq \mathcal{A}_r \text{ or } \mathcal{A}_r \not\subseteq U_r\right]$$
$$\leq \sum_{r=1}^{u_i} \Pr\left[L_* \subseteq \mathcal{A}_{r'} \subseteq U_{r'} \; \forall r' < r+1, L_* \not\subseteq \mathcal{A}_{r+1} \text{ or } \mathcal{A}_{r+1} \not\subseteq U_{r+1}\right]. \tag{20}$$

We next decompose each term in the summation from (20) as:

$$\Pr\left[L_* \subseteq \mathcal{A}_{r'} \subseteq U_{r'} \; \forall r' < r+1, L_* \not\subseteq \mathcal{A}_{r+1} \text{ or } \mathcal{A}_{r+1} \not\subseteq U_{r+1}\right]$$
$$= \Pr\left[L_* \subseteq \mathcal{A}_{r'} \subseteq U_{r'} \; \forall r' < r+1, \exists i_* \in L_* : i_* \notin \mathcal{A}_{r+1}\right] \tag{21}$$
$$+ \Pr\left[L_* \subseteq \mathcal{A}_{r'} \subseteq U_{r'} \; \forall r' < r+1, \exists j \in U_r \setminus U_{r+1} : j \in \mathcal{A}_{r+1}\right]. \tag{22}$$

To bound (21), first note that, by definition of Algorithm 2:

$$\Pr\left[L_* \subseteq \mathcal{A}_{r'} \subseteq U_{r'} \; \forall r' < r+1, \exists i_* \in L_* : i_* \notin \mathcal{A}_{r+1}\right]$$
$$= \Pr\left[L_* \subseteq \mathcal{A}_{r'} \subseteq U_{r'} \; \forall r' < r+1, \exists i_* \in L_* : \widehat{\mu}_{\max}(r) - \widehat{\mu}_{i_*}(r) > 2^{-r}\right]$$
$$= \Pr\left[L_* \subseteq \mathcal{A}_{r'} \subseteq U_{r'} \; \forall r' < r+1, \exists i_* \in L_*, j \in \mathcal{A}_r : \widehat{\mu}_j(r) - \widehat{\mu}_{i_*}(r) > 2^{-r}\right]$$
$$\leq \sum_{j \in [K]} \sum_{i_* \neq j} \Pr\left[L_* \subseteq \mathcal{A}_{r'} \subseteq U_{r'} \; \forall r' < r+1, i_* \in L_*, j \in \mathcal{A}_r, \widehat{\mu}_j(r) - \widehat{\mu}_{i_*}(r) > 2^{-r}\right]. \tag{23}$$

Then, recall that, by Lemma C.7, and assuming $L_* \subseteq \mathcal{A}_{r'} \subseteq U_{r'}$ for every $r' \leq r$, and since $i_*, j \in \mathcal{A}_r$ and $f(\cdot) \geq 0$ by Assumption 2.1:

$$\widetilde{\mu}_j(r) - \widetilde{\mu}_{i_*}(r)$$
$$= -\Delta_j \bar{f}_j(r \mid \mathcal{A}_1, \ldots, \mathcal{A}_r) - \mu_{i_*} \xi_{j,i_*}(r \mid \mathcal{A}_1, \ldots, \mathcal{A}_r)$$
$$\leq \mu_1 |\xi_{j,i_*}(r \mid \mathcal{A}_1, \ldots, \mathcal{A}_r)|$$
$$\leq \frac{\mu_1 L}{m_r}\left(1 + \left(1 + \frac{T_{\max}^0 - T_{\min}^0}{|\mathcal{A}_r|}\right) \log\left(1 + \frac{|\mathcal{A}_r| m_r}{T^0 + \sum_{r'=1}^{r-1} |\mathcal{A}_{r'}| m_{r'}}\right)\right)$$
$$\leq \begin{cases} \frac{\mu_1 L}{2^7 \log\left(\frac{12}{\pi^2} K^2 n\right)}\left(1 + \left(1 + \frac{T_{\max}^0 - T_{\min}^0}{K}\right) \log\left(1 + 2^8 \log\left(\frac{12}{\pi^2} K^2 n\right)\right)\right) & \text{if } r = 1 \\ \frac{\mu_1 L}{2^{2r+5} \log\left(\frac{12}{\pi^2} K^2 r^2 n\right)}\left(1 + \left(1 + T_{\max}^0 - T_{\min}^0\right) \log\left(1 + \frac{2^{2r+6} \log\left(\frac{12}{\pi^2} K^2 r^2 n\right)}{\sum_{r'=1}^{r-1} 2^{2r'+6} \log\left(\frac{12}{\pi^2} K^2 (r')^2 n\right)}\right)\right) & \text{if } r > 1 \end{cases}$$
$$\leq \begin{cases} \frac{\mu_1 L}{2^7 \log\left(\frac{12}{\pi^2} K^2 n\right)}\left(1 + \left(1 + \frac{T_{\max}^0 - T_{\min}^0}{K}\right) \log\left(1 + 2^8 \log\left(\frac{12}{\pi^2} K^2 n\right)\right)\right) & \text{if } r = 1 \\ \frac{\mu_1 L}{2^{2r+5} \log\left(\frac{12}{\pi^2} K^2 r^2 n\right)}\left(1 + \left(1 + T_{\max}^0 - T_{\min}^0\right) \log(13)\right) & \text{if } r > 1 \end{cases}$$

where in the last line, we used the choice of $m_r$ and the fact that $\mathcal{A}_r \subseteq \mathcal{A}_{r-1}$. Now, for any $r \geq 2$,

$$
\begin{aligned}
\frac{2^{2r+5} \log\left(\frac{12}{\pi^2} K^2 r^2 n\right)}{\sum_{r'=1}^{r-1} 2^{2r'+5} \log\left(\frac{12}{\pi^2} K^2 (r')^2 n\right)} &\leq \frac{2^{2r+5} \log\left(\frac{12}{\pi^2} K^2 r^2 n\right)}{2^{2(r-1)+5} \log\left(\frac{12}{\pi^2} K^2 (r-1)^2 n\right)} \\
&= 4 \frac{\log\left(\frac{12}{\pi^2} K^2 n\right) + 2\log(r)}{\log\left(\frac{12}{\pi^2} K^2 n\right) + 2\log(r-1)} \\
&\leq 4 \frac{\log\left(\frac{12}{\pi^2} K^2 n\right) + 2\log(2)}{\log\left(\frac{12}{\pi^2} K^2 n\right)} \\
&\leq 12
\end{aligned}
$$

where the penultimate inequality above follows from the fact that the function $\frac{a+2\log(r)}{a+2\log(r-1)}$ is decreasing in $r$ for any $a > 0$ and $r \geq 1$. Hence, our assumption in (11) that $n$ is sufficiently large thus guarantees that:

$$
\widetilde{\mu}_{j'}(r) - \widetilde{\mu}_j(r) \leq 2^{-2r} \leq 2^{-r-1}
$$

Plugging these conclusions into (23), we conclude with the following upper-bound on (21):

$$
\begin{aligned}
&\Pr\left[L_* \subseteq \mathcal{A}_{r'} \subseteq U_{r'} \; \forall r' < r+1, \exists i_* \in L_* : i_* \notin \mathcal{A}_{r+1}\right] \\
&\leq \sum_{j \neq j' \in [K]} \Pr\left[\widehat{\mu}_{j'}(r) - \widehat{\mu}_j(r) - (\widetilde{\mu}_{j'}(r) - \widetilde{\mu}_j(r)) > 2^{-r-1}\right].
\end{aligned} \tag{24}
$$

We bound (22) using nearly identical arguments to (21), as follows: begin by noticing that, by our assumption (WLOG) that $1 \in L_*$:

$$
\begin{aligned}
&\Pr\left[L_* \subseteq \mathcal{A}_{r'} \subseteq U_{r'} \; \forall r' < r+1, \exists j \in U_r \setminus U_{r+1} : j \in \mathcal{A}_{r+1}\right] \\
&= \Pr\left[L_* \subseteq \mathcal{A}_{r'} \subseteq U_{r'} \; \forall r' < r+1, \exists j \in U_r \setminus U_{r+1} : \widehat{\mu}_{\max}(r) - \widehat{\mu}_j(r) \leq 2^{-r}\right] \\
&\leq \Pr\left[L_* \subseteq \mathcal{A}_{r'} \subseteq U_{r'} \; \forall r' < r+1, \exists j \in U_r \setminus U_{r+1} : \widehat{\mu}_1(r) - \widehat{\mu}_j(r) \leq 2^{-r}\right] \\
&\leq \sum_{j \in [K]} \Pr\left[L_* \subseteq \mathcal{A}_{r'} \subseteq U_{r'} \; \forall r' < r+1, j \in U_r \setminus U_{r+1} : \widehat{\mu}_1(r) - \widehat{\mu}_j(r) \leq 2^{-r}\right]
\end{aligned}
$$

Then, by Lemma C.7, assuming $L_* \subseteq \mathcal{A}_{r'} \subseteq U_{r'}$ for all $r' \leq r$, since $1 \in U_r$, by Lemma C.3, and using our previous argument to upper-bound $|\xi_{1,j}(r \mid \mathcal{A}_1, \ldots, \mathcal{A}_r)|$:

$$
\begin{aligned}
\widetilde{\mu}_1(r) - \widetilde{\mu}_j(r) &= \Delta_j \bar{f}_1(r \mid \mathcal{A}_1, \ldots, \mathcal{A}_r) + \mu_j \xi_{1,j}(r \mid \mathcal{A}_1, \ldots, \mathcal{A}_r) \\
&\geq \Delta_j \bar{f}_j^{\min}(r) - \mu_1 |\xi_{1,j}(r \mid \mathcal{A}_1, \ldots, \mathcal{A}_r)| \\
&\geq \Delta_j \bar{f}_j^{\min}(r) - 2^{-r-1}.
\end{aligned}
$$

Now, if $j \in U_r \setminus U_{r+1}$, then $\Delta_j \bar{f}_j^{\min}(r) > 2^{-r+1}$. Therefore, combining the above bounds, we obtain the following bound on (22):

$$
\begin{aligned}
&\Pr\left[L_* \subseteq \mathcal{A}_{r'} \subseteq U_{r'} \; \forall r' < r+1, \exists j \in U_r \setminus U_{r+1} : j \in \mathcal{A}_{r+1}\right] \\
&\leq \sum_{j \in [K]} \Pr\left[\widehat{\mu}_1(r) - \widehat{\mu}_j(r) - (\widetilde{\mu}_1(r) - \widetilde{\mu}_j(r)) \leq -(2^{-r+1} - 2^{-r-1} - 2^{-r})\right] \\
&\leq \sum_{j \in [K]} \Pr\left[\widehat{\mu}_1(r) - \widehat{\mu}_j(r) - (\widetilde{\mu}_1(r) - \widetilde{\mu}_j(r)) \leq -2^{-r-1}\right].
\end{aligned} \tag{25}
$$

Therefore, by the Chernoff bound for subGaussian random variables together with our choice of $m_r$ to bound (24) and (25),

we obtain the bound on (20):

$$\Pr\left[i \in \mathcal{A}_{u_i+1} \text{ or } \exists r \in [u_i] : L_r \not\subseteq \mathcal{A}_r \text{ or } \mathcal{A}_r \not\subseteq U_r\right]$$

$$\leq \sum_{r=1}^{u_i} \Pr\left[L_{r'} \subseteq \mathcal{A}_{r'} \subseteq U_{r'} \ \forall r' < r+1, L_{r+1} \not\subseteq \mathcal{A}_{r+1} \text{ or } \mathcal{A}_{r+1} \not\subseteq U_{r+1}\right]$$

$$\leq \sum_{r=1}^{u_i} \sum_{j\in[K]} \sum_{i_* \neq j} 2\exp\left(-\frac{m_r 2^{-2(r+1)}}{4}\right)$$

$$= \sum_{r=1}^{u_i} \sum_{j\in[K]} \sum_{i_* \neq j} 2\exp\left(-\frac{8\log\left(\frac{12}{\pi^2}K^2 r^2 n\right)}{4}\right)$$

$$\leq \sum_{r=1}^{u_i} \frac{6}{\pi^2 r^2 n}$$

$$\leq \frac{1}{n},$$

as claimed. $\qquad\square$

Using the high probability guarantee from Lemma C.8, we can now obtain an upper bound on the number of times a suboptimal arm is played by Algorithm 2.

**Lemma C.9.** *Suppose that Algorithm 2 is run using the sampling schedule $m_r$ from (10) for a time horizon $n$ satisfying (11). Then, for any suboptimal arm $i$ (i.e., $\Delta_i > 0$), Algorithm 2 satisfies:*

$$\mathbb{E}\left[T_i(n)\right] \leq 1 + \frac{2^{11}}{3}\log\left(\frac{12}{\pi^2}K^2 n^3\right)\left(\frac{1}{\Delta_i \bar{f}_i^{\min}(u_i-1)}\right)^2,$$

*where $\bar{f}_i^{\min}(r)$ is as defined in Definition C.2.*

*Proof.* We begin by decomposing $\mathbb{E}\left[T_i(n)\right]$ as follows:

$$\mathbb{E}\left[T_i(n)\right] = \mathbb{E}\left[T_i(n)\mathbb{1}\{i \notin \mathcal{A}_{u_i+1}\}\right] + \mathbb{E}\left[T_i(n)\mathbb{1}\{i \in \mathcal{A}_{u_i+1}\}\right],$$

where $u_i$ is as defined in Definition C.2. Then, by Lemma C.8, and since $T_i(n) \leq n$ deterministically by definition,

$$\mathbb{E}\left[T_i(n)\mathbb{1}\{i \in \mathcal{A}_{u_i+1}\}\right] \leq n\Pr\left[i \in \mathcal{A}_{u_i+1}\right] \leq 1.$$

For the remaining term, whenever arm $i$ is removed at round $u_i + 1$, and since there are at most $n$ rounds (since more than one arm is pulled during each round), we have:

$$\mathbb{E}\left[T_i(n)\mathbb{1}\{i \notin \mathcal{A}_{u_i+1}\}\right] \leq \sum_{r=1}^{\min\{u_i,n\}} m_r = \sum_{r=1}^{\min\{u_i,n\}} 2^{2r+5}\log\left(\frac{12}{\pi^2}K^2 r^2 n\right) \leq \frac{2^7}{3}\log\left(\frac{12}{\pi^2}K^2 n^3\right)2^{2u_i}.$$

Now, by definition of $u_i$ in Definition C.2, we know that $i \in U_{u_i}$, i.e., $\Delta_i \bar{f}_i^{\min}(u_i-1) \leq 2^{-(u_i-1)+1}$. Therefore, $2^{u_i} \leq \frac{4}{\Delta_i \bar{f}_i^{\min}(u_i-1)}$, which implies:

$$\mathbb{E}\left[T_i(n)\right] \leq 1 + \frac{2^{11}}{3}\log\left(\frac{12}{\pi^2}K^2 n^3\right)\left(\frac{1}{\Delta_i \bar{f}_i^{\min}(u_i-1)}\right)^2,$$

as claimed. $\qquad\square$

Finally, using Lemma C.9, we can establish Theorem C.4.

*Proof of Theorem C.4.* Plugging in Lemma C.9 to the regret decomposition from (1), we obtain:

$$
\begin{aligned}
R_{\boldsymbol{\nu},\pi}(n) &= \sum_{i:\Delta_i>0} \Delta_i \mathbb{E}\left[T_i(n)\right] \\
&\leq \sum_{i:\Delta_i>0} \Delta_i \left(1 + \frac{2^{11}}{3}\log\left(\frac{12}{\pi^2}K^2 n^3\right)\left(\frac{1}{\Delta_i \bar{f}_i^{\min}(u_i-1)}^2\right)\right) \\
&= \sum_{i:\Delta_i>0} \Delta_i + \frac{2^{11}}{3}\log\left(\frac{12}{\pi^2}K^2 n^3\right)\frac{1}{\Delta_i \bar{f}_i^{\min}(u_i-1)^2},
\end{aligned}
$$

as claimed in (12).

(13) follows by decomposing the divergence decomposition in two parts: for a fixed $\Delta$, note that we can write

$$
\begin{aligned}
R_{\boldsymbol{\nu},\pi}(n) &= \sum_{i:0<\Delta_i\leq\Delta} \Delta_i \mathbb{E}\left[T_i(n)\right] + \sum_{i:\Delta_i>\Delta} \Delta_i \mathbb{E}\left[T_i(n)\right] \\
&\leq \Delta \sum_{i:0<\Delta_i\leq\Delta} \mathbb{E}\left[T_i(n)\right] + \sum_{i:\Delta_i>\Delta} \Delta_i \mathbb{E}\left[T_i(n)\right] \\
&\leq n\Delta + \sum_{i:\Delta_i>\Delta} \Delta_i \mathbb{E}\left[T_i(n)\right].
\end{aligned}
$$

Choosing $\Delta = \max_{i\notin L_*}\sqrt{\frac{2^{11}K\log\left(\frac{12}{\pi^2}K^2 n^3\right)}{3n\bar{f}_i^{\min}(u_i-1)^2}}$ and applying Lemma C.9, we obtain:

$$
\begin{aligned}
R_{\boldsymbol{\nu},\pi}(n) &\leq \max_{i\notin L_*}\sqrt{\frac{2^{11}nK\log\left(\frac{12}{\pi^2}K^2 n^3\right)}{3\bar{f}_i^{\min}(u_i-1)^2}} + \sum_{i:\Delta_i>\Delta}\Delta_i + \frac{2^{11}}{3}\log\left(\frac{12}{\pi^2}K^2 n^3\right)\frac{1}{\Delta_i \bar{f}_i^{\min}(u_i-1)^2} \\
&\leq K + \max_{i\notin L_*}\sqrt{\frac{2^{11}nK\log\left(\frac{12}{\pi^2}K^2 n^3\right)}{3\bar{f}_i^{\min}(u_i-1)^2}} + \sum_{i:\Delta_i>\Delta}\frac{2^{11}}{3}\log\left(\frac{12}{\pi^2}K^2 n^3\right)\frac{1}{\Delta\bar{f}_i^{\min}(u_i-1)^2} \\
&\leq K + \max_{i\notin L_*}\sqrt{\frac{2^{11}nK\log\left(\frac{12}{\pi^2}K^2 n^3\right)}{3\bar{f}_i^{\min}(u_i-1)^2}} + \sum_{i:\Delta_i>\Delta}\sqrt{\frac{2^{11}n\log\left(\frac{12}{\pi^2}K^2 n^3\right)}{3K}}\min_{j\notin L_*}\frac{\bar{f}_j^{\min}(u_j-1)}{\bar{f}_i^{\min}(u_i-1)^2} \\
&\leq K + 2\max_{i\notin L_*}\sqrt{\frac{2^{11}nK\log\left(\frac{12}{\pi^2}K^2 n^3\right)}{3\bar{f}_i^{\min}(u_i-1)^2}}.
\end{aligned}
$$

We further simplify this expression as follows. Recalling the definition of $\bar{f}_i^{\min}(r)$ from Definition C.2:

$$
\begin{aligned}
\bar{f}_i^{\min}(r) &= \bar{f}_i(r \mid U_1,\ldots,U_r) \\
&= \frac{1}{m_r}\sum_{\ell\in[m_r]} f\left(\frac{T_i^0 + \sum_{r'<r}m_{r'} + (\ell-1)}{t_0^{\text{bias}} + \sum_{r'<r}|U_{r'}|m_{r'} + |U_r|(\ell-1)b_i(U_r)}\right).
\end{aligned}
$$

Observing that, since $[K]=U_1 \supseteq U_{r'} \supseteq U_r$ for every $r' < r$ by definition, and hence also since $b_i(U_r) = \sum_{j\in U_r}\mathbb{1}\{j <$

$i\} \leq b_i(U_{r'}) \leq b_i(U_1) = i - 1,$

$$\frac{T_i^0 + \sum_{r' < r} m_{r'} + (\ell - 1)}{t_0^{\text{bias}} + \sum_{r' < r} |U_{r'}| m_{r'} + |U_r|(\ell - 1) + b_i(U_r)}$$

$$\geq \frac{T_i^0 + \sum_{r' < r} m_{r'} + (\ell - 1)}{t_0^{\text{bias}} + K \sum_{r' < r} m_{r'} + K(\ell - 1) + (i - 1)}$$

$$\geq \frac{T_i^0}{t_0^{\text{bias}} + (i - 1)} \frac{t_0^{\text{bias}} + (i - 1)}{t_0^{\text{bias}} + K \sum_{r' < r} m_{r'} + K(\ell - 1) + (i - 1)}$$

$$+ \frac{\sum_{r' < r} m_{r'} + (\ell - 1)}{K \left( \sum_{r' < r} m_{r'} + (\ell - 1) \right)} \left( 1 - \frac{t_0^{\text{bias}} + (i - 1)}{t_0^{\text{bias}} + K \sum_{r' < r} m_{r'} + K(\ell - 1) + (i - 1)} \right)$$

$$\geq \min \left\{ \frac{T_i^0}{t_0^{\text{bias}} + i - 1}, \frac{1}{K} \right\}$$

$$\geq \frac{T_{\min}^0}{t_0^{\text{bias}} + K - 1}.$$

Therefore, since $f(\cdot)$ is non-decreasing by Assumption 2.1, it follows that

$$\bar{f}_i^{\min}(r) \geq f \left( \min \left\{ \frac{T_i^0}{t_0^{\text{bias}} + i - 1}, \frac{1}{K} \right\} \right) \geq f \left( \frac{T_{\min}^0}{t_0^{\text{bias}} + K - 1} \right).$$

Plugging in this bound to the above expression, we conclude that:

$$R_{\boldsymbol{\nu}, \pi}(n) \leq K + 2 \sqrt{\frac{2^{11} nK \log \left( \frac{12}{\pi^2} K^2 n^3 \right)}{3 f \left( \frac{T_{\min}^0}{t_0^{\text{bias}} + K - 1} \right)^2}}.$$

which gives (13).

and using the fact that $\bar{f}_{\min}(u_{i_1} - 1) \leq \bar{f}_{\min}(u_i - 1)$ (since $f(x)$ is nondecreasing in $x$ and $U_{r+1} \subseteq U_r$ for all $r$, and thus $\bar{f}_{\min}(r)$ is increasing as a function of $r$) yields the claim. $\qquad \square$

## D. Asymptotic instance-dependent lower bound – Deferred proofs

We show the following lower bound for any "reasonable" algorithm for our setting:

**Definition 5.1** (Consistent policy). Let $\mathcal{E}$ be a set of unbiased bandit environments $\boldsymbol{\nu}$ with bias model following Assumption 2.1 with a *fixed and common* set of initial biases $\{T_i^0\}_{i \in [K]}$ and reweighting function $W_i(t) = f \left( T_i^{\text{bias}}(t-1) / (t^{\text{bias}} - 1) \right)$. We call a family of bandit policies $\{\pi_n\}_{n \geq 1}$ *consistent* for an environment class $\mathcal{E}$ (with its associated bias model) if there are constants[8] $C > 0$ and $a \in (0, 1)$ such that, for all $n \geq 1$ and $\boldsymbol{\nu} \in \mathcal{E}$, the regret of policy $\pi_n$ is bounded as: $R_{\boldsymbol{\nu}, \pi_n}(n) \leq C \cdot n^a$.

When the dependence of $\pi_n$ on $n$ is clear from context, we adopt a slight abuse of notation and call the policy $\pi$ consistent.

**Theorem D.1** (Formal statement of Theorem 5.2). *Fix any $K > 1$, initial biases $\{T_i^0\}_{i \in [K]}$, and time horizon $n \geq 1$. Consider the following environment class:*

$$\mathcal{E}_{\mathcal{N}} = \left\{ \boldsymbol{\nu} = (\nu_i)_{i \in [K]} : \nu_i = \mathcal{N}(\mu_i, 1) \text{ for some } \Delta_i \in [0, 1] \right\}.$$

*Consider any instance $\boldsymbol{\nu}$ such that, for some constants $\gamma \in (0, 1)$ and $c, c' \geq 0$, there is a subset of suboptimal arms $\mathcal{A}_{c, c'} \subset [K]$ satisfying:*

$$|\mathcal{A}_{c, c'}| \geq \gamma K \quad \text{and} \quad \log \left( \frac{\Delta_i}{\Delta_i} \right) \leq c \log(K)^{c'} \ \forall i, j \in \mathcal{A}_{c, c'}.$$

---

[8]Note that this constant $C$ may depend on the common environment parameters of $\mathcal{E}$ such as $K$, the initial biases $T_i^0$, and the bias function $f(\cdot)$.

*Let $\boldsymbol{\nu}^{\text{bias}}$ be the associated biased environment which satisfies Assumption 2.1 where the mean reweighting function $f(\cdot)$ additionally satisfies $-\log(f(2/\gamma K)) \leq c\log(K)^{c'}$. Let $\pi$ be any consistent bandit policy for $\mathcal{E}_{\mathcal{N}}$ (with constants $C > 0$ and $a \in (0,1)$ as in Definition 5.1). Then, for any bandit policy $\pi$ and any $\varepsilon \in (0, \frac{1-\mu_{\max}}{\Delta_{\max}}]$, there is a set $B' \subseteq |\mathcal{A}_{c,c'}|$ such that $|B'| \geq |\mathcal{A}_{c,c'}|/4$ and parameter $\beta' = \mathcal{O}(1/\gamma(\log(K/\gamma) + c\log(K)^{c'}))$ such that:*

$$R_{\boldsymbol{\nu},\pi}(n) \gtrsim f\left(\frac{\beta'}{K}\right)^{-2} \sum_{i \in B':\Delta_i>0} \frac{\gamma(1-a)\log(n)}{\Delta_i} + \sum_{i \notin B':\Delta_i>0} \frac{(1-a)\log(n)}{\Delta_i}. \tag{26}$$

*In particular, whenever $\Delta_{\max}/\Delta_{\min} \leq c$ and $\boldsymbol{\nu}$ has a unique optimal arm $i_*$, then we may take $\mathcal{A}_{c,0} = [K] \setminus \{i_*\}$ and $\gamma = (K-1)/K$, and we obtain the simplified regret bound:*

$$R_{\boldsymbol{\nu},\pi}(n) \gtrsim f\left(\frac{\beta'}{K}\right)^{-2} \sum_{i:\Delta_i>0} \frac{\log(n)}{\Delta_i}. \tag{27}$$

*Remark* 5.3 (Comparison to standard bandit regret lower bound). We note that in the standard, unbiased setting, Lai & Robbins (1985) established a lower bound for Gaussian bandits of the form $R_{\boldsymbol{\nu},\pi}(n) \geq \sum_{i:\Delta_i>0} \frac{2\log(n)}{\Delta_i} - \mathcal{O}(1)$. They also gave an algorithm with regret asymptotically matching their lower bound. Our lower bound shows that, at least in the setting where the maximum ratio of (nonzero) suboptimality gaps is not too small, then the regret in the biased setting we study must be at least a factor (roughly) $f(\mathcal{O}(\log(K)/K))^{-2}$ larger than in the standard setting.

**Corollary 5.4** (Comparison of Theorem 4.1 and Theorem 5.2). *Under the conditions in Theorems 4.1 and 5.2a, suppose additionally that the bias model satisfies the following: for any $x \in (0,1)$ and $\mu \in (1, 1/x)$, there is a constant $L' > 0$ such that $f(\mu x) \leq \mu^{L'} f(x)$. Then, for sufficiently large time horizons, the regret bound of Algorithm 1 in Theorem 4.1 matches Theorem 5.2 up to a multiplicative $\mathcal{O}(\log(K)^{2L'})$ factor.*

*Proof.* The corollary is an immediate consequence of the fact that, from the additional assumption on the bias model $f(\cdot)$,

$$f\left(\frac{c\log(K)}{K}\right) = f\left(15c\log(K)\frac{1}{15K}\right) \leq (15c\log(K))^{L'} f\left(\frac{1}{15K}\right).$$

$\square$

*Remark* D.2 (A comment on the additional assumption in Corollary 5.4). Many natural examples of functions satisfying Assumption 2.1 also satisfy the assumption from Corollary 5.4. For example, $f(x) = \min\{x^\alpha, c\}$ is $\max\{\alpha(\rho_{\min})^{\alpha-1}, 1\}$-Lipschitz for every $c \in [0,1]$ and also satisfies the assumption in Corollary 5.4. However, not all functions satisfying Assumption 2.1 also satisfy this additional assumption. For example, consider:

$$f(x) = \begin{cases} \frac{1}{K^{100}} & \text{if } x \in [0, \beta/K] \\ \left(x - \frac{\beta}{K} + \frac{1}{K^{10}}\right)^{10} & \text{if } x \in (\beta/K, 1] \end{cases}$$

for a constant $c \in (1, \log(K))$. Notice that this function is 10-Lipschitz, nondecreasing, and also $-\log(f(\beta/K)) = \mathcal{O}(\log(K))$ (thus satisfying all assumptions from Assumption 2.1 as well as the additional assumption in Corollary 5.4 from Theorem 5.2). However, for $\beta < \beta' = \mathcal{O}(\log(K)^{c'})$ and any constant $L'$,

$$\frac{f(\beta'/K)}{f(\beta/K)} = \frac{\left(\frac{\beta'-\beta}{K} + \frac{1}{K^{10}}\right)^{10}}{\frac{1}{K^{100}}} = K^{10}(\beta'-\beta)^{10} \gg (\beta')^{L'}.$$

## D.1. Notation overview

We briefly discuss some notation that will be used throughout the proofs. Given an (unbiased) stochastic bandit environment $\boldsymbol{\nu} = (\nu_i)_{i \in [K]}$ and a bias model satisfying Assumption 2.1, we will use $\boldsymbol{\nu}^{\text{bias}}$ to denote the associated environment with biased feedback. A bandit policy interacts with $\boldsymbol{\nu}^{\text{bias}}$ sequentially over $n$ rounds, selecting an action $A_t \in [K]$ and observing feedback $Y_t$ sampled according to the associated bias model. We denote $\mathcal{H}_n = (U_0, \mathcal{A}_1, Y_1, U_1 \ldots, \mathcal{A}_n, Y_n, U_n)$ to be the $n$-round interaction history between the policy $\pi$ and environment $\boldsymbol{\nu}^{\text{bias}}$, where $U_{t-1}$ denotes any additional randomness used by the policy $\pi$ in selecting action $A_t$. $\mathcal{F}_n = \sigma\{\mathcal{H}_n\}$ is the associated sigma algebra. We write $\Pr_{\boldsymbol{\nu}^{\text{bias}},\pi}[\cdot]$ and $\mathbb{E}_{\boldsymbol{\nu}^{\text{bias}},\pi}[\cdot]$ as the probability and expectations induced by the interactions between policy $\pi$ and environment $\boldsymbol{\nu}^{\text{bias}}$.

### D.2. Proofs

Our proof begins by generalization of (Kaufmann et al., 2016, Lemma 1) and (Garivier et al., 2019, Eq. (8)).

**Lemma D.3** (A divergence decomposition for biased environments; Formal statement of Lemma 5.5). *Fix a time horizon $n \geq 1$, and let $\boldsymbol{\nu} \in \mathcal{E}_{\mathcal{N}}$ be an environment with associated biased environment $\boldsymbol{\nu}^{\mathrm{bias}}$. Fix any $i \in [K]$ which is suboptimal in $\boldsymbol{\nu}$, and let $\boldsymbol{\nu}^{(i)} \in \mathcal{E}_{\mathcal{N}}$ be an environment satisfying $\nu_j^{(i)} = \nu_j$ for every $j \neq i$. Let $\pi$ be any consistent bandit policy satisfying Definition 5.1. Let $\tau_i \leq n$ be a stopping time w.r.t. the filtration $\mathcal{F}_t$. Let $Z_i \in [0,1]$ be any $\mathcal{F}_{\tau_i}$-measurable random variable. Then,*

$$D_{\mathrm{KL}}(\nu_i \| \nu_i^{(i)}) \underset{\boldsymbol{\nu}^{\mathrm{bias}}, \pi}{\mathbb{E}} \left[ \sum_{t \in [\tau_i]} W_i(t)^2 \mathbb{1}\{A_t = i\} \right] = D_{\mathrm{KL}}(\mathrm{Pr}_{\boldsymbol{\nu}^{\mathrm{bias}}, \pi}^{\mathcal{H}_{\tau_i}} \| \mathrm{Pr}_{\boldsymbol{\nu}^{\mathrm{bias}, (i)}, \pi}^{\mathcal{H}_{\tau_i}}) \tag{28}$$

$$\geq \mathrm{kl}\left( \underset{\boldsymbol{\nu}^{\mathrm{bias}}, \pi}{\mathbb{E}}[Z_i], \underset{\boldsymbol{\nu}^{\mathrm{bias}, (i)}, \pi}{\mathbb{E}}[Z_i] \right). \tag{29}$$

Before proving Lemma 5.5, we discuss a couple of immediate applications of this result to our setting. From Lemma 5.5, we have the following immediate consequence:

**Claim D.4** (Formal statement of Claim 5.6). *Consider the same setting as in Lemma 5.5, where arm $i$ is suboptimal in $\boldsymbol{\nu}$. Choose $\boldsymbol{\nu}^{(i)}$ satisfying $\nu_j^{(i)} = \nu_j$ for $j \neq i$ and $\nu_i^{(i)} = \mathcal{N}(\mu_i + (1+\varepsilon)\Delta_i, 1)$ for any $\varepsilon > 0$ such that $\mu_i + (1+\varepsilon)\Delta_i \leq 1$. Then, for any consistent policy $\pi$ (according to Definition 5.1),*

$$\underset{\boldsymbol{\nu}^{\mathrm{bias}}, \pi}{\mathbb{E}}[T_i(n)] \geq \frac{2}{(1+\varepsilon)^2 \Delta_i^2} \left( \left( 1 - \frac{C}{\Delta_i n^{1-a}} \right) \log\left( \frac{\varepsilon \Delta_i n^{1-a}}{C} \right) - \log(2) \right)$$

$$= \frac{2(1-a)}{(1+\varepsilon)^2 \Delta_i^2} \log(n) - \mathcal{O}(1). \tag{30}$$

*If, in addition, $\pi$ has the property that $\frac{T_i^{\mathrm{bias}}(t-1)}{t^{\mathrm{bias}}-1} \leq \frac{\beta'}{K}$ for every $t \in (n_0, \tau_i]$ for any stopping time $n_0 \leq \tau_i \leq n$ w.r.t. $\mathcal{F}_t = \sigma\{\mathcal{H}_t\}$ such that $\mathbb{E}_{\boldsymbol{\nu}^{\mathrm{bias}}, \pi}[\tau_i] \geq cn$ for $c \in (0,1)$, then:*

$$\underset{\boldsymbol{\nu}^{\mathrm{bias}}, \pi}{\mathbb{E}}[T_i(n_0, \tau_i)]$$

$$\geq f\left(\frac{\beta'}{K}\right)^{-2} \frac{2}{(1+\varepsilon)^2 \Delta_i^2} \left( \left( c - \frac{C}{\Delta_i n^{1-a}} \right) \log\left( \frac{\varepsilon \Delta_i n^{1-a}}{C} \right) - \log(2) - \frac{(1+\varepsilon)^2 \Delta_i^2}{2} \underset{\boldsymbol{\nu}^{\mathrm{bias}}, \pi}{\mathbb{E}}[T_i(n_0)] \right)$$

$$\geq f\left(\frac{\beta'}{K}\right)^{-2} \frac{2c(1-a)}{(1+\varepsilon)^2 \Delta_i^2} \left( \log(n) - \frac{(1+\varepsilon)^2 \Delta_i^2}{2c(1-a)} \underset{\boldsymbol{\nu}^{\mathrm{bias}}, \pi}{\mathbb{E}}[T_i(n_0)] - \mathcal{O}(1) \right) \tag{31}$$

$$\geq f\left(\frac{\beta'}{K}\right)^{-2} \frac{2c(1-a)}{(1+\varepsilon)^2 \Delta_i^2} \left( \log(n) - \frac{(1+\varepsilon)^2 \Delta_i^2}{2c(1-a)} n_0 - \mathcal{O}(1) \right), \tag{32}$$

*where $f(x)$ is as defined in Assumption 2.1.*

*Proof.* Note that $\frac{T_i^{\mathrm{bias}}(t-1)}{t^{\mathrm{bias}}-1} \leq 1$ deterministically, and:

$$\mathrm{kl}(p, q) = p \log\left( \frac{1}{q} \right) + (1-p) \log\left( \frac{1}{1-q} \right) + (p \log(p) + (1-p) \log(1-p))$$

$$\geq p \log\left( \frac{1}{q} \right) - \log(2), \tag{33}$$

which follows from the facts that $p \log(p) + (1-p) \log(1-p) \geq -\log(2)$ and $(1-p) \log\left( \frac{1}{1-q} \right) \geq 0$. Thus Lemma 5.5 together with (33) implies that, for any stopping time $\tau_i$ and any $\mathcal{F}_{\tau_i}$-measurable $Z_i$:

$$D_{\mathrm{KL}}(\nu_i \| \nu_i^{(i)}) \underset{\boldsymbol{\nu}^{\mathrm{bias}}, \pi}{\mathbb{E}}[T_i(\tau_i)] \geq \underset{\boldsymbol{\nu}^{\mathrm{bias}}, \pi}{\mathbb{E}}[Z_i] \log\left( \frac{1}{\mathbb{E}_{\boldsymbol{\nu}^{\mathrm{bias},'}, \pi}[Z_i]} \right) - \log(2). \tag{34}$$

which is essentially the result of (Garivier et al., 2019, Eq. (8)). In particular, let us choose $\nu_i^{(i)} = \mathcal{N}(\mu_i + (1 + \varepsilon)\Delta_i, 1)$, $\tau_i = n$, and

$$Z_i = \frac{n - T_i(n)}{n} = \frac{\sum_{j \neq i} T_j(n)}{n}.$$

Then (34) implies:

$$
\begin{aligned}
\frac{(1 + \varepsilon)^2 \Delta_i^2}{2} \mathop{\mathbb{E}}_{\boldsymbol{\nu}^{\text{bias}}, \pi} [T_i(n)] &\geq \left(1 - \frac{\mathbb{E}_{\boldsymbol{\nu}^{\text{bias}}, \pi} [T_i(n)]}{n}\right) \log\left(\frac{n}{\sum_{j \neq i} \mathbb{E}_{\boldsymbol{\nu}^{\text{bias},(i)}, \pi} [T_i(n)]}\right) - \log(2) \\
&\geq \left(1 - \frac{C}{\Delta_i n^{1-a}}\right) \log\left(\frac{\varepsilon \Delta_i n^{1-a}}{C}\right) - \log(2),
\end{aligned}
\tag{35}
$$

where, in the first line, we used the fact that $D_{\text{KL}}(\mathcal{N}(\mu, 1) \| \mathcal{N}(\mu', 1)) = \frac{(\mu - \mu')^2}{2}$ and, in the last line, we used the fact that, by assumption on $\pi$,

$$\Delta_i \mathop{\mathbb{E}}_{\boldsymbol{\nu}^{\text{bias}}, \pi} [T_i(n)] \leq R_{\boldsymbol{\nu}, \pi}(n) \leq Cn^a, \text{ and} \tag{36}$$

$$\varepsilon \Delta_i \sum_{j \neq i} \mathop{\mathbb{E}}_{\boldsymbol{\nu}^{\text{bias},(i)}, \pi} [T_j(n)] \leq \sum_{j \neq i} (\Delta_j + \varepsilon \Delta_i) \mathop{\mathbb{E}}_{\boldsymbol{\nu}^{\text{bias},(i)}, \pi} [T_j(n)] \leq R_{\boldsymbol{\nu}^{(i)}, \pi}(n) \leq Cn^a, \tag{37}$$

which establishes the first part of the claim.

For the second part of the claim, we begin by recalling that, since $\frac{T_i^{\text{bias}}(t-1)}{t^{\text{bias}} - 1} \leq \beta'/K$ for all $t \in (n_0, \tau_i]$ by assumption on $\pi$, we have that, by Assumption 2.1, $W_i(t) = f\left(\frac{T_i^{\text{bias}}(t-1)}{t^{\text{bias}} - 1}\right) \leq f\left(\frac{\beta'}{K}\right)$ for all $t \in (n_0, \tau_i]$ (since $f(x)$ is nondecreasing in $x$). Using this bound together with the (trivial) bound $W_i(t) \leq 1$, we obtain:

$$
\begin{aligned}
\mathop{\mathbb{E}}_{\boldsymbol{\nu}^{\text{bias}}, \pi} \left[\sum_{t \in [\tau_i]} W_i(t)^2 \mathbb{1}\{A_t = i\}\right] &\leq \mathop{\mathbb{E}}_{\boldsymbol{\nu}^{\text{bias}}, \pi} \left[\sum_{t \in [n_0]} \mathbb{1}\{A_t = i\}\right] + \mathop{\mathbb{E}}_{\boldsymbol{\nu}^{\text{bias}}, \pi} \left[\sum_{t = n_0 + 1}^{\tau_i} W_i(t)^2 \mathbb{1}\{A_t = i\}\right] \\
&\leq \mathop{\mathbb{E}}_{\boldsymbol{\nu}^{\text{bias}}, \pi} [T_i(n_0)] + f\left(\frac{\beta'}{K}\right)^2 \mathop{\mathbb{E}}_{\boldsymbol{\nu}^{\text{bias}}, \pi} [T_i(n_0, \tau_i)].
\end{aligned}
\tag{38}
$$

Now, let us choose

$$Z_i = \frac{\tau_i - T_i(\tau_i)}{n} = \frac{\sum_{j \neq i} T_j(\tau_i)}{n},$$

which, we note is almost the same as the choice of $Z_i$ used in proving (30), except modified to guarantee that $Z$ is $\mathcal{F}_{\tau_i}$-measurable. Then, using (33), we have that:

$$
\begin{aligned}
\text{kl}&\left(\mathop{\mathbb{E}}_{\boldsymbol{\nu}^{\text{bias}}, \pi} [Z_i], \mathop{\mathbb{E}}_{\boldsymbol{\nu}^{\text{bias},(i)}, \pi} [Z_i]\right) + \log(2) \\
&\geq \frac{\sum_{j \neq i} \mathbb{E}_{\boldsymbol{\nu}^{\text{bias}}, \pi} [T_j(\tau_i)]}{n} \log\left(\frac{n}{\sum_{j \neq i} \mathbb{E}_{\boldsymbol{\nu}^{\text{bias},(i)}, \pi} [T_j(\tau_i)]}\right) \\
&= \frac{\mathbb{E}_{\boldsymbol{\nu}^{\text{bias}}, \pi} [\tau_i] - \mathbb{E}_{\boldsymbol{\nu}^{\text{bias}}, \pi} [T_i(\tau_i)]}{n} \log\left(\frac{n}{\sum_{j \neq i} \mathbb{E}_{\boldsymbol{\nu}^{\text{bias},(i)}, \pi} [T_j(\tau_i)]}\right) \\
&\geq \frac{cn - \mathbb{E}_{\boldsymbol{\nu}^{\text{bias}}, \pi} [T_i(n)]}{n} \log\left(\frac{n}{\sum_{j \neq i} \mathbb{E}_{\boldsymbol{\nu}^{\text{bias},(i)}, \pi} [T_j(n)]}\right),
\end{aligned}
$$

where, in the last line, we used the fact that $T_i(\tau_i) \leq T_i(n)$ and $\mathbb{E}_{\boldsymbol{\nu}^{\text{bias}}, \pi} [\tau_i] \geq cn$. Therefore, by plugging in the above

bound and (38) to Lemma 5.5, we obtain:

$$
\frac{(1+\varepsilon)^2 \Delta_i^2}{2} \left( \mathop{\mathbb{E}}_{\boldsymbol{\nu}^{\mathrm{bias}}, \pi} [T_i(n_0)] + f\left(\frac{\beta'}{K}\right)^2 \mathop{\mathbb{E}}_{\boldsymbol{\nu}^{\mathrm{bias}}, \pi} [T_i(n_0, \tau_i)] \right)
$$

$$
\geq D_{\mathrm{KL}}(\nu_i \| \nu_i^{(i)}) \mathop{\mathbb{E}}_{\boldsymbol{\nu}^{\mathrm{bias}}, \pi} \left[ \sum_{t \in [\tau_i]} W_i(t)^2 \mathbb{1}\{A_t = i\} \right]
$$

$$
\geq D_{\mathrm{KL}}(\mathop{\mathbb{E}}_{\boldsymbol{\nu}^{\mathrm{bias}}, \pi} [Z] \| \mathop{\mathbb{E}}_{\boldsymbol{\nu}^{\mathrm{bias},(i)}, \pi} [Z])
$$

$$
\geq \frac{cn - \mathbb{E}_{\boldsymbol{\nu}^{\mathrm{bias}}, \pi}[T_i(n)]}{n} \log\left( \frac{n}{\sum_{j \neq i} \mathbb{E}_{\boldsymbol{\nu}^{\mathrm{bias},(i)}, \pi}[T_j(n)]} \right),
$$

which, after lower-bounding the RHS above using (36) and rearranging, is (31). (32) follows immediately from (31), using the fact that $T_i(n_0) \leq n_0$.

$\square$

*Proof of Lemma D.3.* Our proof extends the arguments used in proving (Garivier et al., 2019, Eq. (8)) and (Kaufmann et al., 2016, Lemma 1) to biased feedback settings. We start by recalling (Garivier et al., 2019, Lemma 1):

**Lemma D.5** (Lemma 1, (Garivier et al., 2019)). *Let $(\Omega, \mathcal{F})$ be a measurable space equipt with probability measures $\mathbb{P}_1, \mathbb{P}_2$, and let $Z : \Omega \to [0, 1]$ be a $\mathcal{F}$-measurable function. Denote $\mathbb{E}_1$ and $\mathbb{E}_2$ be the expectations under $\mathbb{P}_1$ and $\mathbb{P}_2$, respectively. Then,*

$$
D_{\mathrm{KL}}(\mathbb{P}_1 \| \mathbb{P}_2) \geq \mathrm{kl}\left( \mathbb{E}_1[Z], \mathbb{E}_2[Z] \right),
$$

*where $\mathrm{kl}(p, q) = p \log\left(\frac{p}{q}\right) + (1-p)\log\left(\frac{1-p}{1-q}\right)$ denotes the KL-divergence between two Bernoulli random variables with means $p$ and $q$.*

We will apply Lemma D.5 as follows: let $\boldsymbol{\nu}^{\mathrm{bias}}$ and $\boldsymbol{\nu}^{\mathrm{bias},'}$ be two biased bandit instances (both having the same initial biases $\{T_i^0\}_{i \in [K]}$), let $n \in \mathbb{N}$ be a time horizon, and let $\pi$ be a bandit policy (possibly depending on $n$). Let $\Omega = \mathbb{R}^{nk}$ and $\mathcal{F} = \mathcal{B}(\Omega)$ denotes the Borel $\sigma$-algebra on $\Omega$, and let $\mathrm{Pr}_{\boldsymbol{\nu}^{\mathrm{bias}}, \pi}$ and $\mathrm{Pr}_{\boldsymbol{\nu}^{\mathrm{bias},'}, \pi}$ be the probability measures on $(\Omega, \mathcal{F})$ induced by the $n$-round interaction between the bandit policy $\pi$ and the environment $\boldsymbol{\nu}^{\mathrm{bias}}$ and $\boldsymbol{\nu}^{\mathrm{bias},'}$, respectively. Let $\mathcal{H}_t = (U_0^{\mathrm{a}}, A_1, Y_1, \ldots, U_{t-1}^{\mathrm{a}}, A_t, Y_t, U_t^{\mathrm{a}})$ denote the interaction history (plus any auxiliary randomness $U_t^{\mathrm{a}}$, sampled from Uniform$([0,1])$ WLOG, used by the policy $\pi$ when selecting action $A_t$) until time $t$. Then, let $\tau \in [n]$ be a stopping time w.r.t. the filtration $\mathcal{F}_t = \sigma\{\mathcal{H}_t\}$, and take $\mathrm{Pr}_{\boldsymbol{\nu}^{\mathrm{bias}}, \pi}^{\mathcal{H}_\tau}$ and $\mathrm{Pr}_{\boldsymbol{\nu}^{\mathrm{bias},'}, \pi}^{\mathcal{H}_\tau}$ to be the pushforward measures of $\mathcal{H}_\tau$ under $\mathbb{P}_{\nu, \pi}$ and $\mathbb{P}_{\nu', \pi}$, respectively. Thus, by Lemma D.5, we have that, for any $\mathcal{F}_\tau$-measurable random variable $Z$:

$$
D_{\mathrm{KL}}(\mathrm{Pr}_{\boldsymbol{\nu}^{\mathrm{bias}}, \pi}^{\mathcal{H}_\tau} \| \mathrm{Pr}_{\boldsymbol{\nu}^{\mathrm{bias},'}, \pi}^{\mathcal{H}_\tau}) \geq \mathrm{kl}\left( \mathbb{E}_{\boldsymbol{\nu}^{\mathrm{bias}}, \pi}^{\mathcal{H}_\tau}[Z], \mathbb{E}_{\boldsymbol{\nu}^{\mathrm{bias},'}, \pi}^{\mathcal{H}_t}[Z] \right) \tag{39}
$$

Now, our claim follows by a standard application of Wald's lemma and the divergence decomposition (similar to the arguments, e.g., in (Lattimore & Szepesvári, 2020, Lemma 15.1 and Exercise 15.7)) to the LHS of (39). Indeed, since $D_{\mathrm{KL}}(\nu_{i,t}^{\mathrm{bias}} \| \nu_{i,t}^{\mathrm{bias},'}) < \infty$ for all $i, t$, and denoting $p_{\boldsymbol{\nu}^{\mathrm{bias}}, \pi}(\cdot)$ (resp., $p_{\boldsymbol{\nu}^{\mathrm{bias},'}, \pi}(\cdot)$) as the density of $\mathrm{Pr}_{\boldsymbol{\nu}^{\mathrm{bias}}, \pi}^{\mathcal{H}_\tau}$ (resp., $\mathrm{Pr}_{\boldsymbol{\nu}^{\mathrm{bias},'}, \pi}^{\mathcal{H}_\tau}$), we can write the KL divergence as follows:

$$
D_{\mathrm{KL}}(\mathrm{Pr}_{\boldsymbol{\nu}^{\mathrm{bias}}, \pi}^{\mathcal{H}_\tau} \| \mathrm{Pr}_{\boldsymbol{\nu}^{\mathrm{bias},'}, \pi}^{\mathcal{H}_\tau})
$$

$$
= \mathbb{E}_{\boldsymbol{\nu}^{\mathrm{bias}}, \pi}^{\mathcal{H}_\tau} \left[ \log\left( \frac{p_{\boldsymbol{\nu}^{\mathrm{bias}}, \pi}(\mathcal{H}_\tau)}{p_{\boldsymbol{\nu}^{\mathrm{bias},'}, \pi}(\mathcal{H}_\tau)} \right) \right]
$$

$$
= \mathbb{E}_{\boldsymbol{\nu}^{\mathrm{bias}}, \pi}^{\mathcal{H}_\tau} \left[ \log\left( \frac{p_{\boldsymbol{\nu}^{\mathrm{bias}}, \pi}(U_0^{\mathrm{a}}) \prod_{t \in [\tau]} p_{\boldsymbol{\nu}^{\mathrm{bias}}, \pi}(A_t \mid \mathcal{H}_{t-1}) p_{\boldsymbol{\nu}^{\mathrm{bias}}, \pi}(Y_t \mid \mathcal{H}_{t-1}, A_t) p_{\boldsymbol{\nu}^{\mathrm{bias}}, \pi}(U_t^{\mathrm{a}} \mid \mathcal{H}_{t-1}, A_t, Y_t)}{p_{\boldsymbol{\nu}^{\mathrm{bias},'}, \pi}(U_0^{\mathrm{a}}) \prod_{t \in [\tau]} p_{\boldsymbol{\nu}^{\mathrm{bias},'}, \pi}(A_t \mid \mathcal{H}_{t-1}) p_{\boldsymbol{\nu}^{\mathrm{bias},'}, \pi}(Y_t \mid \mathcal{H}_{t-1}, A_t) p_{\boldsymbol{\nu}^{\mathrm{bias},'}, \pi}(U_t^{\mathrm{a}} \mid \mathcal{H}_{t-1}, A_t, Y_t)} \right) \right].
$$

First, notice that $U_t^{\mathrm{a}}$ is sampled i.i.d Uniform $([0,1])$ in both environments, so these terms in the above expression cancel. Further, by definition of the bandit policy $\pi$, given a fixed observation history $\mathcal{H}_{t-1}$, the action $A_t$ in environment $\boldsymbol{\nu}^{\mathrm{bias}}$ and $\boldsymbol{\nu}^{\mathrm{bias},'}$ is the same (since $A_t = \pi_t(\mathcal{H}_{t-1})$). That is, $p_{\boldsymbol{\nu}^{\mathrm{bias}},\pi}(A_t \mid \mathcal{H}_{t-1}) = p_{\boldsymbol{\nu}^{\mathrm{bias},'},\pi}(A_t \mid \mathcal{H}_{t-1})$ for all $t, \mathcal{H}_{t-1}, A_t$. Therefore, the above expression simplifies to:

$$
\begin{aligned}
D_{\mathrm{KL}}(\mathrm{Pr}_{\boldsymbol{\nu}^{\mathrm{bias}},\pi}^{\mathcal{H}_\tau} \| \mathrm{Pr}_{\boldsymbol{\nu}^{\mathrm{bias},'},\pi}^{\mathcal{H}_\tau}) &= \mathbb{E}_{\boldsymbol{\nu}^{\mathrm{bias}},\pi}^{\mathcal{H}_\tau} \left[ \log \left( \frac{\prod_{t\in[\tau]} p_{\boldsymbol{\nu}^{\mathrm{bias}},\pi}(Y_t \mid \mathcal{H}_{t-1}, A_t)}{\prod_{t\in[\tau]} p_{\boldsymbol{\nu}^{\mathrm{bias},'},\pi}(Y_t \mid \mathcal{H}_{t-1}, A_t)} \right) \right] \\
&= \mathbb{E}_{\boldsymbol{\nu}^{\mathrm{bias}},\pi}^{\mathcal{H}_\tau} \left[ \sum_{t\in[\tau]} \log \left( \frac{p_{\boldsymbol{\nu}^{\mathrm{bias}},\pi}(Y_t \mid \mathcal{H}_{t-1}, A_t)}{p_{\boldsymbol{\nu}^{\mathrm{bias},'},\pi}(Y_t \mid \mathcal{H}_{t-1}, A_t)} \right) \right].
\end{aligned}
$$

Now, since $\tau \leq n$ is a stopping time w.r.t. $\mathcal{F}_t$, the event $\{\tau \geq t\} \in \mathcal{F}_{t-1}$. Further, $A_t = \pi_t(\mathcal{H}_{t-1})$ is $\mathcal{F}_{t-1}$-measurable since $\mathcal{F}_{t-1} = \sigma\{\mathcal{H}_{t-1}\}$. Thus, applying the tower rule of expectations and the fact that $\sum_{i\in[K]} \mathbb{1}\{A_t = i\} = 1$:

$$
\begin{aligned}
D_{\mathrm{KL}}(\mathrm{Pr}_{\boldsymbol{\nu}^{\mathrm{bias}},\pi}^{\mathcal{H}_\tau} \| \mathrm{Pr}_{\boldsymbol{\nu}^{\mathrm{bias},'},\pi}^{\mathcal{H}_\tau}) &= \sum_{t\in[n]} \mathbb{E}_{\boldsymbol{\nu}^{\mathrm{bias}},\pi}^{\mathcal{H}_\tau} \left[ \log \left( \frac{p_{\boldsymbol{\nu}^{\mathrm{bias}},\pi}(Y_t \mid \mathcal{H}_{t-1}, A_t)}{p_{\boldsymbol{\nu}^{\mathrm{bias},'},\pi}(Y_t \mid \mathcal{H}_{t-1}, A_t)} \right) \mathbb{1}\{\tau \geq t\} \right] \\
&= \sum_{t\in[n]} \sum_{i\in[K]} \mathbb{E}_{\boldsymbol{\nu}^{\mathrm{bias}},\pi}^{\mathcal{H}_\tau} \left[ \log \left( \frac{p_{\boldsymbol{\nu}^{\mathrm{bias}},\pi}(Y_t \mid \mathcal{H}_{t-1}, A_t = i)}{p_{\boldsymbol{\nu}^{\mathrm{bias},'},\pi}(Y_t \mid \mathcal{H}_{t-1}, A_t = i)} \right) \mathbb{1}\{\tau \geq t, A_t = i\} \right] \\
&= \sum_{t\in[n]} \sum_{i\in[K]} \mathbb{E}_{\boldsymbol{\nu}^{\mathrm{bias}},\pi}^{\mathcal{H}_\tau} \left[ \mathbb{E}_{\boldsymbol{\nu}^{\mathrm{bias}},\pi}^{\mathcal{H}_\tau} \left[ \log \left( \frac{p_{\boldsymbol{\nu}^{\mathrm{bias}},\pi}(Y_t \mid \mathcal{H}_{t-1}, i)}{p_{\boldsymbol{\nu}^{\mathrm{bias},'},\pi}(Y_t \mid \mathcal{H}_{t-1}, i)} \right) \mid \mathcal{F}_{t-1} \right] \mathbb{1}\{\tau \geq t, A_t = i\} \right].
\end{aligned}
$$

Finally, observing that[9]

$$
\begin{aligned}
\mathbb{E}_{\boldsymbol{\nu}^{\mathrm{bias}},\pi}^{\mathcal{H}_\tau} \left[ \log \left( \frac{p_{\boldsymbol{\nu}^{\mathrm{bias}},\pi}(Y_t \mid \mathcal{H}_{t-1}, i)}{p_{\boldsymbol{\nu}^{\mathrm{bias},'},\pi}(Y_t \mid \mathcal{H}_{t-1}, i)} \right) \mid \mathcal{F}_{t-1} \right] &= D_{\mathrm{KL}}(\nu_{i,t}^{\mathrm{bias}} \| \nu_{i,t}^{\mathrm{bias},'}) \\
&= D_{\mathrm{KL}}(\mathcal{N}(\mu_i W_i(t), 1) \| \mathcal{N}(\mu_i' W_i(t), 1)),
\end{aligned}
$$

and, recalling that:

$$
D_{\mathrm{KL}}(\mathcal{N}(x\mu, 1) \| \mathcal{N}(x\mu', 1)) = \frac{(x\mu - x\mu')^2}{2} = x^2 D_{\mathrm{KL}}(\mathcal{N}(\mu, 1) \| \mathcal{N}(\mu', 1)) \ \forall x, \mu, \mu' \in \mathbb{R},
$$

we conclude that:

$$
\begin{aligned}
D_{\mathrm{KL}}(\mathrm{Pr}_{\boldsymbol{\nu}^{\mathrm{bias}},\pi}^{\mathcal{H}_\tau} \| \mathrm{Pr}_{\boldsymbol{\nu}^{\mathrm{bias},'},\pi}^{\mathcal{H}_\tau}) &= \sum_{t\in[n]} \sum_{i\in[K]} D_{\mathrm{KL}}(\nu_i \| \nu_i') \mathbb{E}_{\boldsymbol{\nu}^{\mathrm{bias}},\pi}^{\mathcal{H}_\tau} \left[ W_i(t)^2 \mathbb{1}\{\tau \geq t, A_t = i\} \right] \\
&= \sum_{i\in[K]} D_{\mathrm{KL}}(\nu_i \| \nu_i') \mathbb{E}_{\boldsymbol{\nu}^{\mathrm{bias}},\pi}^{\mathcal{H}_\tau} \left[ \sum_{t\in[\tau]} W_i(t)^2 \mathbb{1}\{A_t = i\} \right].
\end{aligned}
$$

Together with (39), this establishes that, for any $\mathcal{F}_\tau$-measurable $Z$,

$$
\sum_{i\in[K]} D_{\mathrm{KL}}(\nu_i \| \nu_i') \mathop{\mathbb{E}}_{\boldsymbol{\nu}^{\mathrm{bias}},\pi} \left[ \sum_{t\in[\tau]} W_i(t)^2 \mathbb{1}\{A_t = i\} \right] \geq \mathrm{kl} \left( \mathop{\mathbb{E}}_{\boldsymbol{\nu}^{\mathrm{bias}},\pi} [Z], \mathop{\mathbb{E}}_{\boldsymbol{\nu}^{\mathrm{bias},'},\pi} [Z] \right).
$$

In particular, whenever $\boldsymbol{\nu}' = \boldsymbol{\nu}^{\mathrm{bias},(i)}$, where $\nu_j^{(i)} = \nu_j$ for all $j \neq i$, then since $D_{\mathrm{KL}}(\nu_j \| \nu_j^{(i)}) = 0$, the above simplifies to:

$$
D_{\mathrm{KL}}(\nu_i \| \nu_i') \mathop{\mathbb{E}}_{\boldsymbol{\nu}^{\mathrm{bias}},\pi} \left[ \sum_{t\in[\tau]} W_i(t)^2 \mathbb{1}\{A_t = i\} \right] \geq \mathrm{kl} \left( \mathop{\mathbb{E}}_{\boldsymbol{\nu}^{\mathrm{bias}},\pi} [Z], \mathop{\mathbb{E}}_{\boldsymbol{\nu}^{\mathrm{bias},'},\pi} [Z] \right),
$$

as claimed. $\qquad\square$

---

[9]Note that, in the equation below, we adopt a slight abuse of notation. Indeed, the KL-divergence between $\nu_{i,t}^{\mathrm{bias}}$ and $\nu_{i,t}^{\mathrm{bias},'}$ is taken *conditioned* on the filtration $\mathcal{F}_{t-1}$, as the LHS of the expression makes clear.

Notice that (30) gives the standard bandit lower bound. This shows the intuitive fact that our biased setting is at least as hard as the unbiased stochastic bandit setting. However, to obtain our stronger lower bound, we will exploit the fact that $\frac{T_i^{\text{bias}}(t-1)}{t^{\text{bias}}-1}$ cannot be close to 1 for all arms simultaneously. Indeed, a pigeonholing argument shows that:

**Lemma 5.7** (Size of the small bias set). *Consider a bandit policy $\pi$ interacting in an environment $\nu$. Let us denote, for any $t \in [n]$ and $\beta > 1$,*

$$S_t(\beta) = \left\{ i \in [K] : T_i^{\text{bias}}(t)/t^{\text{bias}} \leq \beta/K \right\}.$$

*Then, $|S_t(\beta)| \geq (1 - 1/\beta)K$.*

*Proof.* By definition of $T_i(t)$ and the set $S_t(\beta)$:

$$t = \sum_{i \in [K]} T_i(t) \geq \sum_{i \in S_t(\beta)^c} T_i(t) = \sum_{i \in S_t(\beta)^c} T_i^{\text{bias}}(t) - T_i^0 > \frac{\beta t^{\text{bias}}}{K} |S_t(\beta)^c| - \sum_{i \in S_t(\beta)^c} T_i^0.$$

We may rearrange the above inequality to conclude that:

$$|S_t(\beta)^c| < \frac{K(t + \sum_{i \in S_t(\beta)^c} T_i^0)}{\beta t^{\text{bias}}} \leq \frac{K}{\beta},$$

Since $|S_t(\beta)^c| = K - |S_t(\beta)|$, we obtain the claimed result. $\qquad\square$

Notice that, while Lemma 5.7 guarantees that a constant fraction of arms have a ratio smaller than $\mathcal{O}(1/K)$, it says nothing about how this set evolves over time. To make use of (31), we need to identify a subset of arms which have a small ratio for many time steps. In the next result, we show that this is, in fact, possible:

**Lemma D.6** (A small bias set which is stable over time; formal (generalized) statement of Lemma 5.8). *Let $\pi$ be any bandit policy interacting in an environment $\nu^{\text{bias}}$ with initial biases $\left\{ T_i^0 \right\}_{i \in [K]}$ and suboptimality gaps $\Delta_i$. For constants $c, c' \geq 0$, let $\mathcal{A}_{c,c'}$ be any subset of suboptimal arms such that*

$$\mathcal{A}_{c,c'} = \left\{ i, j \in [K] : \log\left( \frac{\Delta_i}{\Delta_j} \right) \leq c \log(K)^{c'} \right\}$$

*Let $S_t(\beta)$ (for $\beta > 1$) be the set from Lemma 5.7. Fix any $n_0 \in [n]$, and define*

$$\widetilde{S}_{n_0}(\beta; c, c') := S_{n_0}(\beta) \cap \mathcal{A}_{c,c'}.$$

*Fix any $\frac{K}{|\mathcal{A}_{c,c'}|} < \beta < \beta'$. Then, for any $\alpha \in (0, 1)$, there exists a set of arms $B := B(n_0, c, c', \beta, \beta', \alpha) \subseteq \widetilde{S}_{n_0}(\beta; c, c')$ such that $|B| = |\widetilde{S}_{n_0}(\beta; c, c')| - \alpha K$, and each arm $i \in B$ satisfies one of the following:*

***Case 1***. *$\frac{T_i^{\text{bias}}(t)}{t^{\text{bias}}} \leq \frac{\beta'}{K} \ \forall t \in [n_0, n)$. Let $\mathcal{B}_{1,i}$ denote the event that this case occurs.*
***Case 2***. *$T_i(n_0, n) \geq \frac{(\beta' - \beta)(n_0 + \sum_{i \in [K]} T_i^0)}{K} \exp(\alpha \beta')$. Let $\mathcal{B}_{2,i}$ denote the event that this case occurs.*

This proof, and subsequent ones, will rely on the following definition:

**Definition D.7** (The first large bias time). Let $\frac{K}{|\mathcal{A}_{c,c'}|} < \beta < \beta'$, $n_0 \in [n]$, and $S_{n_0}(\beta)$ be as in Lemma 5.7. For each $i \in [K]$, we define the following stopping time:

$$\tau_i(\beta') = \min\left\{ t \geq n_0 : \frac{T_i^{\text{bias}}(t)}{t^{\text{bias}}} > \frac{\beta'}{K} \text{ or } t = n \right\}.$$

When $\beta'$ is clear from context, we will sometimes abuse notation by writing this time as $\tau_i$.

Notice that $\tau_i$ is adapted to the filtration $\mathcal{F}_t$. Using this notation, we will first state a stronger bound than Lemma D.6, and use it to establish that claim.

**Lemma D.8.** *Let $i_1, \ldots, i_{|\widetilde{S}_{n_0}(\beta;c,c')|}$ be an ordering on the arms in $\widetilde{S}_{n_0}(\beta;c,c')$ (the set from Lemma D.6) satisfying $\tau_{i_j}(\beta') \leq \tau_{i_{j+1}}(\beta')$ for each $j < |\widetilde{S}_{n_0}(\beta;c,c')|$. Then, each arm $i_j \in \widetilde{S}_{n_0}(\beta;c,c')$ satisfies one of the following:*

$$\tau_{i_j}(\beta') = n \quad or \quad T_{i_j}(n_0, \tau_{i_j}(\beta')) \geq \frac{(\beta' - \beta)(t_0^{\mathrm{bias}} + n_0)}{K} \exp\left(\frac{\beta' j}{K}\right),$$

*where $\tau_i(\beta')$ is the stopping time from Definition D.7 and $T_i(a,b) = \sum_{t=a+1}^{b} \mathbb{1}\{A_t = i\} = T_i(b) - T_i(a)$ is the number of times policy $\pi$ plays arm $i$ during $t \in (a, b]$.*

We first show how Lemma D.6 follows directly from Lemma D.8:

*Proof of Lemma D.6.* Consider $i_j \in \widetilde{S}_{n_0}(\beta;c,c')$ for $j \geq \alpha K$. By Lemma D.8, there are two cases: in the first, we have that:

$$T_{i_j}(n_0, n) \geq T_{i_j}(n_0, \tau_{i_j}) \geq \frac{(\beta' - \beta)(t_0^{\mathrm{bias}} + n_0)}{K} \exp\left(\frac{\beta' j}{K}\right)$$
$$\geq \frac{(\beta' - \beta)(t_0^{\mathrm{bias}} + n_0)}{K} \exp\left(\alpha \beta'\right),$$

which is the first case of Lemma 5.8. Otherwise, Lemma D.8 implies that $\tau_{i_j} = n$. Recalling Definition D.7, this implies:

$$\frac{T_i^{\mathrm{bias}}(t-1)}{t^{\mathrm{bias}} - 1} \leq \frac{\beta'}{K} \quad \forall t \in (n_0, n],$$

which is the second condition of Lemma 5.8. Thus, taking

$$B = \{i_j \in S_{n_0}(\beta) : \alpha K < j \leq |S_{n_0}(\beta)|\},$$

and noticing that $|B| = |S_{n_0}(\beta)| - \alpha K$, we obtain the claimed result. $\qquad\square$

We now establish Lemma D.8.

*Proof of Lemma D.8.* Let us assume, for the notational simplicity of this proof only, that $\widetilde{S}_{n_0}(\beta;c,c') = \left\{1, 2, \ldots, |\widetilde{S}_{n_0}(\beta;c,c')|\right\}$, and that each arm $i \in \widetilde{S}_{n_0}(\beta;c,c')$ is removed from this set before $i+1$ (i.e., $\tau_i \leq \tau_{i+1}$ for all $i, i+1 \in \widetilde{S}_{n_0}(\beta;c,c')$). Indeed, this is always true up to a relabelling of the arm indices.

Let us denote, for any $i, j \in [K]$:

$$x_{i,j} = \sum_{t=n_0+1}^{\tau_j} \mathbb{1}\{A_t = i\}.$$

Notice that $x_{i,i} = T_i(n_0, \tau_i)$. Intuitively, $x_{i,j}$ represents the number of times arm $i$ is played on the interval $(n_0, \tau_j]$, i.e., before arm $j$ has bias larger than $\beta'/K$ for the $r$th time after $n_0$.

Fix any arm $i \in \widetilde{S}_{n_0}(\beta;c,c')$, and let us assume that $\tau_i < n$ (since otherwise, the claim follows trivially). Then, by definition of the $x_{i,j}$s, we know that

$$\tau_i - n_0 = \sum_{t=n_0+1}^{\tau_i} \sum_{j \in [k]} \mathbb{1}\{A_t = j\} = \sum_{j \in [k]} x_{j,i}.$$

Further, since $\tau_j \leq \tau_i$ for every $j \leq i \in \widetilde{S}_{n_0}(\beta;c,c')$ (by our assumed ordering of the arms):

$$\sum_{j \leq i} x_{j,i} = \sum_{j \leq i} \sum_{t=n_0+1}^{\tau_i} \mathbb{1}\{A_t = j\} \geq \sum_{j \leq i} \sum_{t=n_0+1}^{\tau_j} \mathbb{1}\{A_t = j\} = \sum_{j \leq i} x_{j,j}.$$

Thus, by definition of $\tau_i$, and since $x_{j,i} \geq 0$ for all $i, j$, we may use the above bounds to conclude:

$$\frac{\beta'}{K} < \frac{T_i^{\text{bias}}(\tau_i)}{t_0^{\text{bias}} + \tau_i} = \frac{T_i^{\text{bias}}(n_0) + x_{i,i}}{t_0^{\text{bias}} + n_0 + \sum_{j \in [K]} x_{j,i}} \leq \frac{T_i^{\text{bias}}(n_0) + x_{i,i}}{t_0^{\text{bias}} + n_0 + \sum_{j \leq i} x_{j,i}} \leq \frac{T_i^{\text{bias}}(n_0) + x_{i,i}}{t_0^{\text{bias}} + n_0 + \sum_{j \leq i} x_{j,j}}$$

Rearranging this expression, we have that:

$$\left(1 - \frac{\beta'}{K}\right) x_{i,i} > \frac{\beta'(t_0^{\text{bias}} + n_0)}{K} \left(1 - \frac{K}{\beta'} \frac{T_i^{\text{bias}}(n_0)}{t_0^{\text{bias}} + n_0} + \frac{\sum_{j<i} x_{j,j}}{t_0^{\text{bias}} + n_0}\right).$$

Thus, as long as $\beta' < K$, then, since $i \in \widetilde{S}_{n_0}(\beta; c, c')$ (i.e., $\frac{T_i^{\text{bias}}(n_0)}{t_0^{\text{bias}} + n_0} \leq \frac{\beta}{K}$),

$$x_{i,i} > \frac{\beta'(t_0^{\text{bias}} + n_0)}{K - \beta'} \left(1 - \frac{\beta}{\beta'} + \frac{\sum_{j<i} x_{j,j}}{t_0^{\text{bias}} + n_0}\right) = \frac{(\beta' - \beta)(t_0^{\text{bias}} + n_0)}{K - \beta'} + \frac{\beta'}{K - \beta'} \sum_{j<i} x_{j,j}.$$

It is a technical exercise (see Lemma D.9) to show that the above bound implies that:

$$x_{i,i} > \frac{(\beta' - \beta)(t_0^{\text{bias}} + n_0)}{K} \exp\left(\frac{\beta' i}{K}\right),$$

which is the claimed bound. $\qquad\square$

In the following, we establish a technical result used in the proof of Lemma D.8.

**Lemma D.9.** *Let $1 < \beta < \beta' < K$, and let $t > 0$. Suppose that:*

$$x_{i,i} > \frac{(\beta' - \beta)t}{K - \beta'} + \frac{\beta'}{K - \beta'} \sum_{j<i} x_{j,j} \quad \forall i \in [K]. \tag{40}$$

*Then, for every $i \in [K]$, we have that:*

$$x_{i,i} > \frac{(\beta' - \beta)t}{K} \exp\left(\frac{\beta' i}{K}\right). \tag{41}$$

*Proof.* We first prove that

$$x_{i,i} > \frac{(\beta' - \beta)t}{K - \beta} \sum_{j=0}^{i-1} \binom{i-1}{j} \left(\frac{\beta'}{K - \beta'}\right)^j. \tag{42}$$

(41) follows from (42) by first noting that, by the Binomial identity, the summation in (42) can be written as

$$\sum_{j=0}^{i-1} \binom{i-1}{j} \left(\frac{\beta'}{K - \beta'}\right)^j = \left(1 + \frac{\beta'}{K - \beta'}\right)^{i-1} = \frac{K - \beta'}{K} \left(1 + \frac{\beta'}{K - \beta'}\right)^i \geq \frac{K - \beta'}{K} \exp\left(\frac{\beta' i}{K}\right).$$

where, in the last line, we used the fact that $1 + x \geq \exp\left(\frac{x}{1+x}\right)$ for any $x > -1$.

We thus focus on proving (42). The proof proceeds by induction on $i$. The case of $i = 1$ is immediate from (40). Assuming the claim holds for $1, \ldots, i$, we have that, using (40):

$$\begin{aligned}
x_{i+1,i+1} &> \frac{(\beta' - \beta)t}{K - \beta'} + \frac{\beta'}{K - \beta'} \sum_{j=1}^{i} x_{j,j} \\
&> \frac{(\beta' - \beta)t}{K - \beta'} + \sum_{j=1}^{i} \frac{(\beta' - \beta)t}{K - \beta} \sum_{\ell=0}^{j-1} \binom{j-1}{\ell} \left(\frac{\beta'}{K - \beta}\right)^{\ell+1} \\
&= \frac{(\beta' - \beta)t}{K - \beta'} \left(1 + \sum_{\ell=0}^{i-1} \left(\frac{\beta'}{K - \beta'}\right)^{\ell+1} \sum_{j=\ell+1}^{i} \binom{j-1}{\ell}\right),
\end{aligned}$$

where in the last step, we exchange the order of summation. Then, by the hockey-stick identity:

$$\sum_{j=\ell+1}^{i} \binom{j-1}{\ell} = \binom{i}{\ell+1},$$

we obtain:

$$x_{i+1,i+1} > \frac{(\beta'-\beta)t}{K-\beta'}\left(1 + \sum_{\ell=0}^{i-1}\left(\frac{\beta'}{K-\beta'}\right)^{\ell+1}\binom{i}{\ell+1}\right)$$

$$= \frac{(\beta'-\beta)t}{K-\beta'}\sum_{\ell=0}^{i}\left(\frac{\beta'}{K-\beta'}\right)^{\ell}\binom{i}{\ell}. \qquad \square$$

To complete the proof, it will be useful to "derandomize" the set $B$, which is accomplished by another pigeonholing argument.

**Lemma D.10** (Derandomizing the set from Lemma D.6). *For $\alpha \in (0,1)$, let $B := B(n_0, \beta, \beta', c, c', \alpha)$ and $\widetilde{S}_{n_0}(\beta; c, c')$ be the (random) sets from Lemma D.6 satisfying $|B| \geq |\widetilde{S}_{n_0}(\beta; c, c')| - \alpha K$ for some $\alpha \in (0,1)$. Then, there exists a deterministic set $B' := B'(n_0, \beta, \beta', c, c', \alpha) \subseteq [K]$ such that $|B'| \geq \frac{1}{1-\alpha/2}\left(\mathbb{E}_{\boldsymbol{\nu}^{\mathrm{bias}}, \pi}\left[|\widetilde{S}_{n_0}(\beta; c, c')|\right] - \frac{3\alpha K}{2}\right)$ and, for each $i \in B'$:*

$$\Pr_{\boldsymbol{\nu}^{\mathrm{bias}}, \pi}[\mathcal{B}_{1,i} \text{ or } \mathcal{B}_{2,i}] \geq \alpha/2,$$

*where $\mathcal{B}_{1,i}$ and $\mathcal{B}_{2,i}$ are the events defined in Lemma D.6.*

*Proof.* By Lemma D.6, there is a set $B \subseteq [K]$ such that:

$$\sum_{i \in [K]} \mathbb{1}\{\mathcal{B}_{1,i} \text{ or } \mathcal{B}_{2,i}\} \geq \sum_{i \in B} \mathbb{1}\{\mathcal{B}_{1,i} \text{ or } \mathcal{B}_{2,i}\} = \sum_{i \in B} 1 = |B|$$

Denote $B'$ as the (possibly empty) set of all arms $i$ satisfying $\Pr_{\boldsymbol{\nu}^{\mathrm{bias}}, \pi}[\mathcal{B}_{1,i} \text{ or } \mathcal{B}_{2,i}] \geq \alpha/2$. Then, taking expectations of the above, and using linearity of expectation, we obtain:

$$\mathbb{E}_{\boldsymbol{\nu}^{\mathrm{bias}}, \pi}[|B|] \leq \sum_{i \in B'}\Pr_{\boldsymbol{\nu}^{\mathrm{bias}}, \pi}[\mathcal{B}_{1,i} \text{ or } \mathcal{B}_{2,i}] + \sum_{i \in [K] \setminus B'}\Pr_{\boldsymbol{\nu}^{\mathrm{bias}}, \pi}[\mathcal{B}_{1,i} \text{ or } \mathcal{B}_{2,i}]$$

$$\leq \sum_{i \in B'} 1 + \sum_{i \in [K] \setminus B'} \alpha/2$$

$$= |B'| + \frac{\alpha}{2}(K - |B'|).$$

Rearranging the above, and using the fact that, by Lemma D.6, $|B| \geq |\widetilde{S}_{n_0}(\beta; c, c')| - \alpha K$, we conclude that $|B'| \geq \frac{1}{1-\alpha/2}\left(\mathbb{E}\left[|\widetilde{S}_{n_0}(\beta; c, c')|\right] - 3\alpha K/2\right)$, as claimed. $\qquad \square$

Now that we have derandomized the set $B$, we are almost ready to conclude our proof. First, we show the following:

**Lemma D.11** (Properties of the derandomized set from Lemma D.10; Restatement of Lemma 5.9). *Consider a policy $\pi$ interacting in environment $\boldsymbol{\nu}^{\mathrm{bias}}$ with initial biases $\{T_i^0\}_{i \in [K]}$. Let $\mathcal{A}_{c,c'}$ be the set from Lemma D.6 for some $c, c' \geq 0$. Fix any $K/|\mathcal{A}_{c,c'}| < \beta < \beta'$, $n_0 \in [n]$, $\alpha \in (0,1)$. Let $B' := B'(n_0, \beta, \beta', c, c', \alpha)$ be the set from Lemma D.10 and $\widetilde{S}_{n_0}(\beta; c, c')$ be the set from Lemma D.6 satisfying $|B'| \geq \frac{1}{1-\alpha/2}(\mathbb{E}\left[|\widetilde{S}_{n_0}(\beta; c, c')|\right] - 3\alpha K/2)$. Then, for every $i \in B'$, one of the following holds:*

***Case 1'.*** $\mathbb{E}_{\boldsymbol{\nu}^{\mathrm{bias}}, \pi}[\tau_i(\beta')] \geq \frac{\alpha n}{4}$

***Case 2'.*** $\mathbb{E}_{\boldsymbol{\nu}^{\mathrm{bias}}, \pi}[T_i(n_0, n)] \geq \frac{\alpha(\beta'-\beta)(n_0 + \sum_{i \in [K]} T_i^0)}{4K}\exp(\alpha\beta')$

*where $\tau_i(\beta')$ is the stopping time from Definition D.7, and $T_i(n_0, n) = \sum_{t=n_0+1}^{n} \mathbb{1}\{A_t = i\} = T_i(n) - T_i(n_0)$ is the number of times policy $\pi$ plays arm $i$ during $t \in (n_0, n]$.*

*Proof.* Recall that, by Lemma D.10,

$$2 \max \left\{ \Pr_{\boldsymbol{\nu}^{\mathrm{bias}}, \pi} [\mathcal{B}_{1,i}], \Pr_{\boldsymbol{\nu}^{\mathrm{bias}}, \pi} [\mathcal{B}_{2,i}] \right\} \geq \Pr_{\boldsymbol{\nu}^{\mathrm{bias}}, \pi} [\mathcal{B}_{1,i}] + \Pr_{\boldsymbol{\nu}^{\mathrm{bias}}, \pi} [\mathcal{B}_{2,i}] \geq \Pr_{\boldsymbol{\nu}^{\mathrm{bias}}, \pi} [\mathcal{B}_{1,i} \text{ or } \mathcal{B}_{2,i}] \geq \frac{\alpha}{2}.$$

Now, if $\Pr_{\boldsymbol{\nu}^{\mathrm{bias}}, \pi} [\mathcal{B}_{1,i}] \geq \frac{\alpha}{4}$, then, recalling that $\mathcal{B}_{1,i}$ (the first event in Lemma D.6) implies $\tau_i = n$:

$$\mathbb{E}_{\boldsymbol{\nu}^{\mathrm{bias}}, \pi} [\tau_i] \geq \mathbb{E}_{\boldsymbol{\nu}^{\mathrm{bias}}, \pi} [\tau_i \mathbb{1}\{\mathcal{B}_{1,i}\}] = n \Pr [\mathcal{B}_{1,i}] \geq \frac{\alpha n}{4}.$$

Otherwise, recalling that $\mathcal{B}_{2,i}$ (the event from Lemma D.6) implies a lower bound on $T_i(n_0, n)$,

$$\mathbb{E}_{\boldsymbol{\nu}^{\mathrm{bias}}, \pi} [T_i(n_0, n)] \geq \mathbb{E}_{\boldsymbol{\nu}^{\mathrm{bias}}, \pi} [T_i(n_0, n) \mathbb{1}\{\mathcal{B}_{1,i}\}] \geq \frac{(\beta' - \beta)(t_0^{\mathrm{bias}} + n_0)}{K} \exp(\alpha\beta') \Pr [\mathcal{B}_{1,i}]$$

$$\geq \frac{\alpha(\beta' - \beta)(t_0^{\mathrm{bias}} + n_0)}{4K} \exp(\alpha\beta').$$

$\square$

We are now ready to prove Theorem D.1.

*Proof of Theorem D.1.* Recall the set $\mathcal{A}_{c,c'}$ from Lemma D.6. Let $\Delta_{\max}(\mathcal{A}_{c,c'}) = \max_{i \in \mathcal{A}_{c,c'}} \Delta_i$ and $\Delta_{\min}(\mathcal{A}_{c,c'}) = \min_{i \in \mathcal{A}_{c,c'}: \Delta_i > 0} \Delta_i$. Notice that, by definition of $\mathcal{A}_{c,c'}$, $\log \left( \Delta_{\max}(\mathcal{A}_{c,c'}) / \Delta_{\min}(\mathcal{A}_{c,c'}) \right) \leq c \log(K)^{c'}$ Let us take:

$$\beta = \frac{2K}{|\mathcal{A}_{c,c'}|} \quad \text{and} \quad \alpha = \frac{|\mathcal{A}_{c,c'}|}{6K} \quad \text{and} \quad \beta' = \frac{1}{\alpha} \left( 2 \log \left( \frac{\Delta_{\max}(\mathcal{A}_{c,c'})}{\Delta_{\min}(\mathcal{A}_{c,c'})} \right) + \log(4K/\alpha) - 2 \log \left( f \left( \frac{\beta}{K} \right) \right) \right),$$

$$\text{and} \quad n_0 = \frac{\alpha(1 - a) \log(n)}{4(1 + \varepsilon)^2 \Delta_{\max}(\mathcal{A}_{c,c'})^2},$$

where $a$ is the exponent in the consistency condition from Definition 5.1, and $\varepsilon \in (0, (1 - \mu_{\max})/\Delta_{\max}]$. Taking $\gamma \in (0, 1)$ satisfying $|\mathcal{A}_{c,c'}| = \gamma K$, we have that the above parameter choices satisfy:

$$\frac{K}{|\mathcal{A}_{c,c'}|} = \frac{1}{\gamma} < \frac{2}{\gamma} = \beta < \frac{6}{\gamma} < \beta',$$

and $\alpha \in (0, 1)$, so our choices of parameters satisfy the conditions in the results of this section. By Lemma 5.7 and definition of $\widetilde{S}_{n_0}(\beta; c, c') = S_{n_0}(\beta) \cap \mathcal{A}_{c,c'}$ (from Lemma D.6), we know that

$$|\widetilde{S}_{n_0}(\beta; c, c')| \geq |S_{n_0(\beta)}| - |\mathcal{A}_{c,c'}^c| \geq (1 - 1/\beta)K - |[K] \setminus \mathcal{A}_{c,c'}| = \frac{|\mathcal{A}_{c,c'}|}{2},$$

where, in the last line, we used our choice of $\beta$. Recall the set $B' := B'(n_0, \beta, \beta', c, c', \alpha)$ be the set from Lemma D.10, where $|B'| \geq \frac{1}{1 - \alpha/2} \left( \mathbb{E}_{\boldsymbol{\nu}^{\mathrm{bias}}, \pi} \left[ |\widetilde{S}_{n_0}(\beta; c, c')| \right] - \frac{3\alpha K}{2} \right)$. Then, by the previous observation and our choice of $\alpha$,

$$|B'| \geq \frac{1}{1 - \alpha/2} \left( \mathbb{E}_{\boldsymbol{\nu}^{\mathrm{bias}}, \pi} \left[ |\widetilde{S}_{n_0}(\beta; c, c')| \right] - \frac{3\alpha K}{2} \right)$$

$$\geq \frac{1}{1 - \alpha/2} \left( \frac{1}{2} |\mathcal{A}_{c,c'}| - \frac{3\alpha K}{2} \right)$$

$$= \frac{3K}{12K - |\mathcal{A}_{c,c'}|} |\mathcal{A}_{c,c'}|$$

$$\geq \frac{|\mathcal{A}_{c,c'}|}{4}.$$

Fix any arm $i \in B' \subseteq \mathcal{A}_{c,c'}$, and construct the alternative environment $\boldsymbol{\nu}^{(i)}$ as in Claim D.4. We use Lemma D.11 to lower bound $\mathbb{E}_{\boldsymbol{\nu}^{\text{bias}},\pi}[T_i(n)]$. There are two cases.

**Case 1':** $\mathbb{E}_{\boldsymbol{\nu}^{\text{bias}},\pi}[\tau_i(\beta')] \geq \alpha n/4$. Recall that, by definition of $\tau_i(\beta')$ in Definition D.7, $\frac{T_i^{\text{bias}}(t-1)}{t^{\text{bias}}-1} \leq \beta'/K$ for all $t \in (n_0, \tau_i(\beta')]$. Thus, by (31) in Claim D.4 (taking $c \leftarrow \alpha/4$ in the notation of that equation), and since $n_0 = \frac{\alpha(1-a)\log(n)}{4(1+\varepsilon)^2\Delta_{\max}(\mathcal{A}_{c,c'})^2}$, and since $\Delta_{\max}(\mathcal{A}_{c,c'}) \geq \Delta_i$ by definition of $i \in \mathcal{A}_{c,c'}$, we have that:

$$\mathbb{E}_{\boldsymbol{\nu}^{\text{bias}},\pi}[T_i(n)] \geq f\left(\frac{\beta'}{K}\right)^{-2} \frac{\alpha(1-a)}{2(1+\varepsilon)^2\Delta_i^2}\left(\log(n) - \frac{2(1+\varepsilon)^2\Delta_i^2}{\alpha(1-a)}n_0 - \mathcal{O}(1)\right)$$
$$\geq f\left(\frac{\beta'}{K}\right)^{-2} \frac{\alpha(1-a)}{2(1+\varepsilon)^2\Delta_i^2}\left(\frac{\log(n)}{2} - \mathcal{O}(1)\right)$$
$$\geq f\left(\frac{\beta'}{K}\right)^{-2} \frac{\alpha(1-a-o(1))\log(n)}{4(1+\varepsilon)^2\Delta_i^2}.$$

**Case 2':** Otherwise, by our choice of $\beta, \beta'$, and $n_0$, we directly conclude, using the lower bound from Lemma D.11, that

$$\mathbb{E}_{\boldsymbol{\nu}^{\text{bias}},\pi}[T_i(n)] \geq \frac{\alpha(\beta'-\beta)(t_0^{\text{bias}}+n_0)}{4K}\exp(\alpha\beta')$$
$$\geq f\left(\frac{\beta}{K}\right)^{-2}\left(\frac{\Delta_{\max}(\mathcal{A}_{c,c'})}{\Delta_{\min}(\mathcal{A}_{c,c'})}\right)^2 n_0$$
$$\geq f\left(\frac{\beta}{K}\right)^{-2}\frac{\alpha(1-a)\log(n)}{4(1+\varepsilon)^2\Delta_i^2}$$
$$\geq f\left(\frac{\beta'}{K}\right)^{-2}\frac{\alpha(1-a)\log(n)}{4(1+\varepsilon)^2\Delta_i^2},$$

where, in the second line, we used the fact that $\beta' - \beta > 1$ and $t_0^{\text{bias}} \geq 0$. In the penultimate line, we used the fact that $\Delta_{\min}(\mathcal{A}_{c,c'}) \leq \Delta_i$ for all $i \in \mathcal{A}_{c,c'}$. In the final line, we used the fact that $f(x)$ is nondecreasing in $x$ by definition. Observe that we obtain, up to vanishing factors, the same bound in Cases 1' and 2'. Thus, repeating this argument for each arm $i \in B'$, using the bound in (30) from Claim D.4 to bound the remaining arms $i \notin B'$, and plugging in the resulting bounds to the regret decomposition (1), we obtain the claimed regret lower bound. $\qquad\square$

# E. Linear regret of UCB when biased structure is ignored

**Theorem E.1.** *Consider the UCB policy, which first selects each arm once, then, for each $n \geq t > K$, computes a upper confidence estimate for each arm $i \in [K]$:*

$$\mathsf{UCB}_{i,t} = \widehat{\mu}_{i,t-1} + \sqrt{\frac{2\log(t)}{T_i(t-1)}} \quad \text{where} \quad \widehat{\mu}_{i,t-1} = \frac{\sum_{s\in[t-1]} Y_t \mathbb{1}\{A_t = i\}}{T_i(t-1)}.$$

*then selects an arm $A_t$ as follows:*

$$A_t = \begin{cases} t & \text{if } t \in [K] \\ \arg\max_{i\in[K]} \mathsf{UCB}_{i,t} & \text{otherwise.} \end{cases} \tag{UCB}$$

*Then, there is a 2-armed Bernoulli bandit instance $\boldsymbol{\nu}^{\text{bias}}$ under bias model (2) with unbiased means $\mu_1 = .9 > .8 = \mu_2$ and arm biases $T_1^0 = 10$ and $T_2^0 = 16216^{1.5}$ such that, for any $n \geq 16207$, (UCB) suffers linear regret: $R_{\boldsymbol{\nu},\mathsf{UCB}}(n) \geq .098n$.*

*Remark* E.2 (On bias initialization). Given the result of Theorem E.1, one might wonder if linear regret is inevitable, even when policy is initially biased towards the optimal arm 1 (i.e., $T_1^0 \geq T_2^0$). Whenever the initial biases are constant and the environment is Bernoulli, then (UCB) must suffer linear regret, and this follows straightforwardly from the proof of Theorem E.1. Indeed, with a constant probability, starting at any initial biases, (UCB) can pull the suboptimal arm 2 the majority of time steps over any constant-length time window (this can happen, e.g., if the samples from arm 2 are always

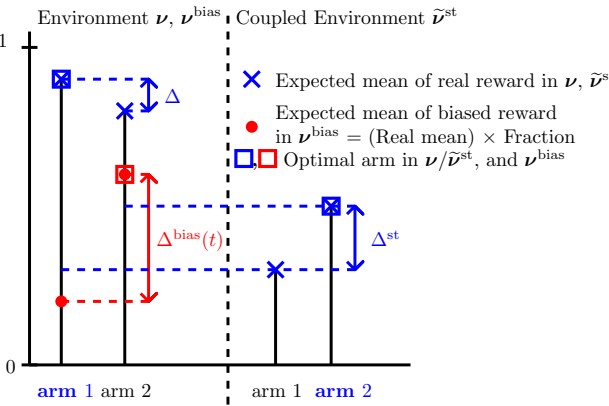

Figure 5: A depiction of the environment construction for Theorem E.1 at $t = 1$. The left side of the figure shows the means in the original environments $\boldsymbol{\nu}, \boldsymbol{\nu}^{\mathrm{bias}}$. The right side shows the "frozen" environment $\widetilde{\boldsymbol{\nu}}^{\mathrm{st}}$ used for our proof. Notice that the biased optimal arm is arm 2, not the true optimal arm 1. Further, $\widetilde{\Delta}^{\mathrm{st}} < \Delta^{\mathrm{bias}}(1)$.

1, and samples from 1 are always 0 in this window). After a sufficiently large constant initialization window $w$ when $T_1^{\mathrm{bias}}(w) \ll T_2^{\mathrm{bias}}(w)$, we can apply the same arguments as in the proof of Theorem E.1 to show that, in this case too, $R_{\boldsymbol{\nu}, \mathsf{UCB}}(n) = \Omega(n)$ when $n$ is sufficiently large.

### E.1. Environment construction

Fix a 2-armed Bernoulli bandit environment $\boldsymbol{\nu}$, where 1 denotes the optimal arm, and 2 denotes the suboptimal arm, and $\mu_1 > \mu_2$. Let $\boldsymbol{\nu}^{\mathrm{bias}}$ be the associated biased Bernoulli environment, where we take the initial arm biases to be $T_1^0 < T_2^0$ (we will choose these parameters explicitly later in the proof). Denote the mean of arm $i$ in $\boldsymbol{\nu}^{\mathrm{bias}}$ at time $t$ as $\mu_i^{\mathrm{bias}}(t)$. Notice that, by construction, and recalling the notation $t_0^{\mathrm{bias}} = T_1^0 + T_2^0$, we have that:

$$\frac{T_1^0}{t_0^{\mathrm{bias}}} \mu_1 = \mu_1^{\mathrm{bias}}(1) < \mu_2^{\mathrm{bias}}(1) = \frac{T_2^0}{t_0^{\mathrm{bias}}} \mu_2,$$

so that, at time $t = 1$, the suboptimal arm 2 *appears* optimal.

Given this environment, we construct a static, unbiased Bernoulli bandit environment $\widetilde{\boldsymbol{\nu}}^{\mathrm{st}}$, with means denoted as $\widetilde{\mu}_i^{\mathrm{st}}$, such that:

$$\mu_1^{\mathrm{bias}}(1) \leq \mu_1 \frac{\widetilde{T}_1^0}{\widetilde{t}_0^{\mathrm{st}}} = \widetilde{\mu}_1^{\mathrm{st}} < \widetilde{\mu}_2^{\mathrm{st}} = \mu_2 \frac{T_2^0}{\widetilde{t}_0^{\mathrm{st}}} \leq \mu_2^{\mathrm{bias}}(1),$$

where $T_1^0 \leq \widetilde{T}_1^0 < \frac{\mu_2}{\mu_1} T_2^0$ and $\widetilde{t}_0^{\mathrm{st}} = \widetilde{T}_1^0 + T_2^0$. In particular, in $\widetilde{\boldsymbol{\nu}}^{\mathrm{st}}$ arm 2 (the suboptimal arm from $\boldsymbol{\nu}^{\mathrm{bias}}$) is optimal. Intuitively, since the "observable" suboptimality gap in $\boldsymbol{\nu}^{\mathrm{bias}}$ at time $t = 1$ is smaller than that of $\widetilde{\boldsymbol{\nu}}^{\mathrm{st}}$, UCB in the static environment should pull arm $b$ less often than in $\boldsymbol{\nu}^{\mathrm{bias}}$, at least as long as:

$$\mu_1^{\mathrm{bias}}(t) \leq \widetilde{\mu}_1^{\mathrm{st}} < \widetilde{\mu}_2^{\mathrm{st}} \leq \mu_2^{\mathrm{bias}}(t).$$

Refer to Figure 5 for a graphic depicting our environment construction.

### E.2. Proofs

Intuitively, since the "observable" suboptimality gap in $\boldsymbol{\nu}^{\mathrm{bias}}$ at time $t = 1$ is larger than that of $\widetilde{\boldsymbol{\nu}}^{\mathrm{st}}$, UCB in the static environment should pull arm 2 less often than in $\boldsymbol{\nu}^{\mathrm{bias}}$, at least as long as $\mu_1^{\mathrm{bias}}(t) \leq \widetilde{\mu}_1^{\mathrm{st}} < \widetilde{\mu}_2^{\mathrm{st}} \leq \mu_2^{\mathrm{bias}}(t)$. We define the event that condition holds for all times in $[t]$ as:

$$\mathcal{B}_t = \bigcap_{s \in [t]} \left\{ \mu_1^{\mathrm{bias}}(s) \leq \widetilde{\mu}_1^{\mathrm{st}} < \widetilde{\mu}_2^{\mathrm{st}} \leq \mu_2^{\mathrm{bias}}(s) \right\} \tag{43}$$

Our goal is to construct an coupling between these two environments such that, whenever $\mathcal{B}_t$ is true, then arm 1 is played more often in $\widetilde{\boldsymbol{\nu}}^{\mathrm{st}}$ than in $\boldsymbol{\nu}^{\mathrm{bias}}$. Towards this goal, we are able to show the following:

**Lemma E.3** (Stochastic dominance). *Let $\widetilde{T}_i(s)$ (resp., $T_i(i)$) denote the number of times arm $i$ was played in $\widetilde{\nu}^{\mathrm{st}}$ (resp., $\nu^{\mathrm{bias}}$) over $[s]$. Then, there exists a coupling between (UCB) in environments $\nu^{\mathrm{bias}}$ and $\widetilde{\nu}^{\mathrm{st}}$ such that, for any $t$, whenever $\mathcal{B}_t$ is true, then $\widetilde{T}_1(s) \geq T_1(s)$ (thus also $\widetilde{T}_2(s) \leq T_2(s)$) for all $s \leq t$.*

We will defer the proof of Lemma E.3 until later, and focus for now on consequences of this result. We first show it is possible to construct a sufficient event $\widehat{\mathcal{B}}_R$ for $\mathcal{B}_n$ to occur. This event depends only on the dynamics of (UCB) in the unbiased environment $\widetilde{\nu}^{\mathrm{st}}$, unlike the event $\mathcal{B}_n$, and thus it will be easier to analyze:

**Lemma E.4** (Simplifying the event $\mathcal{B}_t$). *Let $0 = t_0 < t_1 < \ldots < t_R < t_{R+1} = n$ be the sequence of times such that $t_1 = \widetilde{T}_1^0 - T_1^0 + 1$ and $t_{i+1} = t_i + 1 \, \forall i \in [R]$. Fix $\varepsilon = \frac{\widetilde{T}_1^0}{t_0^{\mathrm{st}}}$, and denote*

$$\widetilde{\mathcal{B}}_R = \bigcap_{r=0}^{R} \left\{ \widetilde{T}_1(t_r) \leq \varepsilon t_r \right\}. \tag{44}$$

*Then, under the same coupling from Lemma E.3, we have that $\widetilde{\mathcal{B}}_R \subseteq \mathcal{B}_n$.*

To make use of Lemma E.4, we recall a standard concentration bound for (UCB) from (Audibert et al., 2009):

**Theorem E.5** (Theorem 8, (Audibert et al., 2009)). *Consider the UCB algorithm (UCB) interacting with a 2-armed (unbiased) stochastic bandit instance $\nu = (\nu_1, \nu_2)$, such that the support of each $\nu_i$ is $[0, 1]$ and arm 2 is optimal. Then, for any $n \geq 1$ and $x \geq 1 + \frac{8 \log(n)}{\Delta_1^2}$,*

$$\Pr\left[ T_1(n) > x \right] \leq n^{-\frac{4x}{1 + \frac{8 \log(n)}{\Delta_1^2}} + 1} + \frac{x^{-3}}{3}.$$

Notice that Lemma E.4 allows us translate the condition $\mathcal{B}_n$ in the biased environment $\nu^{\mathrm{bias}}$ to a condition $\widetilde{\mathcal{B}}_R$ in the unbiased environment $\widetilde{\nu}^{\mathrm{st}}$. Using the fact that (UCB) is an *anytime* algorithm, we may apply Theorem E.5 for $n = t_1, t_2, \ldots, t_R$. That is,

$$\Pr\left[ \widetilde{T}_1(n) \leq \varepsilon n \text{ and } \mathcal{B}_n \right] \geq \Pr\left[ \widetilde{T}_1(n) \leq \varepsilon n \text{ and } \widetilde{\mathcal{B}}_R \right] \geq 1 - \sum_{r=1}^{R+1} \Pr\left[ \widetilde{T}_1(t_r) > \varepsilon t_r \right]$$

$$\geq 1 - \sum_{r=1}^{R+1} t_r^{-\frac{4 \varepsilon t_r}{1 + \frac{8 \log(t_r)}{(\Delta_1^{\mathrm{st}})^2}} + 1} + \frac{t_r^{-3}}{3 \varepsilon^3}. \tag{45}$$

Using the above results, we establish the following:

**Lemma E.6** (Explicit selection of environment parameters). *Suppose that the environment $\nu$ has means $\mu_1 = .9 > .8 = \mu_1$. Then, taking $\nu^{\mathrm{bias}}$ and $\widetilde{\nu}^{\mathrm{st}}$ as defined above with:*

$$T_1^0 = 10, \quad \widetilde{T}_1^0 = 16126, \quad \text{and} \quad T_2^0 = 16126^{1.5}.$$

*Then,*

$$\Pr\left[ T_2(n) \geq .99 n \right] \geq .99.$$

*Proof.* Notice first that Lemma E.3 implies that

$$\Pr\left[ T_2(n) \geq (1 - \varepsilon) n \right] = \Pr\left[ T_1(n) \leq \varepsilon n \right] \geq \Pr\left[ T_1(n) \leq \varepsilon n \text{ and } \mathcal{B}_n \right] \geq \Pr\left[ \widetilde{T}_1(n) \leq \varepsilon n \text{ and } \mathcal{B}_n \right].$$

Further, by combining Lemma E.4 with the above, we have that:

$$\Pr\left[ T_2(n) \geq (1 - \varepsilon) n \right] \geq \Pr\left[ \widetilde{T}_1(n) \leq \varepsilon n \text{ and } \widetilde{\mathcal{B}}_n \right] \geq 1 - \sum_{r=1}^{R+1} \Pr\left[ \widetilde{T}_1(t_r) > \varepsilon t_r \right].$$

Applying Theorem E.5, the above implies:

$$\Pr\left[T_2(n) \geq (1-\varepsilon)n\right] \geq 1 - \sum_{r=1}^{R+1} t_r^{-\frac{4\varepsilon t_r}{1 + \frac{8\log(t_r)}{(\widetilde{\Delta}_1^{\mathrm{st}})^2}}+1} + \frac{t_r^{-3}}{3\varepsilon^3}.$$

Therefore, let us choose $t_1$ is chosen to satisfy:

$$\frac{4\varepsilon t_1}{1 + \frac{8\log(t_1)}{(\widetilde{\Delta}_1^{\mathrm{st}})^2}} - 1 \geq 3 \quad \text{or, equivalently,} \quad \varepsilon t_1 \geq 1 + \frac{8\log(t_1)}{(\widetilde{\Delta}_1^{\mathrm{st}})^2}.$$

Noting that the function $f(t) = \varepsilon t - 1 - \frac{8\log(t)}{(\widetilde{\Delta}_1^{\mathrm{st}})^2}$ is non-increasing for $\varepsilon t \geq \frac{8}{(\widetilde{\Delta}_1^{\mathrm{st}})^2}$, then, assuming that:

$$\varepsilon t_1 \geq \max\left\{\frac{8}{(\widetilde{\Delta}_1^{\mathrm{st}})^2}, 1 + \frac{8\log(t_1)}{(\widetilde{\Delta}_1^{\mathrm{st}})^2}\right\}, \tag{46}$$

the above simplifies to

$$\Pr\left[T_2(n) \geq (1-\varepsilon)n\right] \geq 1 - \left(1 + \frac{1}{3\varepsilon^3}\right)\sum_{r=1}^{R} t_r^{-3} \geq 1 - \left(1 + \frac{1}{3\varepsilon^3}\right)\int_{t_1-1}^{\infty} t^{-3}\mathrm{d}t = 1 - \frac{1 + \frac{1}{3\varepsilon^3}}{2(t_1-1)^2}$$

To guarantee that $\Pr\left[T_2(n) \geq (1-\varepsilon)n\right] > 1 - \delta$, as long as (46) is satisfied, it suffices to take:

$$2(1-\delta)(t_1-1)^2 > 1 + \frac{1}{3\varepsilon^3}. \tag{47}$$

Thus, by the choices of $t_1 = \widetilde{T}_1^0 - T_1^0 + 1$ and $\varepsilon = \frac{\widetilde{T}_1^0}{\widetilde{T}_1^0 + T_2^0}$ from Lemma E.4, the conditions (46) and (47) can be rewritten as:

$$\frac{\widetilde{T}_1^0(\widetilde{T}_1^0 - T_1^0 + 1)}{\widetilde{T}_1^0 + T_2^0} \geq 1 + \frac{8}{(\widetilde{\Delta}_1^{\mathrm{st}})^2}\max\left\{\log(\widetilde{T}_1^0 - T_1^0 + 1), 1\right\}$$

$$\text{and} \quad 2(1-\delta)(\widetilde{T}_1^0 - T_1^0)^2 > 1 + \frac{(\widetilde{T}_1^0 + T_2^0)^3}{3(\widetilde{T}_1^0)^3},$$

where $\widetilde{\Delta}^{\mathrm{st}} = \mu_2\frac{T_2^0}{\widetilde{T}_1^0 + T_2^0} - \mu_1\frac{\widetilde{T}_1^0}{\widetilde{T}_1^0 + T_2^0}$ (notice that $\widetilde{\Delta}^{\mathrm{st}} = \Omega(1)$ whenever $T_1^0 \gg \widetilde{T}_2^0$). Thus, the conditions above are satisfied when:

$$T_1^0 \ll \widetilde{T}_1^0 \ll T_2^0 \ll (\widetilde{T}_1^0)^2.$$

Plugging in:

$$\mu_1 = .9 > .8 = \mu_2 \quad \text{and} \quad T_1^0 = 10, \widetilde{T}_1^0 = 16216, T_2^0 = 16216^{1.5},$$

and thus

$$\varepsilon = \frac{1}{1 + \sqrt{16216}} \approx .007, t_1 = 16207, \mu_1^{\mathrm{bias}}(1) \approx 4 \times 10^{-6}, \widetilde{\mu}_1^{\mathrm{st}} \approx .007, \widetilde{\mu}_2^{\mathrm{st}} \approx .793, \mu_2^{\mathrm{bias}}(1) \approx .799,$$

the above implies that

$$\Pr\left[T_2(n) \geq .99n\right] \geq \Pr\left[T_1(n) \geq (1-\varepsilon)n\right] \geq 1 - \frac{1 + \frac{1}{3\varepsilon^3}}{2(t_1-1)^2} \geq .99,$$

as claimed. $\qquad\square$

Notice that Theorem E.1 follows immediately from Lemma E.6, since

$$R_{\boldsymbol{\nu},\mathsf{UCB}}(n) = \Delta_2 \mathbb{E}\left[T_2(n)\right] \geq .1 \mathbb{E}\left[T_2(n)\mathbb{1}\{T_2(n) \geq .99n\}\right]$$
$$\geq .099 n \Pr\left[T_2(n) \geq .99n\right]$$
$$\geq .098n,$$

as claimed. Thus, we will conclude by proving Lemmas E.3 and E.4.

*Proof of Lemma E.4.* We remark that the only property of the coupling that we use in this proof is the result of Lemma E.3. Thus, we will proceed in this proof without explicitly describing the coupling.

Further, since

$$\mu_2^{\mathrm{bias}}(t) \geq \widetilde{\mu}_2^{\mathrm{st}} \iff \frac{T_2^0 + T_2(t-1)}{T_2^0 + T_1^0 + t - 1} \geq \frac{T_2^0}{T_2^0 + \widetilde{T}_1^0} \iff \frac{\widetilde{T}_1^0}{T_2^0 + \widetilde{T}_1^0} \geq \frac{T_1^0 + T_1(t-1)}{T_2^0 + T_1^0 + t - 1}$$
$$\iff \widetilde{\mu}_1^{\mathrm{st}} \geq \mu_1^{\mathrm{bias}}(t),$$

to prove our claim, it suffices to show that $\widetilde{\mathcal{B}}_R \subseteq \left\{\mu_1^{\mathrm{bias}}(t) \leq \widetilde{\mu}_1^{\mathrm{st}}\right\}$.

To begin, we observe that, by choice of $t_1$, $\mathcal{B}_{t_1}$ (defined in (43)) is true deterministically. Indeed, we have that, for any $t$:

$$\mu_1^{\mathrm{bias}}(t) = \frac{T_1^0 + T_1(t-1)}{T_1^0 + T_2^0 + t - 1}\mu_1 \leq \frac{T_1^0 + t - 1}{T_1^0 + T_2^0 + t - 1}\mu_1$$

Thus, since the RHS above is nondecreasing in $t_1 \geq 1$, we conclude $\mu_1^{\mathrm{bias}}(t) \leq \widetilde{\mu}_1^{\mathrm{st}} = \frac{\widetilde{T}_1^0}{\widetilde{T}_1^0 + T_2^0}$ for all $t \leq t_1 = \widetilde{T}_1^0 - T_1^0 + 1$ (and, thus also that $\mathcal{B}_{t_1}$ is true).

Now, we show that, for any $r \in [0, R]$, if $\mathcal{B}_{t_r}$ is true and $\widetilde{T}_1(t_r) \leq \varepsilon t_r$ (where $\varepsilon = \frac{\widetilde{T}_1^0}{\widetilde{T}_1^0 + T_2^0}$), then $\mathcal{B}_{t_{r+1}}$ is also true. Notice that this establishes our claim that $\widetilde{\mathcal{B}}_{t_R} \subseteq \mathcal{B}_n$, since $\mathcal{B}_{t_1}$ is deterministically true. We proceed by induction on $r$. In the base case of $r = 0$ follows immediately from the fact that $\mathcal{B}_{t_1}$ is deterministically true, as we just showed. Now, assume the claim holds at some $r \geq 0$, i.e., that $\mathcal{B}_{t_r}$ is true and $\widetilde{T}_1(t_r) \leq \varepsilon t_r$. Then, by Lemma E.3, it follows that $T_1(t_r) \leq \varepsilon t_r$. Thus, since $t_{r+1} = t_r + 1$,

$$\mu_1^{\mathrm{bias}}(t_{r+1}) = \mu_1 \frac{T_1^0 + T_1(t_{r+1} - 1)}{T_1^0 + T_2^0 + t_{r+1} - 1} = \mu_1 \frac{T_1^0 + T_1(t_r)}{T_1^0 + T_2^0 + t_r} \leq \mu_1 \frac{T_1^0 + \varepsilon t_r}{T_1^0 + T_2^0 + t_r}.$$

Thus, to show that $\mu_1^{\mathrm{bias}}(t_{r+1}) \leq \widetilde{\mu}_1^{\mathrm{st}}$, it suffices to show:

$$\frac{T_1^0 + \varepsilon t_r}{T_1^0 + T_2^0 + t_r} \leq \frac{\widetilde{T}_1^0}{\widetilde{T}_1^0 + T_2^0}.$$

Observing that:

$$\frac{T_1^0 + \varepsilon t_r}{T_1^0 + T_2^0 + t_r} = \frac{T_1^0}{T_1^0 + T_2^0}\left(\frac{T_1^0 + T_2^0}{T_1^0 + T_2^0 + t_r}\right) + \varepsilon\left(\frac{t_r}{T_1^0 + T_2^0 + t_r}\right) \leq \max\left\{\frac{T_1^0}{T_1^0 + T_2^0}, \varepsilon\right\},$$

the claim follows by our choice of $\varepsilon = \frac{\widetilde{T}_1^0}{\widetilde{T}_1^0 + T_2^0}$. $\qquad\square$

It remains only to prove Lemma E.3.

*Proof of Lemma E.3.* We describe the coupling construction in Algorithm 3. Before we begin, let us introduce some notation that will be used throughout the proofs. Let $\mathcal{H}_t^{\mathrm{bias}}$ (resp., $\mathcal{H}_t^{\mathrm{st}}$) denote the observation history of (UCB) in environment $\boldsymbol{\nu}^{\mathrm{bias}}$ (resp., $\widetilde{\boldsymbol{\nu}}^{\mathrm{st}}$) through time $t$, i.e.,:

$$\mathcal{H}_t^{\mathrm{bias}} = (A_s^{\mathrm{bias}}, Y_s^{\mathrm{bias}})_{s \in [t]} \quad \text{and} \quad \mathcal{H}_t^{\mathrm{st}} = (\widetilde{A}_s^{\mathrm{st}}, \widetilde{Y}_s^{\mathrm{st}})_{s \in [t]}.$$

---

**Algorithm 3** Coupling construction

---

**Require:** Time horizon $n \in \mathbb{N}$, unbiased Bernoulli environment $\boldsymbol{\nu} = (\nu_1, \nu_2)$ with means $0 < \mu_2 < \mu_1 < 1$. Associated biased environment $\boldsymbol{\nu}^{\mathrm{bias}}$ with initial biases $(T_1^0, T_2^0)$. Static environment parameters $\widetilde{\mu}_1^{\mathrm{st}}, \widetilde{\mu}_2^{\mathrm{st}}$ and $\widetilde{T}_1^0$ such that $\mathcal{B}_1$ (the event from (43)) is true, i.e.,

$$\mu_1^{\mathrm{bias}}(1) \leq \mu_1 \frac{\widetilde{T}_1^0}{\widetilde{T}_1^0 + T_2^0} = \widetilde{\mu}_1^{\mathrm{st}} < \widetilde{\mu}_2^{\mathrm{st}} = \mu_2 \frac{T_2^0}{\widetilde{T}_1^0 + T_2^0} \leq \mu_2^{\mathrm{bias}}(1).$$

Two algorithm instances of (UCB) associated with $\boldsymbol{\nu}^{\mathrm{bias}}$ and $\widetilde{\boldsymbol{\nu}}^{\mathrm{st}}$, with associated UCB indices $\mathrm{UCB}_{i,t}^{\mathrm{bias}}$ and $\widetilde{\mathrm{UCB}}_{i,t}^{\mathrm{st}}$, empirical means $\widehat{\mu}_{i,t}^{\mathrm{bias}}$ and $\widehat{\widetilde{\mu}}_{i,t}^{\mathrm{st}}$, and actions $A_t^{\mathrm{bias}}$ and $\widetilde{A}_t^{\mathrm{st}}$ respectively.

**Ensure:** At each time $t \in [n]$, produce samples satisfying:

$$Y_t^{\mathrm{bias}} \sim \mathrm{Bernoulli}(\mu_{A_t^{\mathrm{bias}}}^{\mathrm{bias}}(t)) \quad \text{and} \quad \widetilde{Y}_t^{\mathrm{st}} \sim \mathrm{Bernoulli}(\widetilde{\mu}_{\widetilde{A}_t^{\mathrm{st}}}^{\mathrm{st}}),$$

such that $Y_t^{\mathrm{bias}} \perp\!\!\!\perp \left\{ Y_s^{\mathrm{bias}} \right\}_{s<t} \mid A_t^{\mathrm{bias}}$ and $\widetilde{Y}_t^{\mathrm{st}} \perp\!\!\!\perp \left\{ \widetilde{Y}_s^{\mathrm{st}} \right\}_{s<t} \mid \widetilde{A}_t^{\mathrm{st}}$.

1: Let $t \leftarrow 1$, and let $\mathsf{Q}^{\mathrm{st}}, \mathsf{Q}^{\mathrm{bias}}$ be two empty FIFO queues
2: **while** the event $\mathcal{B}_t$ (defined in Equation (43)) is True **do**
3:     Compute $A_t^{\mathrm{bias}}$ and $\widetilde{A}_t^{\mathrm{st}}$ (use a common tiebreaking rule for the UCB indices).
4:     **if** Both environments select arm 1 (i.e., $A_t^{\mathrm{bias}} = 1 = \widetilde{A}_t^{\mathrm{st}}$) **then**
5:         Since $\mathcal{B}_t$ is true, $\mu_1^{\mathrm{bias}}(t) \leq \widetilde{\mu}_1^{\mathrm{st}}$. Thus, we draw samples as:

$$\widetilde{Y}_t^{\mathrm{st}} \sim \mathrm{Bernoulli}(\widetilde{\mu}_1^{\mathrm{st}}) \quad \text{and} \quad Y_t^{\mathrm{bias}} = \mathrm{Bernoulli}(\mu_1^{\mathrm{bias}}(t)/\widetilde{\mu}_1^{\mathrm{st}}) \widetilde{Y}_t^{\mathrm{st}}$$

6:     **else if** Both environments select arm 2 (i.e., $A_t^{\mathrm{bias}} = 2 = \widetilde{A}_t^{\mathrm{st}}$) **then**
7:         Since $\mathcal{B}_t$ is true, $\widetilde{\mu}_2^{\mathrm{st}} \leq \mu_2^{\mathrm{bias}}(t)$. Thus, we do the reverse of the previous case:

$$Y_t^{\mathrm{bias}} \sim \mathrm{Bernoulli}(\mu_2^{\mathrm{bias}}(t)) \quad \text{and} \quad \widetilde{Y}_t^{\mathrm{st}} = \mathrm{Bernoulli}(\widetilde{\mu}_2^{\mathrm{st}}/\mu_2^{\mathrm{bias}}(t)) Y_t^{\mathrm{bias}}$$

8:     **else if** $A_t^{\mathrm{bias}} = 2$, $\widetilde{A}_t^{\mathrm{st}} = 1$ **then**
9:         Note $T_2(\cdot)$ increases but $\widetilde{T}_2(\cdot)$ does not. This is the good case for our analysis. Take:

$$\widetilde{Y}_t^{\mathrm{st}} \sim \mathrm{Bernoulli}(\widetilde{\mu}_1^{\mathrm{st}}) \quad \text{and} \quad Y_t^{\mathrm{bias}} \sim \mathrm{Bernoulli}(\mu_2^{\mathrm{bias}}(t))$$

10:     Add $\widetilde{Y}_t^{\mathrm{st}}$ to $\mathsf{Q}^{\mathrm{st}}$ and $Y_t^{\mathrm{bias}}$ to $\mathsf{Q}^{\mathrm{bias}}$
11:     **else if** $A_t^{\mathrm{bias}} = 1$, $\widetilde{A}_t^{\mathrm{st}} = 2$ and $\mathsf{Q}^{\mathrm{bias}}, \mathsf{Q}^{\mathrm{st}}$ are both non-empty **then**
12:         Note $\widetilde{T}_2(\cdot)$ increases but $T_2(\cdot)$ does not. This is the trickier case for our analysis.
13:         Remove samples $\widetilde{Y}_{t'}^{\mathrm{st}}$ and $Y_{t'}^{\mathrm{bias}}$ from $\mathsf{Q}^{\mathrm{st}}$ and $\mathsf{Q}^{\mathrm{bias}}$, respectively.
14:         Since $\mathcal{B}_{t'}$ and $\mathcal{B}_t$ are true (since $t' < t$), thus $\widetilde{\mu}_2^{\mathrm{st}} \leq \mu_2^{\mathrm{bias}}(t')$ and $\mu_1^{\mathrm{bias}}(t) \leq \widetilde{\mu}_1^{\mathrm{st}}$, take:

$$\widetilde{Y}_t^{\mathrm{st}} = \mathrm{Bernoulli}(\widetilde{\mu}_2^{\mathrm{st}}/\mu_2^{\mathrm{bias}}(t')) Y_{t'}^{\mathrm{bias}} \quad \text{and} \quad Y_t^{\mathrm{bias}} = \mathrm{Bernoulli}(\mu_1^{\mathrm{bias}}(t)/\widetilde{\mu}_1^{\mathrm{st}}) \widetilde{Y}_{t'}^{\mathrm{st}}$$

15:     **else if** $A_t^{\mathrm{bias}} = 1$, $\widetilde{A}_t^{\mathrm{st}} = 2$ and $\mathsf{Q}^{\mathrm{bias}}$ or $\mathsf{Q}^{\mathrm{st}}$ is empty **then**
16:         Break out of loop
17:     **end if**
18:     Update $t \leftarrow t + 1$
19: **end while**
20: Draw samples independently for every remaining $t$:

$$\widetilde{Y}_t^{\mathrm{st}} \sim \mathrm{Bernoulli}(\widetilde{\mu}_{\widetilde{A}_t^{\mathrm{st}}}^{\mathrm{st}}) \quad \text{and} \quad Y_t^{\mathrm{bias}} \sim \mathrm{Bernoulli}(\mu_{A_t^{\mathrm{bias}}}^{\mathrm{bias}}(t))$$

---

Let $\mathcal{F}_t^{\text{bias}} = \sigma\left\{\mathcal{H}_t^{\text{bias}}\right\}$ and $\mathcal{F}_t^{\text{st}} = \sigma\left\{\mathcal{H}_t^{\text{st}}\right\}$ denote the associated $\sigma$-algebras generated by this interaction history.

**Correctness of coupling**. By construction of Algorithm 3, we note that, whenever $\mathcal{B}_t$ is false, the coupling is trivially valid, since the samples are drawn independently. Similarly, whenever $\mathcal{B}_t$ is true and $A_t^{\text{bias}} = 2$, $\widetilde{A}_t^{\text{st}} = 1$ or $A_t^{\text{bias}} = 1$, $\widetilde{A}_t^{\text{st}} = 2$ and one of the queues $\mathsf{Q}^{\text{st}}, \mathsf{Q}^{\text{bias}}$ is empty at $t$, the samples are also drawn independently. In the case when both environments select the same action, notice that $\widetilde{Y}_t^{\text{st}}$ and $Y_t^{\text{bias}}$ are *not* independent. However, their respective marginal distributions are Bernoulli with mean $\widetilde{\mu}_{\widetilde{A}_t^{\text{st}}}^{\text{st}}$ and $\mu_{A_t^{\text{bias}}}^{\text{bias}}(t)$, respectively, since in the case when $A_t^{\text{bias}} = 1 = \widetilde{A}_t^{\text{st}}$:

$$\mathbb{E}\left[\widetilde{Y}_t^{\text{st}} \mid \mathcal{F}_{t-1}\right] = \widetilde{\mu}_1^{\text{st}} \quad \text{and} \quad \mathbb{E}\left[Y_t^{\text{bias}} \mid \mathcal{F}_{t-1}\right] = \mathbb{E}\left[\text{Bernoulli}(\frac{\mu_1^{\text{bias}}(t)}{\widetilde{\mu}_1^{\text{st}}})\widetilde{Y}_t^{\text{st}} \mid \mathcal{F}_{t-1}\right]$$
$$= \frac{\mu_1^{\text{bias}}(t)}{\widetilde{\mu}_1^{\text{st}}}\widetilde{\mu}_1^{\text{st}},$$

and, similarly, when $A_t^{\text{bias}} = 2 = \widetilde{A}_t^{\text{st}}$:

$$\mathbb{E}\left[Y_t^{\text{bias}} \mid \mathcal{F}_{t-1}^{\text{bias}}\right] = \mu_2^{\text{bias}}(t) \quad \text{and} \quad \mathbb{E}\left[\widetilde{Y}_t^{\text{st}} \mid \mathcal{F}_{t-1}^{\text{st}}\right] = \mathbb{E}\left[\text{Bernoulli}(\frac{\widetilde{\mu}_2^{\text{st}}}{\mu_2^{\text{bias}}(t)})Y_t^{\text{bias}} \mid \mathcal{F}_{t-1}^{\text{st}}\right]$$
$$= \frac{\widetilde{\mu}_2^{\text{st}}}{\mu_2^{\text{bias}}(t)}\mu_2^{\text{bias}}(t).$$

Thus, Algorithm 3 produces valid samples in these cases. Finally, in the case when $A_t^{\text{bias}} = 2$, $\widetilde{A}_t^{\text{st}} = 1$, and $\mathsf{Q}^{\text{bias}}, \mathsf{Q}^{\text{st}}$ are non-empty at $t$, let us denote $\widetilde{Y}_{t'}^{\text{st}}$ and $Y_{t'}^{\text{bias}}$ as the samples removed from $\mathsf{Q}^{\text{st}}$ and $\mathsf{Q}^{\text{bias}}$, where $t' < t$ denotes the time these samples were originally added to the queue. Then,

$$\mathbb{E}\left[\widetilde{Y}_t^{\text{st}} \mid \mathcal{F}_{t-1}^{\text{bias}}\right] = \mathbb{E}\left[\text{Bernoulli}(\frac{\widetilde{\mu}_2^{\text{st}}}{\mu_2^{\text{bias}}(t')})Y_{t'}^{\text{bias}} \mid \mathcal{F}_{t-1}^{\text{st}}\right] = \frac{\widetilde{\mu}_2^{\text{st}}}{\mu_2^{\text{bias}}(t')}\mathbb{E}\left[Y_{t'}^{\text{bias}} \mid \mathcal{F}_{t-1}^{\text{st}}\right].$$

Now, of course, $Y_{t'}^{\text{bias}}$ is measurable w.r.t. $\mathcal{F}_{t'}^{\text{bias}}$ (hence also in $\mathcal{F}_{t-1}^{\text{bias}}$ since $t' < t$) by definition. However, since each sample $Y_{t'}^{\text{bias}}$ added to $\mathsf{Q}^{\text{bias}}$ is used at most once to compute a single $\widetilde{Y}_t^{\text{st}}$, it follows that $Y_{t'}^{\text{bias}}$ is *not* measurable w.r.t $\mathcal{F}_{t-1}^{\text{st}}$. Moreover, when a sample is added to $\mathsf{Q}^{\text{bias}}$ at $t'$, it must be the case that $A_{t'}^{\text{bias}} = 2$ by construction. It follows that $\mathbb{E}\left[Y_{t'}^{\text{bias}} \mid \mathcal{F}_{t-1}^{\text{st}}\right] = \mu_2^{\text{bias}}(t')$, and thus that $\widetilde{Y}_t^{\text{st}}$ has the correct marginal distribution. A symmetric argument shows that $Y_t^{\text{bias}}$ also has the correct marginal distribution in this case. We conclude, therefore, that Algorithm 3 is a valid coupling.

**Establishing the claim under this coupling**. Note that we wish to show that, as long as $\mathcal{B}_t$ is true, then $\widetilde{T}_1(s) \geq T_1(s)$ for all $s \leq t$. We will prove the claim via induction on $s$. At time $s = 1$, the claim is true trivially, since (UCB) deterministically selects arm 1 by definition. Now, suppose the claim holds for $1, \ldots, s$, but not at time $s + 1$. If this were true, then it must be that $T_1(s) = \widetilde{T}_1(s)$, and $A_{s+1}^{\text{bias}} = 1$ and $\widetilde{A}_{s+1}^{\text{st}} = 2$ (indeed, this is the only way to have $T_1(s) \leq \widetilde{T}_1(s)$ and $T_1(s) > \widetilde{T}_1(s)$). This implies that:

$$\widetilde{\text{UCB}}_{1,s+1}^{\text{st}} - \text{UCB}_{1,s+1}^{\text{bias}} = \widehat{\widetilde{\mu}}_{1,s}^{\text{st}} + \sqrt{\frac{2\log(s+1)}{T_1(s)}} - \widehat{\mu}_{1,s}^{\text{bias}} - \sqrt{\frac{2\log(s+1)}{T_1(s)}}$$
$$= \widehat{\widetilde{\mu}}_{1,s}^{\text{st}} - \widehat{\mu}_{1,s}^{\text{bias}}$$
$$= \frac{\sum_{\ell \in [s]} \widetilde{Y}_\ell^{\text{st}} \mathbb{1}\{\widetilde{A}_\ell^{\text{st}} = 1\} - Y_\ell^{\text{bias}}\mathbb{1}\{A_\ell^{\text{bias}} = 1\}}{T_1(s)}$$

Let us now examine $\widetilde{Y}_\ell^{\text{st}}\mathbb{1}\{\widetilde{A}_\ell^{\text{st}} = 1\} - Y_\ell^{\text{bias}}\mathbb{1}\{A_\ell^{\text{bias}} = 1\}$ for each $\ell \leq s$. Whenever $\widetilde{A}_\ell^{\text{st}} = 1 = A_\ell^{\text{bias}}$, then Algorithm 3 guarantees that $\widetilde{Y}_\ell^{\text{st}} \geq Y_\ell^{\text{bias}}$. Whenever $A_\ell^{\text{bias}} = 1$ and $\widetilde{A}_\ell^{\text{st}} = 2$, then there are two cases:

**Case 1**: if $\mathsf{Q}^{\text{st}}$ is non-empty at time $\ell$, then let us denote $\ell'$ as the index of the sample from this queue used at time $\ell$. By definition of Algorithm 3, $A_{\ell'}^{\text{bias}} = 2$ and $\widetilde{A}_{\ell'}^{\text{st}} = 1$. Thus:

$$\widetilde{Y}_{\ell'}^{\text{st}}\underbrace{\mathbb{1}\{\widetilde{A}_{\ell'}^{\text{st}} = 1\}}_{=1} - Y_{\ell'}^{\text{bias}}\underbrace{\mathbb{1}\{A_{\ell'}^{\text{bias}} = 1\}}_{=0} + \widetilde{Y}_\ell^{\text{st}}\underbrace{\mathbb{1}\{\widetilde{A}_\ell^{\text{st}} = 1\}}_{=0} - Y_\ell^{\text{bias}}\underbrace{\mathbb{1}\{A_\ell^{\text{bias}} = 1\}}_{=1} = \widetilde{Y}_{\ell'}^{\text{st}} - Y_\ell^{\text{bias}}$$

By construction of Algorithm 3, $\widetilde{Y}_{\ell'}^{\mathrm{st}} \geq Y_{\ell}^{\mathrm{bias}}$.

**Case 2**: if $Q^{\mathrm{st}}$ is empty at time $\ell$, then, WLOG, let us assume $\ell$ is the first such time when this happens. It follows then that $T_1(\ell-1) = \widetilde{T}_1(\ell-1)$ (indeed, $\widetilde{T}_1(\ell-1) - T_1(\ell-1)$ measures the length of the queue at time $\ell-1$). Since $A_{\ell}^{\mathrm{bias}} = 1$ and $\widetilde{A}_{\ell}^{\mathrm{st}} = 2$, we thus have that:

$$T_1(\ell) - \widetilde{T}_1(\ell) = \underbrace{T_1(\ell-1) - \widetilde{T}_1(\ell-1)}_{=0} + \underbrace{\mathbb{1}\{A_{\ell}^{\mathrm{bias}} = 1\}}_{=1} - \underbrace{\mathbb{1}\{\widetilde{A}_{\ell}^{\mathrm{st}} = 1\}}_{=0} > 0,$$

which contradicts the induction hypothesis that $T_1(\ell) \leq \widetilde{T}_1(\ell)$. Therefore, we conclude that Case 2 cannot happen as long as $\mathcal{B}_t$ is true. Taking the results of these cases together, we conclude that

$$\frac{\sum_{\ell \in [s]} \widetilde{Y}_{\ell}^{\mathrm{st}} \mathbb{1}\{\widetilde{A}_{\ell}^{\mathrm{st}} = 1\} - Y_{\ell}^{\mathrm{bias}} \mathbb{1}\{A_{\ell}^{\mathrm{bias}} = 1\}}{T_1(s)} \geq 0 \quad \text{and thus} \quad \widetilde{\mathsf{UCB}}_{1,s+1}^{\mathrm{st}} \geq \mathsf{UCB}_{1,s+1}^{\mathrm{bias}}$$

By symmetric arguments, we also have that:

$$\mathsf{UCB}_{2,s+1}^{\mathrm{bias}} \geq \widetilde{\mathsf{UCB}}_{2,s+1}^{\mathrm{st}}$$

Additionally, by assumption, $\widetilde{A}_{s+1}^{\mathrm{st}} = 2$, so

$$\widetilde{\mathsf{UCB}}_{2,s+1}^{\mathrm{st}} \geq \widetilde{\mathsf{UCB}}_{1,s+1}^{\mathrm{st}}$$

However, these inequalities imply that:

$$\mathsf{UCB}_{2,s+1}^{\mathrm{bias}} \geq \widetilde{\mathsf{UCB}}_{2,s+1}^{\mathrm{st}} \geq \widetilde{\mathsf{UCB}}_{1,s+1}^{\mathrm{st}} \geq \mathsf{UCB}_{1,s+1}^{\mathrm{bias}}$$

Therefore, either (i) all inequalities are equalities, in which case the algorithms make the same decisions (since they use the same tiebreaking rule by Algorithm 3), or (ii) one of the inequalities is strict. In either case, it must be that $A_s^{\mathrm{bias}} = 2$, which contradicts our assumption! Thus, the claim is established. $\qquad\square$

## F. Details on experiment setup

Here, we give additional details on the experiments included in the main body of our paper. All experiments were performed locally on a Mac operating system, using Python 3.9 and PyCharm. In each of the plots below, we average our results and display error bars representing the standard deviation of the estimated quantity.

### F.1. Ignoring the biased feedback

In Figure 3, we demonstrate empirically that ignoring the bias structure of our problem leads to linear regret for many standard bandit algorithms, such as UCB1 (Auer et al., 2002), EXP3 (Auer et al., 1995), and EXP-IX (Kocák et al., 2014). We run each of these algorithms on a 2-armed Bernoulli bandit instance, where $\mu_1 = .4 < .6 = \mu_2$, with bias structure $W_i(t) = \frac{T_i^{\mathrm{bias}}(t-1)}{t^{\mathrm{bias}}-1}$, where the initial number of arm plays for each arm are: $T_2^0 = 10$, and we vary $T_1^0 \in \{1, 3, 5, 10, 15, 20, 25, 30, 40, 50, 70, 90, 200\}$. The time horizon is $n = 20,000$. Each experiment is repeated $r = 50$ times. The $x$-axis of the plot is the initial reweighting for arm 1, $\frac{T_1^0}{T_1^0 + T_2^0}$. The $y$-axis is the empirical probability of the suboptimal arm (arm 1) being pulled more than $n/2$ times, i.e., $\frac{1}{r} \sum_{\ell \in [r]} \mathbb{1}\{T_1(n) > n/2 \text{ on experiment repeat } r\}$.

### F.2. The impacts of debiasing samples

In Figure 4, we demonstrate the challenges for regret minimization for UCB when the bias model is known. Given the bias model $W_{A_t}(t)$ and biased feedback $Y_t$, an algorithm can, by (2), obtain unbiased samples from the reward distribution of arm $A_t$ by computing $Z_{t,A_t} = Y_t W_{A_t}(t)^{-1}$. For this experiment, we take $W_i(t) = \frac{T_i^{\mathrm{bias}}(t-1)}{t^{\mathrm{bias}}-1}$. Thus, even though the sample is unbiased, the variance scales up by a factor of $\left(\frac{t^{\mathrm{bias}}-1}{T_i^{\mathrm{bias}}(t-1)}\right)^2$. Moreover, even when $Y_t$ has bounded support, $Z_{t,A_t}$ may have a support that scales with the time horizon. We consider a 2-armed Bernoulli bandit instance, where

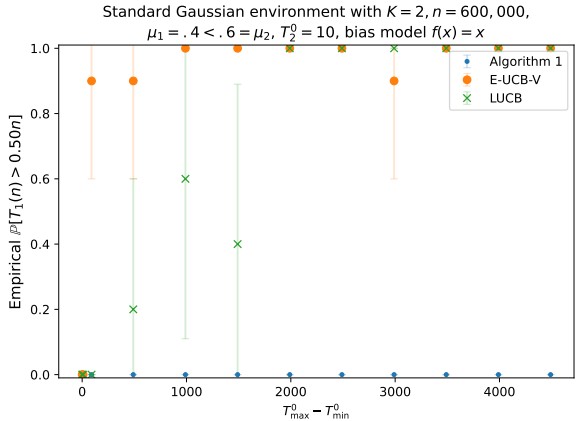

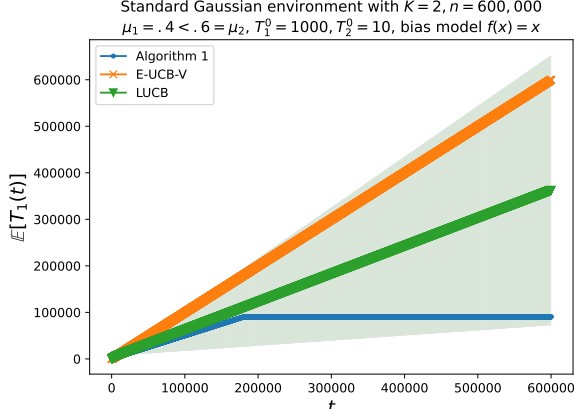

Figure 6: The fraction of times, out of 10 independent experiments, that each algorithm pulls arm 1 more than half of the time, for different values of initial arm pull gaps $T_{\max}^0 - T_{\min}^0$.

Figure 7: The average number of times, over 10 independent experiments, that each algorithm pulls arm 1 during the time horizon $n$

$\mu_1 = .4 < .6 = \mu_2$, and the initial number of times each arm is played is $T_1^0 = 100, T_2^0 = 10$. We consider a time horizon $n = 200,000$, and repeat each experiment 40 times. We run UCB-V (Audibert et al., 2007) in two configurations: (i) one in which it observes the true rewards $X_{t,A_t}$ as feedback (i.e., this is the standard bandit setting with no biases), and (ii) one in which the algorithm knows the bias model, and uses the debiased samples $Z_t$ to compute its estimates. We use the recommended parameter settings from Corollary 1 of that paper. We remark that UCB-V assumes a uniform upper-bound on the rewards. In case (i), this upper bound is 1 (since the rewards are Bernoulli), but in (ii), since the algorithm uses the debiased feedback, the samples used by the algorithm have potentially unbounded support. For this reason, we slightly modify this algorithm to adaptively estimate the support size for each arm based on the largest sample value observed for that arm so far. For each of these algorithms, we plot the empirical $\frac{\mathbb{E}[T_1(t)]}{\sqrt{t}}$ as $t$ varies from 1 to $200,000$.

### F.3. Failure of other optimistic algorithms

In Figures 6 and 7, we compare the performance of Algorithm 1 against two alternative algorithms, Efficient-UCBV (Mukherjee et al., 2018), and an implementation of LUCB (Jamieson & Nowak, 2014). We note that, unlike the phased elimination algorithm used in our paper (Algorithm 1), neither of these algorithms are phased, but instead use optimistic estimates of the arm means for arm selection and elimination at each round. Our experimental results suggest that neither of these algorithms is well-suited for our problem setting, as both suffer linear regret even in a simple biased Gaussian environment for which our algorithm succeeds. This experiment highlights the challenging nature of designing algorithms which provably succeed in our biased setting. Indeed, many optimistic algorithms which work well in standard, unbiased bandit settings fail in our setting.

### F.4. Suboptimal arm pull scaling

In Figures 8 and 9, we demonstrate the scaling of the expected number of suboptimal arm pulls for varying suboptimality gaps, where we fix the mean of the suboptimal arm and vary the mean of the optimal arm. Figure 8 shows that, as the suboptimality gap increases, the expected number of suboptimal arm pulls decreases, as expected. Moreover, the change points occur when the suboptimality gap normalized by the number of arms is $2^{-i}$. This is because the means of the observed feedback for each arm is (roughly) the true mean divided by K before the first arm is eliminated. Figure 9 shows that the expected plays of the suboptimal arm scales proportionally to $\Delta^{-2}$, as predicted by our theory. Note that the step patterns in this plot are a standard consequence of the phases in our phased elimination algorithm.

### F.5. Sensitivity to Lipschitz constant

In Figure 10, we investigate the sensitivity of Algorithm 1 to the Lipschitz constant. We consider a standard Gaussian bandit environment under bias model $f(x) = x^\alpha$ for $\alpha \in [1, 4.5]$. We observe that, for a fixed time horizon, increasing the value of a eventually causes the algorithm to fail. However, increasing the time horizon causes a corresponding shift in the point of

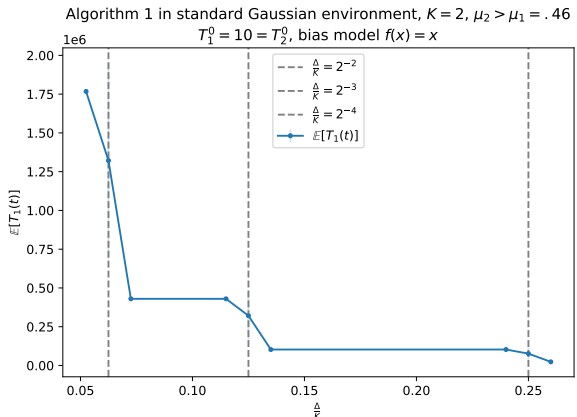

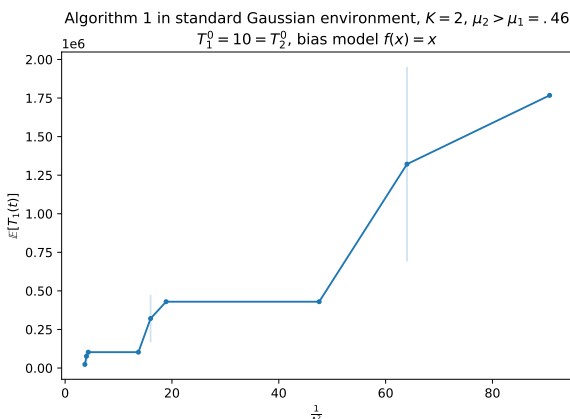

Figure 8: The average number of times, over 3 independent experiments, that Algorithm 1 pulls arm 1 during the time horizon $n$, for varying values of "effective" suboptimality gap $\Delta/K$ (by varying the value of $\mu_2$.

Figure 9: The average number of times, over 3 independent experiments, that Algorithm 1 pulls arm 1 during the time horizon $n$, as we vary $1/\Delta^2$ (by varying the value of $\mu_2$).

failure to a larger value of $\alpha$.

### F.6. Sensitivity to initial arm pull gap

In Figure 11, we investigate the sensitivity to the difference in initial arm pulls in the same setting as in the previous experiment, with bias model $f(x) = x$. Similarly as in the last experiment, for a fixed time horizon, increasing the gap in initial biases eventually causes the algorithm to fail, but this failure point occurs later when the time horizon is increased. Figures 9 and 10 demonstrate the purpose of our requirement that $n$ is sufficiently large in Theorem C.4.

### F.7. Sensitivity to number of arms for different bias models

In Figure 12, we compare the (normalized and square root of) the regret of Algorithm 1 for various bias models, as we increase the number of arms $K$. We observe that the bias model $f(x) = x^2$ has a significantly larger regret scaling than the other bias models, and this difference becomes more pronounced as $K$ increases. The scaling in the sigmoid bias function $f(x) = (1 + \exp(-x))^{-1}$ is larger than the unbiased setting ($f(x) = 1$), but does not depend on $K$, since the sigmoid function is bounded on $[1/2, 1]$ for $x \geq 0$.

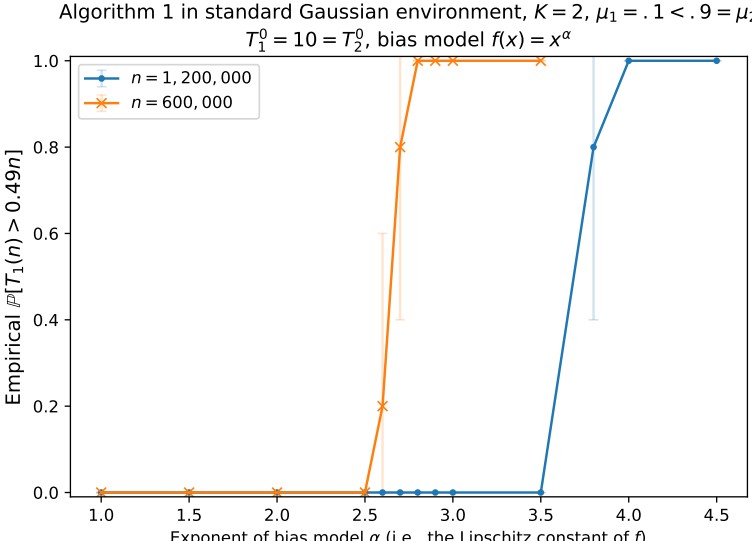

Figure 10: The fraction of times, out of 10 independent experiments, that Algorithm 1 pulls arm 1 more than half of the time, for bias models $f(x) = x^\alpha$ for varying $\alpha$.

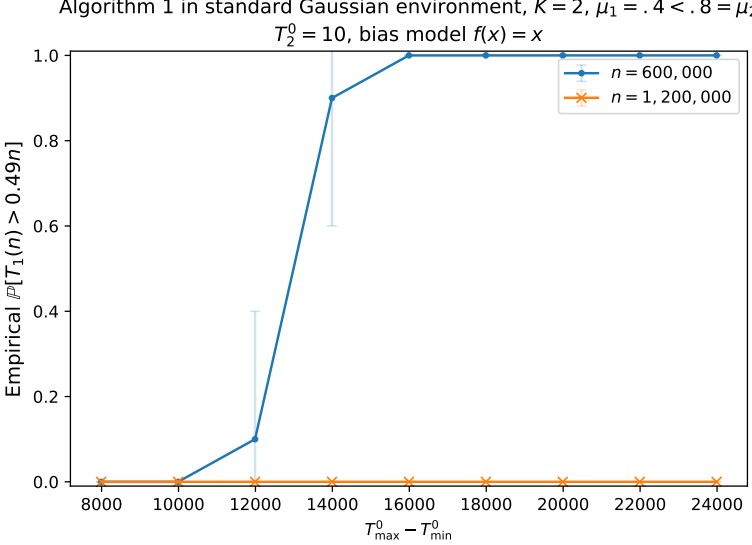

Figure 11: The fraction of times, out of 10 independent experiments, that Algorithm 1 pulls arm 1 more than half of the time, for different values of initial arm pull gaps $T_{\max}^0 - T_{\min}^0$.

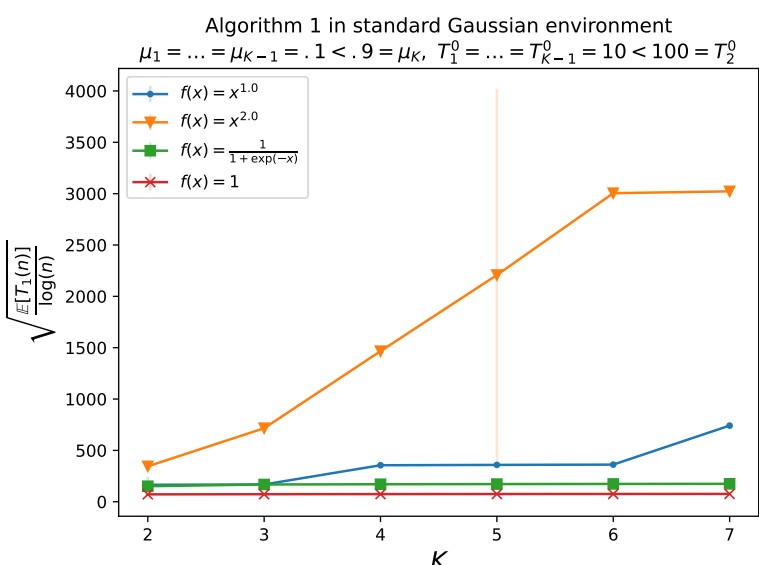

Figure 12: A comparison of the regret of Algorithm 1 (estimated over of 10 independent experiments) for different bias models, as we vary the number of arms $K$.

