# OpenReview forum: "On Mitigating Affinity Bias through Bandits with Evolving Biased Feedback"
_ICML.cc/2025/Conference — ICML 2025 poster_

### Official Review · Reviewer_P4d3 · 2025-03-12

**Overall Recommendation:** 4

**Summary:**

The authors mathematically analyze the feedback loop created by affinity bias in hiring processes and propose strategies to mitigate its effects. They introduce affinity bandits, a model where biased feedback evolves based on the selection frequency of each arm. Their algorithm operates without prior knowledge of bias models or initial biases, and they establish a lower bound that applies in known and unknown bias scenarios.
They show that ignoring bias leads standard bandit models, such as UCB, to suffer linear regret, even in simple cases. Additionally, infrequently choosing some arms makes obtaining an unbiased sample lead to high variance.  Finally, using synthetic data, they evaluate their framework against standard bandit approaches like UCB, showcasing their algorithm's effectiveness in addressing unconscious bias.


- Updated the score +1 after the rebuttal.

**Claims And Evidence:**

The theoretical claims are well supported with mathematical proofs, and authors supplement their theory with empirical evaluations using synthetic data.
I think the authors propose a neat evolving biased feedback model. However, I am uncertain if the chosen motivation is the right one for the algorithm/method design.

**Essential References Not Discussed:**

Given the motivation of the paper being on affinity bias, and hiring processes, some works on homophilous relationships (e.g., "*Identification of Homophily and Preferential Recruitment in Respondent-Driven Sampling*", "*Diversity through Homophily? The Paradox of How Increasing Similarities between Recruiters and Recruits Can Make an Organization More Diverse*", among others) and social networks (e.g. Stoica et al. *Seeding Network Influence in Biased Networks and the Benefits of Diversity*) could have added more context.

**Experimental Designs Or Analyses:**

Yes, in the main paper and the appendix.

**Methods And Evaluation Criteria:**

Yes and no. The theoretical claims are well backed with mathematical proofs and the empirical results support the claims. However, I am uncertain if the chosen motivation is the right one for the algorithm/method design.

**Other Comments Or Suggestions:**

The main paper is well-written.

**Other Strengths And Weaknesses:**

My main concern is with the motivation. While the authors present an interesting evolving biased feedback model, I am unsure whether the chosen motivation aligns well with the algorithm's design.

Integrating the feedback loop aspect is interesting, but the current formulation is somewhat limited, and the problem setup does not fully cater to it. For instance, assuming a static environment is restrictive since real-world conditions change and true values may evolve with the biased ones.  Consider a scenario where Groups A and B start with the same skill levels. If Group A is repeatedly given genuine growth opportunities, its average skill level could increase over time. This compounding effect might eventually result in Group A significantly surpassing Group B, which, by contrast, may have been denied opportunities or even disadvantaged. The reverse is also true, a group's mean skill level and or interest in a given industry could actually reduce over time due to hiring patterns that continuously disadvantage the group. Thus, repeatedly favoring one group not only reinforces biased reward allocations but may also increase the group's true underlying value, and some hiring patterns might potentially reduce a group's true reward.

Furthermore, the approach of modeling groups as arms, each with a true and biased value, inherently embeds and reinforces biases. Even if this differentiation is based on skill level, confounding factors complicate such assumptions. No group is entirely homogeneous, even for ''*objective*'' measurable skills.

**Questions For Authors:**

The authors propose a clean evolving bandits algorithm that they proved to succeed in a biased setting. However, the motivation does not align well with the problem setup. If the authors could refine their motivation to better match the algorithm design, I would consider raising my score to 4/5.

**Relation To Broader Scientific Literature:**

The proposed evolving bandits feedback model is interesting and could potentially be a worthwhile contribution to the community. However, for the use case proposed, I am uncertain if the proposed method is the best way to address the shortcomings in hiring.

**Theoretical Claims:**

Yes, mainly in the main paper and briefly in the appendix

---

> ### Author Rebuttal · Authors · 2025-04-01
>
> We thank the reviewer for their helpful feedback on our paper. We respond to the main points and questions below. Please let us know if you require any further clarifications. We hope that the reviewer will consider raising their score if we have sufficiently addressed their concerns.
>
>
> > Given the motivation of the paper being on affinity bias, and hiring processes, some works on homophilous relationships (e.g., "Identification of Homophily and Preferential Recruitment in Respondent-Driven Sampling", "Diversity through Homophily? The Paradox of How Increasing Similarities between Recruiters and Recruits Can Make an Organization More Diverse", among others) and social networks (e.g. Stoica et al. Seeding Network Influence in Biased Networks and the Benefits of Diversity) could have added more context.
>
>
> Thank you for the pointers to additional related works. We will update our related works section to discuss these works as well.
>
>
> > Integrating the feedback loop aspect is interesting, but the current formulation is somewhat limited, and the problem setup does not fully cater to it. For instance, assuming a static environment is restrictive since real-world conditions change and true values may evolve with the biased ones. Consider a scenario where Groups A and B start with the same skill levels. If Group A is repeatedly given genuine growth opportunities, its average skill level could increase over time. This compounding effect might eventually result in Group A significantly surpassing Group B, which, by contrast, may have been denied opportunities or even disadvantaged. The reverse is also true, a group's mean skill level and or interest in a given industry could actually reduce over time due to hiring patterns that continuously disadvantage the group. Thus, repeatedly favoring one group not only reinforces biased reward allocations but may also increase the group's true underlying value, and some hiring patterns might potentially reduce a group's true reward.
>
> This is an important point and we are glad you brought it up. As we understand your comment, you seem to be describing a problem setting which combines ours (where feedback changes adaptively with past decisions) with various time-varying bandit problems (where the underlying reward distributions are time-varying). In order to study this more challenging setting (which generalizes many already-difficult problems in non-stationary bandits), a necessary first step is to understand what challenges arise when the underlying reward distributions do not change. As our work shows, there are many challenges which arise (e.g., in developing new lower bounds to understand fundamental difficulties, and in designing algorithms). Thus, we believe that our work gives a framework upon which one could study more challenging problem settings, like the one you describe.
>
>
> > Furthermore, the approach of modeling groups as arms, each with a true and biased value, inherently embeds and reinforces biases. Even if this differentiation is based on skill level, confounding factors complicate such assumptions. No group is entirely homogeneous, even for ''objective'' measurable skills.
>
>
> Combined with your previous comment, those are very interesting points which should guide the creation of more nuanced models, which would be closer to reality. We are not aware of any existing work with the level of nuance you are proposing, or even with the level of nuance we have in this work. In this paper, with adding the additional detail of affinity feedback loops, we believe we are making a significant step towards your vision. We discuss the limitations of our work in this regard in the Impact Statement, lines 454-466, and we will add a sentence about this point.

---

> > ### Comment · Reviewer_P4d3 · 2025-04-03
> >
> > I appreciate the authors’ efforts in addressing the reviewers’ feedback. However, I still see a disconnect between the paper’s motivation and technical contributions, a concern also raised by M2kH and, to some extent, TDwV. That said, I agree with the other reviewers that the paper presents novel theoretical contributions. Based on this, I will give the paper a score of 4.

---

### Official Review · Reviewer_HnUV · 2025-03-13

**Overall Recommendation:** 4

**Summary:**

This paper examines how affinity bias influences feedback loops and impacts decision-making in multi-armed bandit problems. The novel formulation assumes biased reward values, where the bias toward an arm depends on the fraction of arms with the same set of trials. The authors establish a new lower bound that is a factor of $K$ larger than in standard MAB problems. Additionally, they present an elimination-style algorithm that nearly achieves this lower bound.

**Claims And Evidence:**

Yes

**Essential References Not Discussed:**

The authors should discuss the connection to rising bandits, another important category of non-stationary MAB problems. A more detailed comparison would strengthen the paper's positioning within the broader literature.

**Experimental Designs Or Analyses:**

Yes

**Methods And Evaluation Criteria:**

Yes

**Other Comments Or Suggestions:**

NA

**Other Strengths And Weaknesses:**

The paper represents a valuable contribution to the community by introducing a novel problem formulation with comprehensive theoretical analysis through both lower and upper bounds. While I have no major concerns about the work, I suggest the authors provide a high-level intuitive explanation of why their elimination algorithm is particularly effective for affinity bandits, which would help readers better understand the algorithm's design principles and why it succeeds where traditional UCB/EXP3 approaches fail.

**Questions For Authors:**

NA

**Relation To Broader Scientific Literature:**

# Contribution

1. **Affinity bandit model**: The authors introduce a valuable variant of non-stationary MAB called affinity bandits. This model addresses biased feedback within a challenging setting where conventional UCB/EXP3 algorithms perform poorly. Beyond job hiring applications, this model shows potential for broader application.
2. **Theoretical lower bound**: The paper provides a comprehensive lower bound for the affinity bandit problem. The additional $K$ factor effectively demonstrates the inherent challenge of this problem formulation.


3. **Near-optimal algorithm**: The authors develop an algorithm that nearly matches the established lower bound, completing the theoretical analysis.

**Theoretical Claims:**

Yes, I checked the main body of the work.

---

> ### Author Rebuttal · Authors · 2025-04-01
>
> We thank the reviewer for their helpful feedback on our paper. We respond to the main points and questions below. Please let us know if you require any further clarifications.
>
>
> > The authors should discuss the connection to rising bandits, another important category of non-stationary MAB problems. A more detailed comparison would strengthen the paper's positioning within the broader literature.
>
>
> Thank you for the suggestion. We will add a reference in our related works.
>
>
> > I suggest the authors provide a high-level intuitive explanation of why their elimination algorithm is particularly effective for affinity bandits, which would help readers better understand the algorithm's design principles and why it succeeds where traditional UCB/EXP3 approaches fail.
>
>
> Note that we discussed the challenges that arise in our model, particularly for algorithms such as UCB and EXP3, in Section 3 (see e.g., Figures 3-4, also Figures 6-7 in the appendix). We aimed to give intuition for why our algorithm works in the second and third paragraphs of Section 4. The essential feature of our algorithm, when compared to, say, UCB, is that it plays a round-robin strategy to ensure that the ordering of arms according to their mean feedback at each round is (essentially) the same as the ordering of the arms according to their (unobserved) mean rewards. We can add additional clarifications on this point in the final version of our paper.

---

### Official Review · Reviewer_M2kH · 2025-03-17

**Overall Recommendation:** 4

**Summary:**

Motivated by affinity biases arising from many decision-making systems, including hiring, the authors *introduced* a stochastic bandit framework that could model the hiring process (affinity bandits setting). In this model, we have $n$ rounds of hiring, and in each arm $\in [K]$ corresponds to a group, and the goal is to minimize pseudo-regret with respect to the true qualification of groups $X_{t,i}$, while the bandit algorithm only observes has bandit feedback access to $Y_{t,i}$. The feedback model in this paper (which is a general one with some plausible properties) is so that upon selection of one arm, roughly speaking the future perceived reward of the selected arm goes higher and the perceived reward of other arms goes down.
- they *empirically* showed that naively ignoring the feedback model or using a simple unbiased operation on feedback can not recover the optimal performance
- they *proved* an instance-dependent lower bound for this problem showing that this problem is harder than the classical stochastic bandit problem
- they provided Algorithm 1 which for large enough $n$ (granted access to $n$ to select $m_r$ appropriately) could *provably* achieve instance dependent regret bound almost matching the lower bound

**Claims And Evidence:**

Theoretical results: Yes

Feedback model: makes sense intuitively, but it is not clear whether it is a novel feedback model, or whether this feedback model is known in literature in computer science or other fields related to biased in perception

**Essential References Not Discussed:**

It is not clear whether the biased feedback model used in this paper (Assumption 2.1) is a new model of perception, or it is a well-known way to model human biases. (I acknowledge there are citations to papers related to bias. However, I do not know whether the feedback model is from those works or it is novel. Also, why do they choose exactly this feedback model among all possible models? In case there are other models of bias in perception in the literature.)

**Experimental Designs Or Analyses:**

No issues.

**Methods And Evaluation Criteria:**

Yes

**Other Comments Or Suggestions:**

If you could include potential directions for future works, that would be greatly helpful for the broader research community. Especially on how we could have more refined models that could capture more details about the hiring process and affinity biases in real-world scenarios, different ways to define arms, and different possible extensions to the feedback model.

**Other Strengths And Weaknesses:**

## Strengths
- The setting is new and valuable. It is possible that some other real-world problems can be formulated this way.
- strong technical contributions (upper bound and lower bound)
- The technical sides are very nicely written
## Weakness
There are not strict weaknesses, but I want to give comments about modelling assumptions. Please let me know if you agree or disagree with these points.


- In this paper, the goal is to minimize regret with respect to picking the group with the best actual value for all rounds. Now, I am wondering if this goal inherently is problematic. Indeed, seeing all members of one group as a single arm can be problematic. Imagine a case where we have two groups. Group A is graduates from a top-rank university, and Group B is graduate students from a medium-rank university (as one of the examples used in this paper). Both groups have the same population size. Assume that 50 percent of group A are good candidates for the job, and 5 percent of group B are good candidates for the job. In this situation, a plausible hiring system would hire candidates from each group proportionally depending on how good candidates are. In this setting, having people from group A being hired all the time and not giving any chance to group B is problematic. However, by the objective of the paper, only picking arm 1 (from group A) has 0 regret. (Although I acknowledge your point in the impact statement.)
- As mentioned in the Impact statement, maybe a more refined and detailed model (contextual bandit) can better capture the affinity bias type behavior in hiring, than the affinity bandit framework introduced in this paper. Maybe among each group A, the is a subset of group $A' \in A$ such that they are good candidates. Indeed, in this example, ideally, if we hire 55 people, the least affinity-biased allocation is to hire 50 from group $A'\in A$ and 5 from group $B' \in B$ just because they are all good.
- If we consider each arm as a set of skills, then as mentioned by authors in the Impact statement, minimizing regret is plausible, although sometimes a hiring candidate might be considered to belong to multiple arms, perhaps if their skillset is broad. Additionally, sometimes we want to hire people from different skill sets as they can complement each other. Again, in this case, minimizing "regret" might not be the best policy.

**Questions For Authors:**

I am very interested to hear your thoughts on my points about modelling assumptions in section **Other Strengths and weaknesses**

**Relation To Broader Scientific Literature:**

The observed reward model is general and it is possible that some other future work uses similar feedback models for different applications. Additionally, it is valuable to extend the work to more realistic settings in which more information about each individual is revealed to the learner, in addition to the group identity (index of the selected arm).

**Theoretical Claims:**

I followed the high-level ideas in Section 3,4. I briefly looked at Section 5.

---

> ### Author Rebuttal · Authors · 2025-04-01
>
> We thank the reviewer for their helpful feedback on our paper. We respond to the main points and questions below. Please let us know if you require any further clarifications.
>
>
> > It is not clear whether the biased feedback model used in this paper (Assumption 2.1) is a new model of perception, or it is a well-known way to model human biases. (I acknowledge there are citations to papers related to bias. However, I do not know whether the feedback model is from those works or it is novel. Also, why do they choose exactly this feedback model among all possible models? In case there are other models of bias in perception in the literature.)
>
>
> We are not aware of prior works which use our Assumption 2.1 as a way of modelling human biases. Our goal in choosing the bias model in Assumption 2.1 is to develop a model which captures some of the essential features of affinity bias, which we outlined on lines 144-149. While understanding regret under more general bias models is an interesting direction for future work, we remark that one cannot hope to minimize regret under arbitrary feedback models which depend jointly on the rewards and fraction of times the arm was played previously. For an example, refer to our response to Reviewer TDwV.
>
>
> > In this paper, the goal is to minimize regret with respect to picking the group with the best actual value for all rounds. Now, I am wondering if this goal inherently is problematic. Indeed, seeing all members of one group as a single arm can be problematic.
>
>
> We agree that in some settings, modelling a group as a single arm can be problematic (indeed, we discuss this in our Impact Statement on lines 454-466). We do agree that studying contextual bandits for the hiring problem is a natural next step. However the current setting already comes with significant technical difficulties. We hope our work can serve as a stepping stone for future work studying more refined models, including contextual bandits.

---

### Official Review · Reviewer_TDwV · 2025-03-17

**Overall Recommendation:** 3

**Summary:**

This work studies a new setting called affinity bandits, which extends the non-stationary bandits setting to capture the affinity biases. They motivate the setting with a hiring feedback loop, where people tend to hire someone with similar features. They made assumptions of the feedback bias in Assumption 2.1 and proved the regret bounds for the elimination type of algorithm.

**Claims And Evidence:**

See "Strengths And Weaknesses"

**Essential References Not Discussed:**

See "Strengths And Weaknesses"

**Experimental Designs Or Analyses:**

See "Strengths And Weaknesses"

**Methods And Evaluation Criteria:**

See "Strengths And Weaknesses"

**Other Comments Or Suggestions:**

See "Strengths And Weaknesses"

**Other Strengths And Weaknesses:**

Strengths:
- This paper is well-written, and well-motivated and the theoretical results are presented in a clear way with good intuitive explanations.
- The problem setting is interesting and useful.
- The theoretical results and intuitive explanation are interesting and novel. The analysis of the elimination algorithm in the non-stationary bias environment can be a separate interest itself.

Weakness:
- The main concern is Assumption 2.1, which is how the weights term is defined in Eq 3. The assumption is selected to make Eq. 5  to be bounded. But it is not clear how realistic this assumption is in general in this Affinity bandits setting, and also in real-world applications (e.g. if we'd like to use it in hiring events). A more detailed explanation and discussion is needed.
- With my concern about Assumption 2.1, it would be useful to add some empirical evaluation (ideally real-world data) to confirm this assumption and the elimination algorithm does work in practice (and is consistent with the theoretical analysis).

**Questions For Authors:**

- Can you provide either theoretical or empirical analysis/discussion if Assumption 2.1 does not hold?
- "Why is the problem difficult"  in Section 3 explains why this setting is difficult for the UCB/EXP3 type of algorithm. Is there any related work in related work (fairness, non-stationary bandits) that would provide a better framework for this new setting and can be compared in terms of theoretical results?

**Relation To Broader Scientific Literature:**

See "Strengths And Weaknesses"

**Theoretical Claims:**

See "Strengths And Weaknesses"

---

> ### Author Rebuttal · Authors · 2025-04-01
>
> We thank the reviewer for their helpful feedback on our paper. We respond to the main points and questions below. Please let us know if you require any further clarifications. We hope that the reviewer will consider raising their score if we have sufficiently addressed their concerns.
>
> > The main concern is Assumption 2.1, which is how the weights term is defined in Eq 3. The assumption is selected to make Eq. 5 to be bounded. But it is not clear how realistic this assumption is in general in this Affinity bandits setting, and also in real-world applications (e.g. if we'd like to use it in hiring events). A more detailed explanation and discussion is needed.
>
> Our goal in Assumption 2.1 is to develop a model which captures the essence of three components of affinity bias, which we outline on lines 144-149. This model gives a tractable framework for studying some of the challenging aspects of sequential decision-making with biased feedback, from the perspective of both upper and lower bounds.
>
> Understanding more general bias models in our setting would be an interesting direction for future work. However, we note that some generalizations of our bias model are intractable. For instance, consider feedback of the form $Y_t = \varphi(X_t, frac_t)$, where $X_t$ is the true reward associated with the arm $A_t$ pulled at time $t$, and $frac_t$ is fraction of times arm $A_t$ was played before time $t. In general, there does not exist a policy with non-trivial regret guarantees for this model (assuming initially unknown bias model).
>
> To see this, consider two bias models (indexed by $a$ and $b$): Consider $\varphi_a(X_t, frac_t) = 1 - X_t$, and $\varphi_b(X_t, frac_t) = X_t$, and take the environment to be a two-armed Bernoulli bandit, where the means are either $\mu_1 = p, \mu_2 = 1-p$ or $\mu_1’ = 1-p, \mu_2’=p$. Suppose that an algorithm achieves sublinear regret simultaneously for means $\mu$ and $\mu’$ under one bias model (say, $\varphi_a$). Then, this immediately implies that the algorithm must suffer linear regret under the other bias model. Indeed, this follows, e.g., by coupling the decisions for the bias model in environment $\mu$ with the decisions of the algorithm for the alternate bias model in environment $\mu’$. Thus, we cannot hope to prove any nontrivial result without some structure on the bias function.
>
> > With my concern about Assumption 2.1, it would be useful to add some empirical evaluation (ideally real-world data) to confirm this assumption and the elimination algorithm does work in practice (and is consistent with the theoretical analysis).
>
> We note that we conducted empirical evaluations of our algorithm (and some other optimistic bandit algorithms) under various bias models in Appendix F (see in particular sections F.3-F.7 and Figures 8-12), in Bernoulli and/or Gaussian bandit environments which satisfy the assumptions in the paper.
>
> While an empirical evaluation on real-world data would be nice, we are not aware of datasets which are readily available and suitable for our problem setting. Moreover, since the main focus and contribution of our paper was characterizing the fundamental difficulties that arise under a simple model for affinity bias, we leave a more thorough empirical evaluation on real-world data to future work.
>
> > "Why is the problem difficult" in Section 3 explains why this setting is difficult for the UCB/EXP3 type of algorithm. Is there any related work in related work (fairness, non-stationary bandits) that would provide a better framework for this new setting and can be compared in terms of theoretical results?
>
> We are not aware of related work which provides a better framework for our setting. Please refer to Appendix A for an extended discussion on some related models to ours.

---

### Decision · Program_Chairs · 2025-05-01

**Decision:**

Accept (poster)

**Comment:**

This paper introduces the affinity bandits framework, a novel extension of the non-stationary stochastic bandit setting designed to model biased feedback loops, with applications such as hiring. The key insight is that feedback—modeled as increasingly biased based on selection history—can lead to self-reinforcing perceptions, making conventional algorithms like UCB or EXP3 ineffective. The authors provide a thorough theoretical analysis, establishing an instance-dependent lower bound and proposing a regret-minimizing elimination-style algorithm that nearly matches this bound (TDwV, M2kH, HnUV). Reviewers commend the paper for its originality, technical rigor, and high-quality exposition. Empirical evaluations with synthetic data complement the theory, offering additional validation of the algorithm’s robustness against biased feedback. The work is recognized as a meaningful step toward modeling real-world decision-making processes that involve implicit bias (P4d3, M2kH).


Despite the strengths, reviewers raise several concerns that warrant further clarification. A key point of critique centers on Assumption 2.1, which governs how bias is introduced through weighted feedback. While mathematically convenient, its realism and applicability to real-world scenarios such as hiring remain uncertain without empirical validation or sensitivity analysis (TDwV, M2kH). Several reviewers (P4d3, M2kH) question whether the paper’s motivation—focused on fairness in hiring—is fully aligned with the algorithmic formulation, suggesting that more detailed models (e.g., contextual bandits) might better capture the heterogeneity of individuals. There is also a call for discussion on related models, such as rising bandits and homophilous feedback structures, as well as broader social dynamics, which the paper currently omits (HnUV, P4d3). Finally, while the theoretical contributions are sound, the paper would benefit from clearer positioning within the literature on bias in perception and a stronger justification for the selected feedback model (M2kH, P4d3). Nonetheless, the overall consensus supports acceptance, given the paper's innovation, technical depth, and potential for future extensions.